# Continuous-time Analysis of Adam in Min-Max Games: Local Convergence and Implicit Gradient Regularization

## Abstract

Although Adam has been widely used in solving min-max games, its dynamical behavior in this setting has not been studied as extensively as in standard minimization. In min-max games, each player aims to minimize their own loss function, which may lead to the intuition that Adam's properties in minimization naturally extend to min-max games with minor adjustments. In this paper, we demonstrate that this belief is incorrect. Specifically, we investigate the local convergence and implicit gradient regularization aspects of Adam in min-max games, focusing on how its first- and second-order momentum parameters influence the dynamics. We show that, in both aspects, the momentum parameters affect Adam's behavior in min-max games in precisely the opposite manner to their effect in minimization. Our methodology is based on the continuous-time analysis techniques commonly used for studying learning algorithms. We further provide numerical experiments to support our theoretical findings.

## 1 Introduction

Min-max games lie at the core of many modern machine learning tasks, including adversarial training (Goodfellow et al., 2014b), GANs (Goodfellow et al., 2014a), and fairness-aware learning (Martinez et al., 2020). While a rich body of theory has been developed for solving min-max games, particularly for optimistic and Extra-gradient methods (Daskalakis et al., 2017; Mokhtari et al., 2020), in practice Adam (Kingma & Ba, 2014) remains the optimizer of choice. For instance, most GANs studies employ Adam for training (Arjovsky et al., 2017; Zhao et al., 2021; Sauer et al., 2023), and Adam is also widely used in adversarial training (Rice et al., 2020; Balunovic & Vechev, 2020; Liu et al., 2020; Kumano et al., 2023). Despite this ubiquity, the theoretical understanding of Adam in min-max games remains far less developed compared to its extensive studied behavior in standard minimization.

The gap between Adam's widespread use in min-max games and the limited theoretical understanding of its behavior in this setting can be partly attributed to its extensive study in minimization (Reddi et al., 2019). It is natural to hope that the intuitions gained from Adam in minimization can be transferred to min-max games with minimal modification. However, previous studies suggest that this intuition may not hold. For instance, Gidel et al. (2019) showed that Adam with a negative first-order momentum can achieve better training performance in GANs, whereas in minimization the first-order momentum of Adam is typically chosen to be nonnegative. Recently, Feng et al. (2025) discovered distinct behaviors of Heavy-ball momentum, a key component of Adam, in min-max games compared to minimization. These observations motivate the following questions:

*How can we theoretically understand Adam's behavior in min-max games? In particular, in which aspects does Adam behave differently in min-max games compared to standard minimization?*

In this work, we try to answer the question by studying Adam's behavior in min-max games through the lens of continuous time analysis. Continuous-time analysis provides differential equations that can accurately approximate the original algorithms and are theoretically tractable. It has been an important tool for studying optimization algorithms since Polyak's seminal work (Polyak, 1964), and future developed in recent years to study the dynamics of various algorithms (Su et al., 2016; Wibisono et al., 2016; Malladi et al., 2022; Suh et al., 2023; Compagnoni et al., 2025).

Our results are twofold, addressing two complementary aspects of Adam in min-max games: **local convergence** and **implicit gradient regularization**. Local convergence characterizes the local behavior of the algorithm when its trajectory is close to an equilibrium, whereas implicit gradient regularization takes a global perspective, describing how the algorithm's trajectories interact with the geometry of the loss landscape. Our main results and contribution can be summarized as follows.

## 1.1 SUMMARY OF CONTRIBUTIONS

**Local Convergence:** We provide quantitative results on the local convergence for both the continuous-time models of Adam (Theorem 4.3) and the Adam algorithm (Theorem 4.4). These results reveal two differences between Adam in min-max games and in standard minimization: First, in min-max games, a *smaller* first-order momentum enhances the algorithm's local convergence over a broader range of step sizes, whereas in minimization, achieving the same effect requires a *larger* first-order momentum (Corollary 4.6). Second, unlike minimization where Adam can converge to a neighborhood of a local minimum under suitable parameters (Zhang et al., 2022b), in min-max games we prove that Adam diverges even for bilinear objectives, regardless of parameter choices (Corollary 4.7). These findings highlight fundamental differences in Adam's dynamics between min-max games and minimization.

**Implicit Gradient Regularization:** We provide qualitative descriptions of how the first- and second-order momentum parameters influence the interaction between the algorithm's trajectories and the flatness of min-max loss landscapes through the analysis of purposed continuous-time models. We find that a *smaller* first-order momentum $\beta$ and a *larger* second-order momentum $\rho$ guide the trajectories toward flatter regions of the loss landscape. Again, this behavior is exactly the opposite of the implicit gradient regularization effect observed in minimization, where achieving a similar effect requires a *larger* $\beta$ and a *smaller* $\rho$ (Cattaneo et al., 2024).

## 1.2 RELATED WORKS

The most close works in the literature to the current work are Rosca et al. (2021) and Feng et al. (2025), which studied the dynamics of (deterministic) gradient descent-ascent and heavy-ball momentum methods in min-max games through the lens of continuous-time analysis. The current work represent a nature extension of their results to the Adam algorithms. In particular, both of Feng et al. (2025) and the current work choose to focus on understanding the local convergence and implicit regularization aspects. Future compare between these works are provided in the Appendix A. In the following, we present more related works around the continuous-time analysis, local convergence in min-max games, and implicit gradient regularization.

**Continuous-time Analysis in Optimization.** Continuous-time analysis was introduced to study momentum in minimization problems, starting with the seminal work of Polyak (1964), and was later extended by Wibisono et al. (2016), especially widely used in stochastic approximation (e.g., Benaim (1996); Borkar (2008) and references therein). Recently, inspired by optimization algorithms that combine momentum and adaptivity such as Adam, continuous-time analysis has also been applied to adaptive methods in minimization, e.g., (Da Silva & Gazeau, 2020; Barakat & Bianchi, 2021; Gadat & Gavra, 2022; Ma et al., 2022; Cattaneo et al., 2024; Compagnoni et al., 2025). For min-max games, Rosca et al. (2021) and Feng et al. (2025) provided continuous-time analyses of vanilla Gradient Descent-Ascent and Heavy-ball momentum, respectively. Suh et al. (2023) employed this methodology to study the anchor acceleration methods. Compagnoni et al. (2024) developed stochastic differential equations for algorithms like Extra-gradient in min-max games under a stochastic setting.

**Local Convergence in Min-Max Games.** Local convergence of learning algorithms in min-max games has received great attention in recent years. Liang & Stokes (2019) analyzed the local convergence of the Gradient Descent-Ascent (GDA) algorithm and its variants, showing the importance of the interaction between players on the dynamics of the algorithms. Fiez & Ratliff (2021) provide a local convergence analysis of GDA under finite timescale separation. Li et al. (2022a) studied the local convergence to a Stackelberg Equilibrium for GDA-based learning algorithms. Zhang et al. (2022a) studied the local convergence of alternating GDA methods and demonstrated

its near-optimal property. Recently, Wang & Chizat (2024) proved the surprising fact that partial curvature generically suffices for the local convergence of GDA. Local convergence results have also been extended to different min-max game settings, such as infinite-dimensional distribution spaces (Wang & Chizat, 2022) and Riemannian manifolds (Zhang, 2025).

**Implicit Gradient Regularization.** The implicit gradient regularization (IGR) effect was first developed for minimization problems. Barrett & Dherin (2021) studied IGR for the gradient descent algorithm. It was later extended to momentum methods by Ghosh et al. (2023), which shows that large momentum parameters usually make algorithms find flatter minima. Recently, Cattaneo et al. (2024) has extended these results to Adam. For min-max games, Rosca et al. (2021) first derived IGR for GDA algorithms, and Feng et al. (2025) extended this approach to momentum methods in min-max games. Inspired by these findings, several recent algorithms that explicitly incorporate gradient regularization to enhance performance have been proposed (Zhang et al., 2023a;b).

## 2 PRELIMINARIES

**Notations.** For matrix $\mathcal{M}$, $\mathrm{Sp}(\mathcal{M})$ denotes the set of its eigenvalues in $\mathbb{C}$. For $\lambda \in \mathrm{Sp}(\mathcal{M})$, $\Re(\lambda)$ and $\Im(\lambda)$ represent the real and imaginary parts of $\lambda$. The notation $\mathcal{M} \preccurlyeq \mathbf{0}$ or $\mathcal{M} \succcurlyeq \mathbf{0}$ means that $\mathcal{M}$ is a negative or positive semi-definite matrix. We use $\mathrm{EigVec}(\mathcal{M})$ to denote the eigenspace of $\mathcal{M}$, and $\mathrm{Ker}(\mathcal{M})$ to represent its kernel space, i.e., $\mathrm{Ker}(\mathcal{M}) = \{\boldsymbol{z} \in \mathbb{C}^d \mid \mathcal{M}\boldsymbol{z} = \mathbf{0}\}$. $\mathcal{I}_d$ denotes the $d$-dimension identity matrix. $\mathrm{Diag}\{\boldsymbol{v}\}$ denotes a diagonal matrix with diagonal elements $\{\boldsymbol{v}_i\}_{i=1}^d$. Without specialization, we use the component-wise multiplication and division of vectors, as well as component-wise addition of vectors.

**Min-Max Games.** A min-max game with smooth loss function $f(\boldsymbol{x}, \boldsymbol{y})$ can be formulated as

$$\min_{\boldsymbol{x} \in \mathbb{R}^{d_1}} \max_{\boldsymbol{y} \in \mathbb{R}^{d_2}} f(\boldsymbol{x}, \boldsymbol{y}) \qquad \text{(Min-Max Games)}$$

If a pair of strategies $(\boldsymbol{x}^*, \boldsymbol{y}^*)$ satisfies $\forall \boldsymbol{x} \in \mathcal{U}$, $f(\boldsymbol{x}, \boldsymbol{y}^*) \geq f(\boldsymbol{x}^*, \boldsymbol{y}^*)$ and $\forall \boldsymbol{y} \in \mathcal{V}$, $f(\boldsymbol{x}^*, \boldsymbol{y}^*) \geq f(\boldsymbol{x}^*, \boldsymbol{y})$ for some $\boldsymbol{x}^*$'s neighborhood $\mathcal{U} \subseteq \mathbb{R}^{d_1}$ and $\boldsymbol{y}^*$'s neighborhood $\mathcal{V} \subseteq \mathbb{R}^{d_2}$, then $(\boldsymbol{x}^*, \boldsymbol{y}^*)$ is called a *local Nash equilibrium*. This is one of the most widely used solution concepts in min-max games, and is the focus of this work Ratliff et al. (2013); Mazumdar et al. (2025). It is also worth noting that local Nash equilibria may not exist in some cases, and other solution concepts have also been proposed in Jin et al. (2020).

**Adam in Min-Max Games.** In min-max games, the $x$-player (resp. $y$-player) aims to minimize (resp. maximize) the objective function $f(\boldsymbol{x}, \boldsymbol{y})$. Accordingly, the Adam algorithm must be adapted: the $x$-player performs gradient descent, while $y$-player performs gradient ascent. We denote this algorithm as Adam Descent-Ascent (Adam-DA) to distinguish it from Adam in minimization:

$$\tilde{\boldsymbol{v}}_{n+1} = \rho\tilde{\boldsymbol{v}}_n + (1-\rho)\left(\nabla_x f(\boldsymbol{x}_n, \boldsymbol{y}_n)\right)^2, \tilde{\boldsymbol{m}}_{n+1} = \beta\tilde{\boldsymbol{m}}_n + (1-\beta)\nabla_x f(\boldsymbol{x}_n, \boldsymbol{y}_n),$$

$$\boldsymbol{x}_{n+1} = \boldsymbol{x}_n - h\frac{\tilde{\boldsymbol{m}}_{n+1}/(1-\beta^{n+1})}{\sqrt{\tilde{\boldsymbol{v}}_{n+1}/(1-\rho^{n+1})} + \epsilon}$$

$$\hat{\boldsymbol{v}}_{n+1} = \rho\hat{\boldsymbol{v}}_n + (1-\rho)\left(\nabla_y f(\boldsymbol{x}_n, \boldsymbol{y}_n)\right)^2, \hat{\boldsymbol{m}}_{n+1} = \beta\hat{\boldsymbol{m}}_n + (1-\beta)\nabla_y f(\boldsymbol{x}_n, \boldsymbol{y}_n),$$

$$\boldsymbol{y}_{n+1} = \boldsymbol{y}_n + h\frac{\hat{\boldsymbol{m}}_{n+1}/(1-\beta^{n+1})}{\sqrt{\hat{\boldsymbol{v}}_{n+1}/(1-\rho^{n+1})} + \epsilon} \qquad \text{(Adam-DA)}$$

Here $h > 0$ is the step size, $\epsilon > 0$ is the numerical stability parameter, $\beta \in (-1, 1)$ is the first-order momentum factor and $\rho \in (0, 1)$ is the second-order momentum factor.

**Local Behaviors of Dynamical Systems.** For a system of differential equations $\dot{\boldsymbol{x}}(t) = g(x)$ where $g: \mathbb{R}^d \to \mathbb{R}^d$ is a differentiable function, let $\tilde{\boldsymbol{x}} \in \mathbb{R}^d$ satisfy $g(\tilde{\boldsymbol{x}}) = 0$. Then the local behavior of the system near $\tilde{\boldsymbol{x}}$ is determined by the eigenvalues of Jacobian $\mathcal{J}_g(\tilde{\boldsymbol{x}}) = \left(\frac{\partial g_i}{\partial x_j}(\tilde{\boldsymbol{x}})\right)_{i,j}$:

**Proposition 2.1.** *(Khalil & Grizzle, 2002) Suppose that $g$ is continuously differentiable. If $\alpha = \max_{\lambda \in \mathrm{Sp}(\mathcal{J}_g)} \Re(\lambda) < 0$, then there exist constants $\delta > 0$ and $C > 0$ such that for all initial conditions satisfying $\|\boldsymbol{x}(0) - \tilde{\boldsymbol{x}}\| \leq \delta$, we have $\|\boldsymbol{x}(t) - \tilde{\boldsymbol{x}}\| \leq Ce^{t\alpha}, \forall t > 0$.*

## 3 CONTINUOUS-TIME MODEL

In this section, we present our continuous-time models for Adam-DA. First, we establish an error bound between the trajectories of discrete-time algorithms and continuous-time models. Then, we compare our model with SignGDA-flow, the min-max adaptation of the continuous-time model proposed by Ma et al. (2022) for Adam in minimization.

The continuous-time model we purposed for Adam-DA in min-max games is the following:

$$\dot{\boldsymbol{x}}(t) = -\mu_\epsilon(\boldsymbol{x}, \boldsymbol{y}) \left( \nabla_x f(\boldsymbol{x}, \boldsymbol{y}) + \frac{h}{2} \mathcal{M}^\mu_{\beta,\rho,\epsilon}(\boldsymbol{x}, \boldsymbol{y}) \cdot \nabla_x \left( \|\nabla_x f(\boldsymbol{x}, \boldsymbol{y})\|_{1,\epsilon} - \|\nabla_y f(\boldsymbol{x}, \boldsymbol{y})\|_{1,\epsilon} \right) \right)$$

$$\dot{\boldsymbol{y}}(t) = \nu_\epsilon(\boldsymbol{x}, \boldsymbol{y}) \left( \nabla_y f(\boldsymbol{x}, \boldsymbol{y}) + \frac{h}{2} \mathcal{M}^\nu_{\beta,\rho,\epsilon}(\boldsymbol{x}, \boldsymbol{y}) \cdot \nabla_y \left( \|\nabla_x f(\boldsymbol{x}, \boldsymbol{y})\|_{1,\epsilon} - \|\nabla_y f(\boldsymbol{x}, \boldsymbol{y})\|_{1,\epsilon} \right) \right).$$

(Continuous Adam-DA)

Here the perturbed $\ell_1$ norm[1] is defined as $\|\boldsymbol{v}\|_{1,\epsilon} := \sum_{i=1}^d \sqrt{\boldsymbol{v}_i^2 + \epsilon}$. Other terms are:

- $\mu_\epsilon(\boldsymbol{x}, \boldsymbol{y}) := \mathrm{Diag} \left\{ \left( \|\partial_{x_j} f(\boldsymbol{x}, \boldsymbol{y})\|_{1,\epsilon}^{-1} \right)_{j=1}^{d_1} \right\}$, $\mathcal{M}^\mu_{\beta,\rho,\epsilon} := \mathcal{K}(\beta, \rho)\mathcal{I}_{d_1} + \frac{\epsilon(1+\rho)}{1-\rho}\mu_\epsilon^2(\boldsymbol{x}, \boldsymbol{y})$

- $\nu_\epsilon(\boldsymbol{x}, \boldsymbol{y}) := \mathrm{Diag} \left\{ \|\left( \partial_{y_i} f(\boldsymbol{x}, \boldsymbol{y})\|_{1,\epsilon}^{-1} \right)_{i=1}^{d_2} \right\}$, $\mathcal{M}^\nu_{\beta,\rho,\epsilon} := \mathcal{K}(\beta, \rho)\mathcal{I}_{d_2} + \frac{\epsilon(1+\rho)}{1-\rho}\nu_\epsilon^2(\boldsymbol{x}, \boldsymbol{y})$

where $\mathcal{K}(\beta, \rho) = (1+\beta)/(1-\beta) - (1+\rho)/(1-\rho)$.

**Theorem 3.1.** *Let $f(\boldsymbol{x}, \boldsymbol{y})$ be a smooth function with bounded derivatives up to the fourth order. Then for any given finite time horizon, the solution trajectories $(\boldsymbol{x}(t), \boldsymbol{y}(t))$ of Continuous Adam-DA is locally $\mathcal{O}(h^3)$-close to the trajectories of Adam-DA after $\max\{\frac{2\log h}{\log|\beta|}, \frac{2\log h}{\log \rho}\}$ steps.*

The proof of Theorem 3.1 relies on the backward error analysis from the numerical analysis literature (Haier et al., 2006), and is inspired by Cattaneo et al. (2024), which derived continuous-time models for the Adam algorithm in the minimization setting to study its implicit regularization effect. However, our equations differ significantly due to the presence of two interacting players inherent to min-max games. Detailed proof of Theorem 3.1 is provided in Appendix B.2.

Theorem 3.1 shows that Continuous Adam-DA approximates Adam-DA with a $\mathcal{O}(h^3)$-local error. To highlight this advantage, we compare it with SignGDA-flow, a min-max adaptation of the continuous-time models proposed by Ma et al. (2022) for deterministic settings and by Compagnoni et al. (2025) for stochastic settings, which approximates Adam with a $\mathcal{O}(h^2)$-local error:

$$\dot{\boldsymbol{x}}(t) = -\mu_\epsilon(\boldsymbol{x}, \boldsymbol{y})\nabla_x f(\boldsymbol{x}, \boldsymbol{y}), \quad \dot{\boldsymbol{y}}(t) = \nu_\epsilon(\boldsymbol{x}, \boldsymbol{y})\nabla_y f(\boldsymbol{x}, \boldsymbol{y}) \qquad \text{(SignGDA-flow)}$$

The name of SignGDA-flow comes from the fact that when $\epsilon \approx 0$, SignGDA-flow depends only on the signs of the partial derivatives. In Figure 1, we present numerical examples from Compagnoni et al. (2024) to compare these three methods. It can be observed that Continuous Adam-DA better approximates Adam-DA, thus highlighting the benefits of $\mathcal{O}(h^3)$-local error.

## 4 LOCAL CONVERGENCE

Unlike minimization algorithms, which almost always converge to a local minimum (Lee et al., 2019), min-max algorithms often fail to converge and may exhibit complex behaviors, such as cycles and chaos (Bailey et al., 2020; Cheung & Piliouras, 2020). However, recent studies show that if the initial strategy pair is sufficiently close to a local equilibrium, convergence can still be achieved (Li et al., 2022a; Wang & Chizat, 2024; Zhang, 2025). This phenomenon is known as *local convergence*. Building upon these findings, we are motivated to investigate the following question:

*How do the parameters in Continuous Adam-DA and Adam-DA influence their local convergence?*

In this section, we answer the above question. In Section 4.1, we describe the Jacobian structure of Continuous Adam-DA at a local equilibrium, which is our main tool to study local convergence. The main results are presented in Section 4.2. Most proofs are deferred to Appendix C.

---

[1]Strictly speaking, $\|\cdot\|_{1,\epsilon}$ is not a norm. The terminology used here follows Cattaneo et al. (2024).

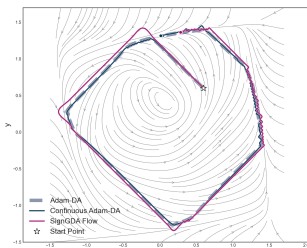 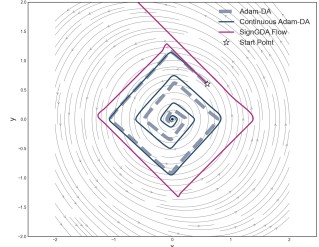 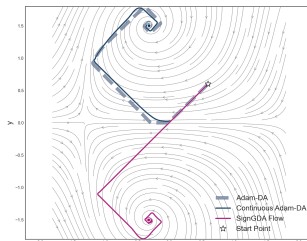

Figure 1: *Trajectories of Adam-DA, Continuous Adam-DA, and SignGDA-flow on three test functions from Compagnoni et al. (2024). Continuous Adam-DA closely approximates Adam-DA. Especially in 1(b) and 1(c), where SignGDA-flow either diverges or approaches to a different equilibrium, while the trajectories of the other two methods remain similar. More details are provided in Appendix B.1.*

### 4.1 STRUCTURES OF JACOBIAN

From dynamical system theory, the Jacobian matrix of a dynamical system completely determines its local behavior around an equilibrium (Hirsch et al., 2012). We first introduce the Jacobian of the following continuous-time model for Gradient Decent-Ascent (GDA), which serves as a fundamental tool for analyzing Continuous Adam-DA and Adam-DA:

$$\dot{\boldsymbol{x}}(t) = -\nabla_x f(\boldsymbol{x}, \boldsymbol{y}), \quad \dot{\boldsymbol{y}}(t) = \nabla_y f(\boldsymbol{x}, \boldsymbol{y}) \qquad \text{(Continuous GDA)}$$

Continuous GDA is the continuous-time model for GDA (Wang & Chizat, 2024). The Jacobian of Continuous GDA at a local Nash equilibrium $(\boldsymbol{x}^*, \boldsymbol{y}^*)$ is defined by

$$\mathcal{J} = \begin{bmatrix} -\nabla_x^2 f(\boldsymbol{x}^*, \boldsymbol{y}^*) & -\nabla_{xy} f(\boldsymbol{x}^*, \boldsymbol{y}^*) \\ \nabla_{yx} f(\boldsymbol{x}^*, \boldsymbol{y}^*) & \nabla_y^2 f(\boldsymbol{x}^*, \boldsymbol{y}^*) \end{bmatrix} \qquad \text{(Jacobian)}$$

Moreover, by the definition of local Nash equilibrium, we have $\nabla_x^2 f(\boldsymbol{x}^*, \boldsymbol{y}^*) \succcurlyeq \boldsymbol{0}$, $\nabla_y^2 f(\boldsymbol{x}^*, \boldsymbol{y}^*) \preccurlyeq \boldsymbol{0}$.

It may seem that Continuous GDA is an over-simplified version of Continuous Adam-DA. Surprisingly, their Jacobian are closely related, as shown in the following proposition:

**Proposition 4.1.** *Let $\mathcal{J}_{\text{Adam}}$ be the Jacobian of Continuous Adam-DA at the local equilibrium $(\boldsymbol{x}^*, \boldsymbol{y}^*)$, then $\mathcal{J}_{Adam}$ is given by a quadratic polynomial of $\mathcal{J}$:*

$$\mathcal{J}_{\text{Adam}} = \frac{1}{\sqrt{\epsilon}} \left( \mathcal{I} - \frac{h(1+\beta)}{2\sqrt{\epsilon}(1-\beta)} \mathcal{J} \right) \mathcal{J}$$

In the following we recall several important perspectives on the structure of $\mathcal{J}$ in previous works. The work of Letcher et al. (2019) introduced a decomposition of $\mathcal{J}$ as a summation of its symmetry part and anti-symmetry part $\mathcal{J} = \mathcal{S} + \mathcal{A}$, where

$$\mathcal{S} = \begin{bmatrix} -\nabla_x^2 f(\boldsymbol{x}^*, \boldsymbol{y}^*) & \boldsymbol{0} \\ \boldsymbol{0}^\top & \nabla_y^2 f(\boldsymbol{x}^*, \boldsymbol{y}^*) \end{bmatrix}, \text{ and } \mathcal{A} = \begin{bmatrix} \boldsymbol{0} & -\nabla_{xy} f(\boldsymbol{x}^*, \boldsymbol{y}^*) \\ \nabla_{yx} f(\boldsymbol{x}^*, \boldsymbol{y}^*) & \boldsymbol{0}^\top \end{bmatrix}$$

In this decomposition, the symmetric part $\mathcal{S}$ describes the part of the dynamics where players independently optimize their own loss functions. The anti-symmetric part $\mathcal{A}$ describes the part where players interact adversarially. In typical min-max games, the magnitude of $\mathcal{A}$ should dominate $\mathcal{S}$; otherwise, the game reduces to a minimization problem where both players minimize their losses independently. This assumption is also used by Wang & Chizat (2024) and Feng et al. (2025) to establish the local convergence of different algorithms in min-max games:

**Assumption 4.2.** *(Wang & Chizat, 2024; Feng et al., 2025) We assume $|\Im(\lambda)| > |\Re(\lambda)|, \ \forall \lambda \in \text{Sp}(\mathcal{J})$ and $\text{EigVec}(\mathcal{A}) \cap \text{Ker}(\mathcal{S}) = \{\boldsymbol{0}\}$.*

Note that since $\mathcal{A}$ is an antisymmetric matrix and has purely imaginary eigenvalues, $|\Im(\lambda)| > |\Re(\lambda)|$, $\forall \lambda \in \mathrm{Sp}(\mathcal{J})$ is a nature corollary of $\mathcal{A} \gg \mathcal{S}$, i.e, the antisymmetric part $\mathcal{A}$ dominates its symmetric part $\mathcal{S}$. Moreover, that the assumption of $\mathrm{EigVec}(\mathcal{A}) \cap \mathrm{Ker}(\mathcal{S}) = \{\mathbf{0}\}$ holds **generically** in the following sense: for any fixed $\mathcal{S} \neq \mathbf{0}$, if $\mathcal{A}$ is sampled independently from an absolutely continuous distribution, then this condition holds with probability one. Thus it is a mild assumption (Wang & Chizat, 2024).

## 4.2 Local Convergence of Adam in Min-Max Games

Now, we present our results on the local convergence. We provide results for both Continuous Adam-DA and Adam-DA, showing that they exhibit similar behavior. This also support the idea that our continuous-time model can accurately predict the behavior of discrete-time algorithms.

**Theorem 4.3.** *Under Assumption 4.2, let $\beta \in (-1, 1)$ and $\rho \in (0, 1)$. Then Continuous Adam-DA achieves local convergence to $(\boldsymbol{x}^*, \boldsymbol{y}^*)$ with an exponential rate iff the step-size $h$ satisfies:*

$$0 < h < \min_{\lambda \in \mathrm{Sp}(\mathcal{J})} \frac{2\sqrt{\epsilon}(1-\beta)|\Re(\lambda)|}{(1+\beta)(\Im(\lambda)^2 - \Re(\lambda)^2)}$$

Under Assumption 4.2, $\Im(\lambda)^2 - \Re(\lambda)^2 > 0$, so the upper bound is positive. By Proposition 2.1, the local convergence of Continuous Adam-DA is governed by the eigenvalues of $\mathcal{J}_{\mathrm{Adam}}$, and the convergence rate equals the largest real part of these eigenvalues. The polynomial relationship between $\mathcal{J}_{\mathrm{Adam}}$ and $\mathcal{J}$ in Proposition 4.1 is key to the proof. The full proof is in Appendix C.2.

It is interesting to ask how well the above theorem for continuous-time models predicts the behavior of the original discrete-time Adam-DA algorithms. We establish the following local convergence result for Adam-DA, and the detailed proof is provided in Appendix C.3:

**Theorem 4.4.** *Under Assumption 4.2, let $\beta \in (-1, 1)$ and $\rho \in (0, 1)$. Then Adam-DA achieves local convergence to the local equilibrium $(\boldsymbol{x}^*, \boldsymbol{y}^*)$ with an exponential rate iff the step-size $h$ satisfies:*

$$0 < h < \min_{\lambda \in \mathrm{Sp}(\mathcal{J})} \frac{2\sqrt{\epsilon}(1-\beta^2)|\Re(\lambda)|}{(1+\beta^2)|\lambda|^2 + 2\beta(\Im(\lambda)^2 - \Re(\lambda)^2)}$$

By comparing Theorem 4.3 and Theorem 4.4, we observe that they share several common properties. We present some of them as corollaries below. This consistency shows that Continuous Adam-DA is a good model of Adam-DA, with the benefit that it is easier to analyze.

**Corollary 4.5.** *For both Continuous Adam-DA and Adam-DA, the presence or absence of local convergence is independent of the second-order momentum factor $\rho$.*

This can be seen from the fact that the upper bounds in Theorem 4.3 and Theorem 4.4 are independent of $\rho$. This phenomenon has also been observed in the local convergence of Adam for minimization problems (Bock & Weiß, 2019; 2021), and we extend it here to min-max games.

Secondly, both in these two theorems, there exist a trade-off between the parameters $h$ and $\beta$:

**Corollary 4.6.** *The upper bound in Theorem 4.3 is a decreasing function of $\beta \in (-1, 1)$. Similarly, the upper bound in Theorem 4.4 is also a decreasing function of $\beta$ when $\beta$ satisfies*

$$\max_{\lambda \in \mathrm{Sp}(\mathcal{J})} \left(|\Re(\lambda)| - |\Im(\lambda)|\right) / \left(|\Re(\lambda)| + |\Im(\lambda)|\right) < \beta < 1 \tag{1}$$

*Therefore, within these regions, using a **smaller** $\beta$ allows Continuous Adam-DA and Adam-DA to achieve local convergence across a broader range of step sizes $h$.*

Note that under Assumption 4.2, the interval in (1) is very close to $(-1, 1)$. Corollary 4.6 reflects a fundamental difference in the role of $\beta$ between min-max games and minimization. In minimization problems, it is known that a **larger** $\beta$ can enable Adam to achieve local convergence with a wider range of step sizes (O'Donoghue & Candes, 2015; Bock & Weiß, 2019). However, as shown by Corollary 4.6, to achieve a similar effect, we need to use a **smaller** $\beta$ in min-max games. We illustrate this point in Figure 2. More experiments are presented in Appendix C.4.

Finally, bilinear games are an important class of min-max games, and many algorithms' theoretical properties are first developed in this setting (Daskalakis et al., 2017; Zhang & Yu, 2020; Wei et al., 2021). Theorem 4.3 and 4.4 imply Adam-DA always diverge in bilinear games:

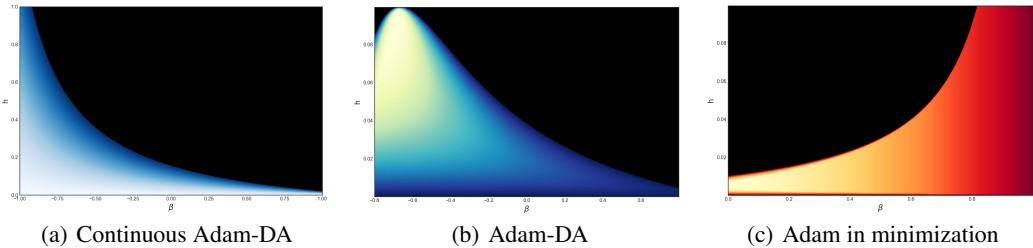

|  (a) Continuous Adam-DA | (b) Adam-DA | (c) Adam in minimization |

Figure 2: *Numerical experiments on quadratic test functions for the local convergence of Continuous Adam-DA, Adam-DA in min-max games, and Adam in minimization. The **black** regions indicate divergence, while lighter regions indicate faster convergence. The X-axis is $\beta$ and Y-axis is $h$. **Smaller** $\beta$ values allow Continuous Adam-DA and Adam-DA to converge over a wider range of step sizes in min-max, while **larger** $\beta$ allow larger $h$ in minimization. These results support Corollary 4.6 and highlight fundamental differences in the role of momentum $\beta$ between min-max and minimization problems. The non-decreasing behavior of $\beta$ for extremely small $\beta$ in 2(b) aligns with Corollary 4.6.*

**Corollary 4.7.** *For bilinear games $f(\boldsymbol{x}, \boldsymbol{y}) = \boldsymbol{x}^\top A \boldsymbol{y}$, Continuous Adam-DA and Adam-DA always diverge regardless of the choice of parameters $\beta$, $\rho$, $\epsilon$ and $h$.*

Corollary 4.7 extends the previous results on the divergence of GDA in the bilinear case to the Adam algorithm. This corollary reflects another fundamental difference between Adam-DA in min-max games and Adam in minimization. In minimization, Adam can always converge to a neighborhood of a local minimum with carefully chosen parameters (Zhang et al., 2022b). However, Corollary 4.7 shows that Adam always diverges in a fundamental class of min-max games, the bilinear games.

## 5 IMPLICIT GRADIENT REGULARIZATION

Implicit gradient regularization (IGR) refers to the terms involving the gradients of loss functions that emerge from the continuous-time models of the discrete-time algorithms. One important example is that in minimization, Barrett & Dherin (2021) derived the following $\mathcal{O}(h^3)$-local error continuous-time model for the Gradient Descent algorithm (GD) $\boldsymbol{x}_{t+1} = \boldsymbol{x}_t - h\nabla_x f(\boldsymbol{x}_t)$:

$$\dot{\boldsymbol{x}}(t) = -\nabla_x f(\boldsymbol{x}) - (h/4)\nabla_x \|\nabla_x f(\boldsymbol{x})\|_2^2, \tag{2}$$

The IGR term in (2) is $-(h/4)\nabla_x \|\nabla_x f(\boldsymbol{x})\|_2^2$, which reveals that besides minimizing function value $f(\boldsymbol{x})$, GD also implicitly minimizing $\|\nabla_x f(\boldsymbol{x})\|_2^2$, thereby guiding the optimization trajectories towards flatter region of the loss landscapes. Here we use the terminology *flatter* regions to refer to regions in the loss landscape with smaller gradient norm, following Barrett & Dherin (2021). Recently, Cattaneo et al. (2024) generalize this approach to the Adam algorithm in minimization. They found in minimization problems, the trajectories of Adam explore flatter regions of the loss landscape with a **larger** $\beta$ and a **smaller** $\rho$. Motivated by these findings, we ask the question:

*How the parameters $\beta$ and $\rho$ affect the interaction between the trajectories of Adam-DA and the flatter regions of loss landscapes in min-max games?*

Our goal in this section is to argue the following thesis, which highlights the distinct behavior of IGR with Adam-DA in min-max games compared to Adam in minimization (Cattaneo et al., 2024):

**Thesis.** In min-max games where players' adversarial interactions dominate the overall dynamics as described in Assumption 4.2, **smaller** $\beta$ and **larger** $\rho$ guides trajectories of Adam-DA towards flatter regions of loss landscapes in terms of $\ell_1$ norm, i.e., regions with lower values of $\|\nabla_x f\|_1 + \|\nabla_y f\|_1$.

The condition of players' adversarial interaction dominates the overall dynamics, i.e., $\nabla_{yx} f(\mathbf{x}, \mathbf{y}) \gg \nabla_x^2 f(\mathbf{x}, \mathbf{y})$ plays an important role in the above discussion. As noted in Assumption 4.2, a numerical indication that such a condition holds is that there are many eigenvalues of Jacobian whose absolute value of the imaginary part is larger than the absolute value of the real part. When this condition does not hold, the overall dynamics will behave more like minimization dynamics, and the implicit regularization effect of Adam should align more closely with the behavior for minimization problems as described in Cattaneo et al. (2024).

In the literature on min–max games in machine learning, it is well understood that the Jacobian of gradient-based updates in GANs training typically exhibits eigenvalues with large imaginary components. This property has been identified as a major source of instability in GANs dynamics, giving rise to the characteristic oscillatory and rotational behaviors observed during training (Mescheder et al., 2017; Liang & Stokes, 2019). A second prominent instance of min–max formulations is adversarial training. However, despite being written in a min–max form, adversarial training behaves quite differently: its optimization dynamics resemble those of standard empirical risk minimization, targeting the minimization of a robust loss rather than the identification of a Nash equilibrium as in GANs (Madry, 2017; Rice et al., 2020). This contrast is clearly reflected in the spectra of the Jacobians of their training dynamics. As shown in Figure 3, eigenvalues of GANs dynamics frequently exhibit substantial imaginary parts, while

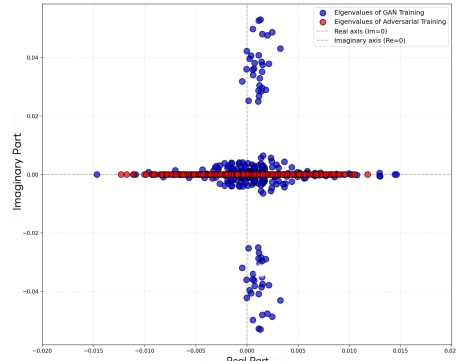

Figure 3: Eigenvalue distribution of GANs and adversarial training on CIFAR-10 dataset. Note that the scales of the axes are different.

those obtained from adversarial training have almost entirely negligible imaginary components. These observations indicate that GAN training satisfies our precondition in which adversarial interactions between players dominate the dynamics, whereas adversarial training operates much more like a minimization problem.

In the following, we present observations from Continuous Adam-DA that lead us to propose above thesis in Section 5.1, and provide experimental results to verify it in Section 5.2.

## 5.1 OBSERVATIONS FROM CONTINUOUS-TIME MODELS

The interesting scenario in understanding implicit gradient regularization occurs when $\epsilon$ is much smaller than the gradient norms[2]. In this scenario, we can let $\epsilon \to 0$, and Continuous Adam-DA reduces to the following equations:

$$\dot{\boldsymbol{x}}(t) = \mu_\epsilon(\boldsymbol{x}, \boldsymbol{y}) \cdot \left[ -\nabla_x f(\boldsymbol{x}, \boldsymbol{y}) - \frac{h}{2}\mathcal{K}(\beta, \rho)\nabla_x\|\nabla_x f(\boldsymbol{x}, \boldsymbol{y})\|_{1,\epsilon} + \frac{h}{2}\mathcal{K}(\beta, \rho)\nabla_x\|\nabla_y f(\boldsymbol{x}, \boldsymbol{y})\|_{1,\epsilon} \right]$$

$$\dot{\boldsymbol{y}}(t) = \nu_\epsilon(\boldsymbol{x}, \boldsymbol{y}) \cdot \left[ \nabla_y f(\boldsymbol{x}, \boldsymbol{y}) + \frac{h}{2}\mathcal{K}(\beta, \rho)\nabla_y\|\nabla_x f(\boldsymbol{x}, \boldsymbol{y})\|_{1,\epsilon} - \frac{h}{2}\mathcal{K}(\beta, \rho)\nabla_y\|\nabla_y f(\boldsymbol{x}, \boldsymbol{y})\|_{1,\epsilon} \right],$$

$$(3)$$

where $\mathcal{K}(\beta, \rho) = (1 + \beta)/(1 - \beta) - (1 + \rho)/(1 - \rho)$ as in Section 3. Here we continue to use $\ell_{1,\epsilon}$ and $\mu_\epsilon, \nu_\epsilon$ to avoid the non-differentiable problem of the $\ell_1$ norm and the zero denominator problem.

We first consider the term $\|\nabla_x f(\boldsymbol{x}, \boldsymbol{y})\|_{1,\epsilon}$. There are two competitive factors affecting its evolution in (3): The $x$-player's equation include a gradient *descent* term on $\mathcal{K}(\beta, \rho)\|\nabla_x f(\boldsymbol{x}, \boldsymbol{y})\|_{1,\epsilon}$, which aims to *minimize* it:

$$-\frac{h}{2}\mathcal{K}(\beta, \rho)\nabla_x\|\nabla_x f(\boldsymbol{x}, \boldsymbol{y})\|_{1,\epsilon} = -\frac{h}{2}\mathcal{K}(\beta, \rho)\left(\nabla_x^2 f(\boldsymbol{x}, \boldsymbol{y}) \cdot \mu_\epsilon(\boldsymbol{x}, \boldsymbol{y}) \cdot \nabla_x f(\boldsymbol{x}, \boldsymbol{y})\right), \quad (4)$$

while $y$-player's equation include a gradient *ascent* term to *maximize* it:

$$\frac{h}{2}\mathcal{K}(\beta, \rho)\nabla_y\|\nabla_x f(\boldsymbol{x}, \boldsymbol{y})\|_{1,\epsilon} = \frac{h}{2}\mathcal{K}(\beta, \rho)\left(\nabla_{yx} f(\boldsymbol{x}, \boldsymbol{y}) \cdot \mu_\epsilon(\boldsymbol{x}, \boldsymbol{y}) \cdot \nabla_x f(\boldsymbol{x}, \boldsymbol{y})\right). \quad (5)$$

The proof of (4) and (5) are provided in Appendix C.1. By comparing (4) and (5), we can predict that the evolution of $\|\nabla_x f(\boldsymbol{x}, \boldsymbol{y})\|_{1,\epsilon}$ in the competition between the two players depends on the relative sizes of $\nabla_x^2 f(\boldsymbol{x}, \boldsymbol{y})$ and $\nabla_{yx} f(\boldsymbol{x}, \boldsymbol{y})$. Moreover, under the condition that the players' interactions dominate the overall dynamics, we have $\nabla_{yx} f(\boldsymbol{x}, \boldsymbol{y}) \gg \nabla_x^2 f(\boldsymbol{x}, \boldsymbol{y})$, i.e., the matrix $\nabla_{yx} f(\boldsymbol{x}, \boldsymbol{y})$ dominates $\nabla_x^2 f(\boldsymbol{x}, \boldsymbol{y})$ in magnitude. This indicates that the $x$-player's impact on the evolution of

---

[2]The default value of $\epsilon$ in PyTorch is $1 \times 10^{-8}$. In practice, gradient norms are much larger than this.

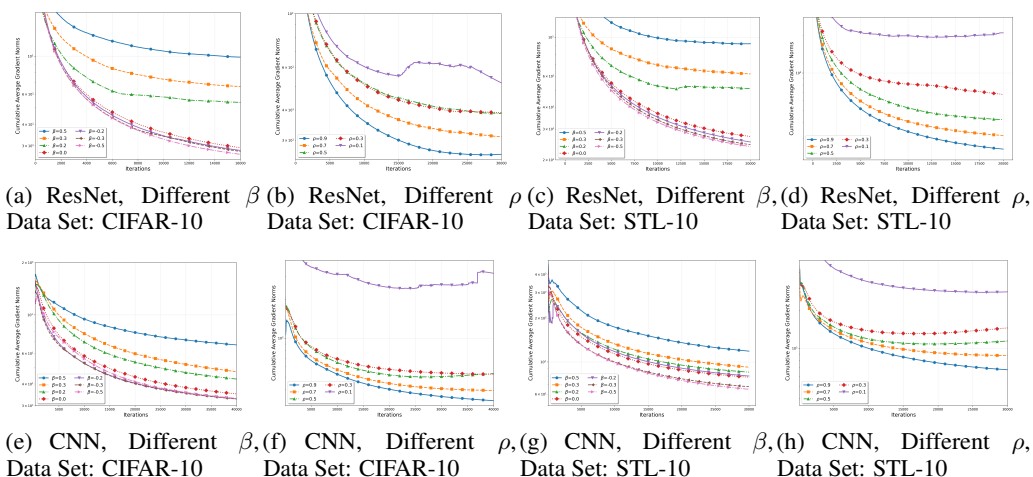

(a) ResNet, Different $\beta$, Data Set: CIFAR-10

(b) ResNet, Different $\rho$, Data Set: CIFAR-10

(c) ResNet, Different $\beta$, Data Set: STL-10

(d) ResNet, Different $\rho$, Data Set: STL-10

(e) CNN, Different $\beta$, Data Set: CIFAR-10

(f) CNN, Different $\rho$, Data Set: CIFAR-10

(g) CNN, Different $\beta$, Data Set: STL-10

(h) CNN, Different $\rho$, Data Set: STL-10

Figure 4: *The $\ell_1$ norm of gradients of Adam-DA with varying $\beta$ and $\rho$ during GANs training. Datasets: CIFAR-10 and STL-10. Architectures: ResNet and CNN. As shown in 4(a), 4(c), 4(e), and 4(g), smaller $\beta$ values result in smaller gradient norms. According to 4(b), 4(d), 4(f), and 4(h), larger $\rho$ values also lead to smaller gradient norms. Both findings support the thesis.*

Table 1: Inception scores under different $\beta$ values.

| Choice of $\beta$ | 0.5 | 0.3 | 0.2 | 0.0 | -0.2 | -0.3 | -0.5 |
|---|---|---|---|---|---|---|---|
| ResNet, CIFAR-10 | 4.160 ±0.122 | 4.698 ±0.158 | 5.217 ±0.232 | 7.022 ±0.282 | 7.020 ±0.255 | **7.087** ±0.358 | 7.002 ±0.313 |
| ResNet, STL-10 | 4.447 ±0.193 | 4.858 ±0.363 | 5.565 ±0.214 | 6.445 ±0.289 | 6.969 ±0.243 | **7.181** ±0.457 | 6.878 ±0.308 |
| CNN, CIFAR-10 | 4.942 ±0.141 | 6.322 ±0.169 | 6.519 ±0.104 | 6.804 ±0.197 | **7.062** ±0.153 | 7.010 ±0.178 | 6.761 ±0.195 |
| CNN, STL-10 | 6.775 ±0.288 | 7.178 ±0.256 | 7.302 ±0.227 | 7.594 ±0.346 | 7.383 ±0.257 | **7.791** ±0.410 | 7.520 ±0.319 |

$\|\nabla_x f(\boldsymbol{x}, \boldsymbol{y})\|_{1,\epsilon}$ is negligible compared to the $y$-player's, so we can focus on (5) to understand how various parameters affect its evolution. From (5), the implicit gradient regularization causes the $y$-player to perform gradient ascent on $\|\nabla_x f(\boldsymbol{x}, \boldsymbol{y})\|_{1,\epsilon}$, effectively trying to maximize it. To encourage the trajectories of Adam-DA to explore flatter regions of the loss landscapes, this effect should be reduced. This can be achieved by selecting a smaller $\beta$ and a larger $\rho$, which lower $\mathcal{K}(\beta, \rho)$; adjusting these parameters accordingly helps minimize its value.

The above argument also applies to the evolution of $\|\nabla_y f(\boldsymbol{x}, \boldsymbol{y})\|_{1,\epsilon}$, due to the symmetry between the two players in (3). In particular, a smaller $\beta$ and a larger $\rho$ are expected to result in lower values of $\|\nabla_y f(\boldsymbol{x}, \boldsymbol{y})\|_{1,\epsilon}$ along the trajectories. Thus, we obtain a justification of the thesis proposed above through a qualitative analysis of Continuous Adam-DA.

## 5.2 EXPERIMENTS

This section presents experimental results on GAN training to support our thesis. GAN training is a min-max game with strong adversarial interactions (Nagarajan & Kolter, 2017; Mescheder et al., 2017), aligning well with our thesis assumptions. It is important to note that our goal here is not to outperform existing methods, but to validate the IGR effect of Adam-DA.

**Experimental Setting.** Our setup generally follows the improved Wasserstein GANs framework (Gulrajani et al., 2017). Data sets include CIFAR-10 and STL-10. Network architectures are ResNet and CNN. We train GANs using the Adam-DA with different $\beta$ and $\rho$. Both generator and

discriminator use the learning rate $2 \times 10^{-4}$. The batch size is $64$. Future details are presented in Appendix D. We use the following cumulative average gradient norms for visualization:

$$\text{AvgS}_{\beta,\rho}(t) = \frac{1}{t} \sum_{s=1}^{t} \left( \|\nabla_x f(\boldsymbol{x}_s, \boldsymbol{y}_s)\|_1 + \|\nabla_y f(\boldsymbol{x}_s, \boldsymbol{y}_s)\|_1 \right).$$

Smaller values of $\text{AvgS}_{\beta,\rho}$ indicate that the algorithms' trajectories are exploring flatter regions of the min-max loss landscapes.

**Experimental Results.** In Figure 4, we show how the cumulative average gradient norms change during training. To evaluate the impact of $\beta$, we fix a value of $\rho = 0.9$ and plot the evolution of these norms for different $\beta$ values in Figures 4(a), 4(c), 4(e), and 4(g). We observe that smaller $\beta$ values lead to smaller gradient norms, supporting our thesis about the role of $\beta$. Similarly, to assess the effect of $\rho$, we fix $\beta = 0$ and plot the norms under different $\rho$ values in Figures 4(b), 4(d), 4(f), and 4(h). Larger $\rho$ values result in smaller gradient norms, which supports our thesis regarding $\rho$. The experimental results are consistency across different architectures and data sets.

It is also interesting to ask how the flatness affects the training performance of Adam-DA. In Table 1 and 2, we present the Inception Scores (IS) of various GANs trained by Adam-DA under parameters presented above. IS is a widely used metric for evaluating the performance of GAN training, with higher scores indicating better performance (Salimans et al., 2016). We find models that exhibit lower cumulative average gradient norms tend to achieve higher IS, which is consistent with the observation that negative momentum helps improve GANs training (Gidel et al., 2019). This is analogous to the phenomena observed in minimization problems, where flatter regions tend to generalize better (Foret et al., 2021).

Table 2: Inception scores under different $\rho$.

| Choice of $\rho$ | 0.9 | 0.7 | 0.5 | 0.3 |
|---|---|---|---|---|
| ResNet, CIFAR-10 | **7.087** $\pm0.358$ | 6.483 $\pm0.129$ | 6.308 $\pm0.270$ | 6.265 $\pm0.198$ |
| ResNet, STL-10 | **7.187** $\pm0.457$ | 6.486 $\pm0.263$ | 6.335 $\pm0.286$ | 5.571 $\pm0.347$ |
| CNN, CIFAR-10 | **7.010** $\pm0.178$ | 6.809 $\pm0.171$ | 6.685 $\pm0.186$ | 6.280 $\pm0.135$ |
| CNN, STL-10 | **7.791** $\pm0.410$ | 7.332 $\pm0.426$ | 6.775 $\pm0.288$ | 6.541 $\pm0.262$ |

## 6 CONCLUSIONS AND FUTURE WORKS

In this paper we provide a continuous-time analysis of Adam in min-max games, focusing on how the first- and second-order momentum parameters influence its dynamics of local convergence and implicit gradient regularization. Surprisingly, in both aspects these parameters affect Adam in exactly the opposite way compared to standard minimization. Several directions remain open for future works. First, our analysis is limited to the deterministic setting, and extending it to the stochastic case would be of great interest, where stochastic differential equations offer powerful tools for continuous-time analysis (Li et al., 2017; Malladi et al., 2022; Compagnoni et al., 2024). Second, our experiments in Section 5 suggest a potential link between the flatness of regions in min-max loss landscapes in terms of gradient norms and the improved performance of model parameters in those regions. Establishing a formal relation between these two, analogous to the connections already established in minimization problems (Haddouche et al., 2025), is an important problem for future study.

**Reproducibility Statement.** All theoretical results in this paper are accompanied by rigorous mathematical proofs, and the assumptions used are explicitly stated. To ensure reproducibility, the experimental code is provided in the supplementary materials.

**The Use of Large Language Models.** This paper utilized a large language model for grammar checking and polishing during its writing process. The code for some experiments in Section 5 of this paper was completed with the assistance of a large language model. The authors of this paper have reviewed all content generated by the large language model and confirm and take responsibility for it.

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

CONTENTS

## A  MORE DISCUSSION ON RELATED WORKS

**Comparison with Rosca et al. (2021) and Feng et al. (2025).**   Now, we would like to make a detailed comparison with Rosca et al. (2021) and Feng et al. (2025), which studied the dynamics of Gradient Descent-Ascent method and Heavy Ball method in min-max games through the lens of continuous-time analysis. Firstly, Rosca et al. (2021) investigated implicit regularization of GDA in min-max games, which involves neither adaptive stepsizes nor any momentum parameters. Next, we mainly focus on clarifying that extending these tools from the relatively simple Heavy-ball momentum method in Feng et al. (2025), which involves only a non-adaptive momentum parameter $\beta$, to Adam is not trivial. Adam's adaptive structure substantially increases the complexity of both the continuous-time model and the discrete-time analysis. For instance, deriving an appropriate $\mathcal{O}(h^3)$ local error equation is significantly more involved than in the Heavy-ball case, which in turn makes the Jacobian analysis considerably more difficult.

Importantly, once these tools are successfully generalized to Adam-DA, they yield new insights that cannot be inferred from Feng et al. (2025). In particular:

- Discrete-time vs. continuous-time behavior: We give a local convergence result for the continuous-time model and also analyze the original discrete-time Adam-DA algorithms. The discrete-time results show that the continuous-time model matches the algorithms well in usual settings. But when the momentum $\beta$ is very small, this match becomes weak, which highlight the potential failure of continuous-time models that do not appear in previous works.

- The role of $\rho$ : Heavy-ball momentum is non-adaptive, so the influence of $\rho$ is entirely absent in Feng et al. (2025). In contrast, we show that in min–max games, larger $\rho$ amplifies Adam's implicit regularization effect—in sharp contrast to the minimization setting, where Cattaneo et al. (2024) shows that smaller $\rho$ strengthens implicit regularization. This is a fundamentally new phenomenon specific to the min–max setting.

- The role of $\epsilon$ : Traditionally, $\epsilon$ is viewed merely as a term preventing division by zero and is assumed to be extremely small. In fact, using large $\epsilon$ is often considered inconsistent with the design philosophy of adaptive gradient methods (Zhang et al., 2022b). However, Theorems 4.3 and 4.4 demonstrate that a large $\epsilon$ can enhance local convergence in min–max games—again revealing behavior entirely absent in the Heavy-ball analysis.

**More Related Works in Min-Max Games.**   Besides converging to local nash equilibrium, other properties of min-max games has been researched. Early rigorous characterizations of local optimality and stability in this setting were developed by Jin et al. (2020), who formalized notions of local min–max optimality in nonconvex–nonconcave landscapes. Fiez et al. (2021) investigated the non-asymptotic convergence rates to $\epsilon$-critical points in several classes of zero-sum games. At the same time, Hsieh et al. (2021) exposed limitations of common min–max algorithms by showing possible convergence to spurious, non-critical sets, highlighting the delicate geometry of min–max flows. Parallel to these algorithmic and discrete-time analyses, recent work has explored adaptive time-scale strategies: Li et al. (2022b) proposed Tiada, a time-scale adaptive method for nonconvex minimax problems, and Yang et al. (2022) developed nested adaptive schemes that achieve parameter-agnostic convergence—complementary approaches that adaptively tune per-variable step sizes and thereby mitigate the need for manual time-scale separation.

## B    ADDITIONAL MATERIALS FOR SECTION 3

### B.1    FUTURE DETAILS OF FIGURE 1

The test functions in Figure 1 comes from (Compagnoni et al., 2024):

- (Figure in the left:) $f_1(x, y) = x(y - 0.45) + \phi(x) - \phi(y), \ \phi(z) = \frac{1}{4}z^2 - \frac{1}{2}z^4 + \frac{1}{6}z^6$.
  Initial point: $(x_0, y_0) = (0.6, 0.6)$. $\beta = 0, \ \rho = 0.5, \ \epsilon = 10^{-6}$. Step size = 0.007.

- (Figure in the middle:)$f_2(x, y) = xy - \frac{1}{10}(\frac{1}{2}y^2 - \frac{1}{4}y^4)$
  Initial point: $(x_0, y_0) = (0.6, 0.6)$. $\beta = -0.3, \ \rho = 0.9, \ \epsilon = 10^{-3}$. Step size = 0.002.

- (Figure in the right:)$f_3(x, y) = \frac{1}{10}x^2 - \frac{1}{10}y^2 + \sin(x)\cos(y)$
  Initial point: $(x_0, y_0) = (0.6, 0.6)$. $\beta = 0.3, \ \rho = 0.5, \ \epsilon = 10^{-4}$. Step size = 0.005.

In the following we also present 30 random initial conditions, and draw the distance between the trajectories of Continuous Adam-DA, SignGDA-flow with Adam-DA. It can be observed that Continuous Adam-DA can better approximate Adam-DA than SignGDA-flow.

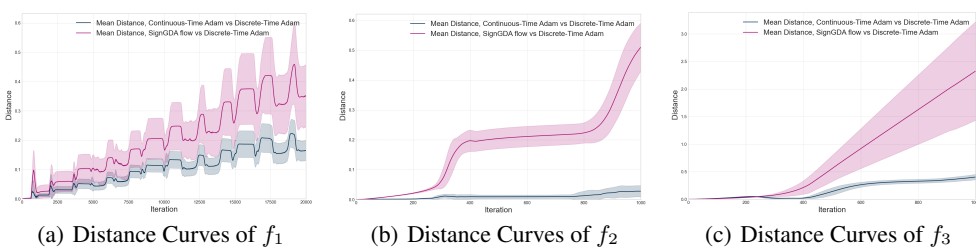

(a) Distance Curves of $f_1$    (b) Distance Curves of $f_2$    (c) Distance Curves of $f_3$

Figure 5: *Figures 5(a), 5(b), and 5(c) show the distances of two continuous-time models between Adam, with results averaged over 30 random initial conditions.*

In the following Figure 6, we present additional experiments with different choice of parameters. The initial points are are chosen to be $(0.3, -0.3)$ in figures of trajectories. For the left figure, $\beta = 0.5, \rho = 0.8, \epsilon = 10^{-6}, h = 0.007$. For the middle figure, $\beta = 0.9, \rho = 0.8, \epsilon = 10^{-3}, h = 0.002$. For the right figure, $\beta = -0.1, \rho = 0.6, \epsilon = 10^{-4}, h = 0.005$

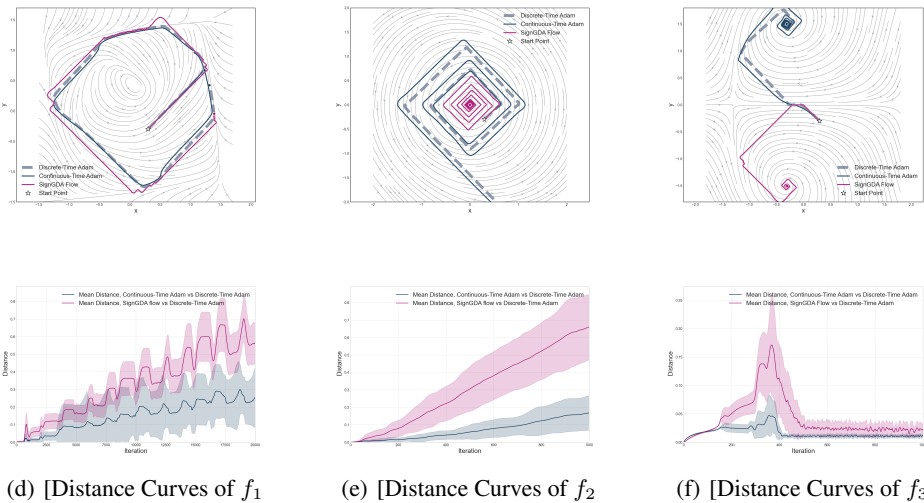

(d) [Distance Curves of $f_1$    (e) [Distance Curves of $f_2$    (f) [Distance Curves of $f_3$

Figure 6: *Additional experiments with different parameters.*

## B.2 Proof of Theorem 3.1

**Notation Statement.** In Appendix B.2, component index and iteration index are used in many quantities. To avoid ambiguity, we use superscripts as coordinate indices (e.g., $\boldsymbol{x}^{(j)}$) and subscripts as iteration indices (e.g., $A_n(\boldsymbol{x}, \boldsymbol{y})$). The derivation of ODEs in Appendix B.2 is written by coordinate-wise, which is the same as the matrix form in Theorem 3.1.

Before providing the formal proof of Theorem 3.1, we provide the sketched proof of Theorem 3.1 consists of several steps:

- Step 1: In Lemma B.2, we derive a family of ODEs that each ODE corresponds to a time region $[t_n, t_{n+1})$ with length $h$ ($h$ is the stepsize of Adam-DA). The solutions of these ODEs achieves good approximation of Adam-DA with local error $\mathcal{O}(h^3)$. It should be pointed out that these ODEs may be not differentiable at time nodes and they are dependent on $n$, which hinder the analysis.

- Step 2: In Lemma B.4, we prove that the family of ODEs in Step 1 will converge to the single equation Continuous Adam-DA with an exponential rate. Due to the exponential rate, after a short time region, we can bound the local error between the family of ODEs and Continuous Adam-DA by $\mathcal{O}(h^3)$. The single ODE Continuous Adam-DA is differentiable everywhere and independent of $n$, which can simplify the analysis of the parameters' behaviors in Min-Max Games.

Define

$$N_n^{\boldsymbol{x}^{(j)}}(\boldsymbol{x}(t_n), \boldsymbol{y}(t_n)) = \frac{\partial f(\boldsymbol{x}(t_n), \boldsymbol{y}(t_n))}{\partial \boldsymbol{x}^{(j)}}, \; D_n^{\boldsymbol{x}^{(j)}}(\boldsymbol{x}(t_n), \boldsymbol{y}(t_n)) = \sqrt{\left(\frac{\partial f(\boldsymbol{x}(t_n), \boldsymbol{y}(t_n))}{\partial \boldsymbol{x}^{(j)}}\right)^2 + \epsilon},$$

$$N_n^{\boldsymbol{y}^{(i)}}(\boldsymbol{x}(t_n), \boldsymbol{y}(t_n)) = \frac{\partial f(\boldsymbol{x}(t_n), \boldsymbol{y}(t_n))}{\partial \boldsymbol{y}^{(i)}}, \; D_n^{\boldsymbol{y}^{(i)}}(\boldsymbol{x}(t_n), \boldsymbol{y}(t_n)) = \sqrt{\left(\frac{\partial f(\boldsymbol{x}(t_n), \boldsymbol{y}(t_n))}{\partial \boldsymbol{y}^{(i)}}\right)^2 + \epsilon},$$

$$Q_n^{\boldsymbol{x}^{(j)}}(\boldsymbol{x}(t_n), \boldsymbol{y}(t_n))$$
$$= \left(\frac{\beta}{1-\beta} - \frac{(n+1)\beta^{n+1}}{1-\beta^{n+1}}\right) \left[\sum_{s=1}^{d_1} \frac{\partial^2 f(\boldsymbol{x}(t_n), \boldsymbol{y}(t_n))}{\partial \boldsymbol{x}^{(j)} \partial \boldsymbol{x}^{(s)}} \frac{N_n^{\boldsymbol{x}^{(s)}}}{D_n^{\boldsymbol{x}^{(s)}}} - \sum_{s=1}^{d_2} \frac{\partial^2 f(\boldsymbol{x}(t_n), \boldsymbol{y}(t_n))}{\partial \boldsymbol{x}^{(j)} \partial \boldsymbol{y}^{(s)}} \frac{N_n^{\boldsymbol{y}^{(s)}}}{D_n^{\boldsymbol{y}^{(s)}}}\right],$$

$$Q_n^{\boldsymbol{y}^{(i)}}(\boldsymbol{x}(t_n), \boldsymbol{y}(t_n))$$
$$= \left(\frac{\beta}{1-\beta} - \frac{(n+1)\beta^{n+1}}{1-\beta^{n+1}}\right) \left[\sum_{s=1}^{d_1} \frac{\partial^2 f(\boldsymbol{x}(t_n), \boldsymbol{y}(t_n))}{\partial \boldsymbol{y}^{(i)} \partial \boldsymbol{x}^{(s)}} \frac{N_n^{\boldsymbol{x}^{(s)}}}{D_n^{\boldsymbol{x}^{(s)}}} - \sum_{s=1}^{d_2} \frac{\partial^2 f(\boldsymbol{x}(t_n), \boldsymbol{y}(t_n))}{\partial \boldsymbol{y}^{(i)} \partial \boldsymbol{y}^{(s)}} \frac{N_n^{\boldsymbol{y}^{(s)}}}{D_n^{\boldsymbol{y}^{(s)}}}\right],$$

$$P_n^{\boldsymbol{x}^{(j)}}(\boldsymbol{x}(t_n), \boldsymbol{y}(t_n)) = \left(\frac{\rho}{1-\rho} - \frac{(n+1)\rho^{n+1}}{1-\rho^{n+1}}\right) \left[\sum_{s=1}^{d_1} \frac{\partial^2 f(\boldsymbol{x}(t_n), \boldsymbol{y}(t_n))}{\partial \boldsymbol{x}^{(j)} \partial \boldsymbol{x}^{(s)}} \frac{\partial f(\boldsymbol{x}(t_n), \boldsymbol{y}(t_n))}{\partial \boldsymbol{x}^{(j)}} \frac{N_n^{\boldsymbol{x}^{(s)}}}{D_n^{\boldsymbol{x}^{(s)}}}\right.$$
$$\left. - \sum_{s=1}^{d_2} \frac{\partial^2 f(\boldsymbol{x}(t_n), \boldsymbol{y}(t_n))}{\partial \boldsymbol{x}^{(j)} \partial \boldsymbol{y}^{(s)}} \frac{\partial f(\boldsymbol{x}(t_n), \boldsymbol{y}(t_n))}{\partial \boldsymbol{x}^{(j)}} \frac{N_n^{\boldsymbol{y}^{(s)}}}{D_n^{\boldsymbol{y}^{(s)}}}\right],$$

$$P_n^{\boldsymbol{y}^{(i)}}(\boldsymbol{x}(t_n), \boldsymbol{y}(t_n)) = \left(\frac{\rho}{1-\rho} - \frac{(n+1)\rho^{n+1}}{1-\rho^{n+1}}\right) \left[\sum_{s=1}^{d_1} \frac{\partial^2 f(\boldsymbol{x}(t_n), \boldsymbol{y}(t_n))}{\partial \boldsymbol{y}^{(i)} \partial \boldsymbol{x}^{(s)}} \frac{\partial f(\boldsymbol{x}(t_n), \boldsymbol{y}(t_n))}{\partial \boldsymbol{y}^{(i)}} \frac{N_n^{\boldsymbol{x}^{(s)}}}{D_n^{\boldsymbol{x}^{(s)}}}\right.$$
$$\left. - \sum_{s=1}^{d_2} \frac{\partial^2 f(\boldsymbol{x}(t_n), \boldsymbol{y}(t_n))}{\partial \boldsymbol{y}^{(i)} \partial \boldsymbol{y}^{(s)}} \frac{\partial f(\boldsymbol{x}(t_n), \boldsymbol{y}(t_n))}{\partial \boldsymbol{y}^{(i)}} \frac{N_n^{\boldsymbol{y}^{(s)}}}{D_n^{\boldsymbol{y}^{(s)}}}\right].$$

$$(6)$$

To simplify the notation, we denote $N_n^{\boldsymbol{x}^{(j)}}(\boldsymbol{x}(t_n), \boldsymbol{y}(t_n))$ by $N_n^{\boldsymbol{x}^{(j)}}$ in the following. Similarly, the same convention applies to $D_n^{\boldsymbol{x}^{(j)}}$, $N_n^{\boldsymbol{y}^{(i)}}$, $D_n^{\boldsymbol{y}^{(i)}}$, $Q_n^{\boldsymbol{x}^{(j)}}$, $P_n^{\boldsymbol{x}^{(j)}}$, $Q_n^{\boldsymbol{y}^{(i)}}$ and $P_n^{\boldsymbol{y}^{(i)}}$.

We proceed to present the following lemma. We will use Lemma B.1 to safely truncate the Taylor-expansion to get an $\mathcal{O}(h^3)$ error.

**Lemma B.1.** *Suppose that $f(\boldsymbol{x}, \boldsymbol{y})$ has bounded derivatives up to the fourth order. Then for any bounded $h$, we can get $\dot{\boldsymbol{x}}^{(j)}(t)$, $\ddot{\boldsymbol{x}}^{(j)}(t)$, $\dddot{\boldsymbol{x}}^{(j)}(t)$ and $\dot{\boldsymbol{y}}^{(i)}(t)$, $\ddot{\boldsymbol{y}}^{(i)}(t)$, $\dddot{\boldsymbol{y}}^{(i)}(t)$ are bounded independent of $h$ for all $j = 1, 2, \cdots, d_1$ and $i = 1, 2, \cdots, d_2$.*

*Proof.* The proof follows directly by the definition of $\dot{\boldsymbol{x}}^{(j)}(t)$, $\ddot{\boldsymbol{x}}^{(j)}(t)$, $\dddot{\boldsymbol{x}}^{(j)}(t)$ and $\dot{\boldsymbol{y}}^{(i)}(t)$, $\ddot{\boldsymbol{y}}^{(i)}(t)$, $\dddot{\boldsymbol{y}}^{(i)}(t)$ together with bounded derivatives of $f(x)$ up to the fourth order. $\square$

**Lemma B.2.** *Let $t_n = nh$ where $h$ is the stepsize of Adam-DA. Then in time region $[t_n, t_{n+1})$, the solutions of the following ODEs:*

$$\frac{d\boldsymbol{x}^{(j)}(t)}{dt}$$

$$= \frac{-\frac{\partial f(\boldsymbol{x}, \boldsymbol{y})}{\partial \boldsymbol{x}^{(j)}}}{\sqrt{\left(\frac{\partial f(\boldsymbol{x}, \boldsymbol{y})}{\partial \boldsymbol{x}^{(j)}}\right)^2 + \epsilon}} - \frac{h}{2\sqrt{\left(\frac{\partial f(\boldsymbol{x}, \boldsymbol{y})}{\partial \boldsymbol{x}^{(j)}}\right)^2 + \epsilon}} \times \left(\frac{\partial}{\partial \boldsymbol{x}^{(j)}} \|\nabla_x f(\boldsymbol{x}, \boldsymbol{y})\|_{1, \epsilon} - \frac{\partial}{\partial \boldsymbol{x}^{(j)}} \|\nabla_y f(\boldsymbol{x}, \boldsymbol{y})\|_{1, \epsilon}\right)$$

$$\times \left[\frac{1 + \beta}{1 - \beta} - \frac{1 + \rho}{1 - \rho} + \frac{\epsilon}{\left(\frac{\partial f(\boldsymbol{x}, \boldsymbol{y})}{\partial \boldsymbol{x}^{(j)}}\right)^2 + \epsilon}\left(\frac{1 + \rho}{1 - \rho} - \frac{(n + 1)\rho^{n+1}}{1 - \rho^{n+1}}\right) - \frac{(n + 1)\beta^{n+1}}{1 - \beta^{n+1}} + \frac{(n + 1)\rho^{n+1}}{1 - \rho^{n+1}}\right],$$

$$\frac{d\boldsymbol{y}^{(i)}(t)}{dt}$$

$$= \frac{\frac{\partial f(\boldsymbol{x}, \boldsymbol{y})}{\partial \boldsymbol{y}^{(i)}}}{\sqrt{\left(\frac{\partial f(\boldsymbol{x}, \boldsymbol{y})}{\partial \boldsymbol{y}^{(i)}}\right)^2 + \epsilon}} + \frac{h}{2\sqrt{\left(\frac{\partial f(\boldsymbol{x}, \boldsymbol{y})}{\partial \boldsymbol{y}^{(i)}}\right)^2 + \epsilon}} \times \left(\frac{\partial}{\partial \boldsymbol{y}^{(i)}} \|\nabla_x f(\boldsymbol{x}, \boldsymbol{y})\|_{1, \epsilon} - \frac{\partial}{\partial \boldsymbol{y}^{(i)}} \|\nabla_y f(\boldsymbol{x}, \boldsymbol{y})\|_{1, \epsilon}\right)$$

$$\times \left[\frac{1 + \beta}{1 - \beta} - \frac{1 + \rho}{1 - \rho} + \frac{\epsilon}{\left(\frac{\partial f(\boldsymbol{x}, \boldsymbol{y})}{\partial \boldsymbol{y}^{(i)}}\right)^2 + \epsilon}\left(\frac{1 + \rho}{1 - \rho} - \frac{(n + 1)\rho^{n+1}}{1 - \rho^{n+1}}\right) - \frac{(n + 1)\beta^{n+1}}{1 - \beta^{n+1}} + \frac{(n + 1)\rho^{n+1}}{1 - \rho^{n+1}}\right].$$

*can approximate the trajectories of Adam-DA with an $\mathcal{O}(h^3)$-local error.*

*Proof.* We rewrite Adam-DA as

$$\boldsymbol{x}_{n+1}^{(j)} = \boldsymbol{x}_n^{(j)} - h\frac{\sum_{k=0}^{n} \frac{\beta^{n-k}(1-\beta)}{1-\beta^{n+1}} \frac{\partial f(\boldsymbol{x}_k, \boldsymbol{y}_k)}{\partial \boldsymbol{x}^{(j)}}}{\sqrt{\sum_{k=0}^{n} \frac{\rho^{n-k}(1-\rho)}{1-\rho^{n+1}} \left(\frac{\partial f(\boldsymbol{x}_k, \boldsymbol{y}_k)}{\partial \boldsymbol{x}^{(j)}}\right)^2 + \epsilon}}, \quad j = 1, \cdots, d_1,$$

$$\boldsymbol{y}_{n+1}^{(i)} = \boldsymbol{y}_n^{(i)} + h\frac{\sum_{k=0}^{n} \frac{\beta^{n-k}(1-\beta)}{1-\beta^{n+1}} \frac{\partial f(\boldsymbol{x}_k, \boldsymbol{y}_k)}{\partial \boldsymbol{y}^{(i)}}}{\sqrt{\sum_{k=0}^{n} \frac{\rho^{n-k}(1-\rho)}{1-\rho^{n+1}} \left(\frac{\partial f(\boldsymbol{x}_k, \boldsymbol{y}_k)}{\partial \boldsymbol{y}^{(i)}}\right)^2 + \epsilon}}, \quad i = 1, \cdots, d_2.$$

In each time interval $[t_n, t_{n+1})$, we define

- $\nabla_x G_n(\boldsymbol{x}, \boldsymbol{y}) = \left(\frac{\partial G_n(\boldsymbol{x}, \boldsymbol{y})}{\partial \boldsymbol{x}^{(1)}}, \cdots, \frac{\partial G_n(\boldsymbol{x}, \boldsymbol{y})}{\partial \boldsymbol{x}^{(j)}}, \cdots, \frac{\partial G_n(\boldsymbol{x}, \boldsymbol{y})}{\partial \boldsymbol{x}^{(d_1)}}\right)$

- $\nabla_y F_n(\boldsymbol{x}, \boldsymbol{y}) = \left(\frac{\partial F_n(\boldsymbol{x}, \boldsymbol{y})}{\partial \boldsymbol{y}^{(1)}}, \cdots, \frac{\partial F_n(\boldsymbol{x}, \boldsymbol{y})}{\partial \boldsymbol{y}^{(i)}}, \cdots, \frac{\partial F_n(\boldsymbol{x}, \boldsymbol{y})}{\partial \boldsymbol{y}^{(d_2)}}\right)$

- $A_n(\boldsymbol{x}, \boldsymbol{y}) = \left( A_n^{(1)}(\boldsymbol{x}, \boldsymbol{y}), \cdots, A_n^{(j)}(\boldsymbol{x}, \boldsymbol{y}), \cdots, A_n^{(d_1)}(\boldsymbol{x}, \boldsymbol{y}) \right)$

- $B_n(\boldsymbol{x}, \boldsymbol{y}) = \left( B_n^{(1)}(\boldsymbol{x}, \boldsymbol{y}), \cdots, B_n^{(i)}(\boldsymbol{x}, \boldsymbol{y}), \cdots, B_n^{(d_2)}(\boldsymbol{x}, \boldsymbol{y}) \right)$

where the terms $G_n(\boldsymbol{x}, \boldsymbol{y}), F_n(\boldsymbol{x}, \boldsymbol{y}), A_n^{(j)}(\boldsymbol{x}, \boldsymbol{y})$ and $B_n^{(i)}(\boldsymbol{x}, \boldsymbol{y})$ are assumed to satisfy the following equations in the time interval $[t_n, t_{n+1}]$:

$$\frac{d\boldsymbol{x}^{(j)}(t)}{dt} = -\frac{\partial G_n(\boldsymbol{x}, \boldsymbol{y})}{\partial \boldsymbol{x}^{(j)}} + h A_n^{(j)}(\boldsymbol{x}, \boldsymbol{y}), \tag{7}$$

$$\frac{d\boldsymbol{y}^{(i)}(t)}{dt} = \frac{\partial F_n(\boldsymbol{x}, \boldsymbol{y})}{\partial \boldsymbol{y}^{(i)}} + h B_n^{(i)}(\boldsymbol{x}, \boldsymbol{y}). \tag{8}$$

Our goal is to find $G_n$, $F_n$, $A_n$ and $B_n$ such that

$$\boldsymbol{x}^{(j)}(t_{n+1}) - \boldsymbol{x}^{(j)}(t_n) = \frac{-h \sum_{k=0}^{n} \frac{\beta^{n-k}(1-\beta)}{1-\beta^{n+1}} \frac{\partial f(\boldsymbol{x}(t_k), \boldsymbol{y}(t_k))}{\partial \boldsymbol{x}^{(j)}}}{\sqrt{\sum_{k=0}^{n} \frac{\rho^{n-k}(1-\rho)}{1-\rho^{n+1}} \left( \frac{\partial f(\boldsymbol{x}(t_k), \boldsymbol{y}(t_k))}{\partial \boldsymbol{x}^{(j)}} \right)^2 + \epsilon}} + \mathcal{O}(h^3), \tag{9}$$

$$\boldsymbol{y}^{(i)}(t_{n+1}) - \boldsymbol{y}^{(i)}(t_n) = \frac{h \sum_{k=0}^{n} \frac{\beta^{n-k}(1-\beta)}{1-\beta^{n+1}} \frac{\partial f(\boldsymbol{x}(t_k), \boldsymbol{y}(t_k))}{\partial \boldsymbol{y}^{(i)}}}{\sqrt{\sum_{k=0}^{n} \frac{\rho^{n-k}(1-\rho)}{1-\rho^{n+1}} \left( \frac{\partial f(\boldsymbol{x}(t_k), \boldsymbol{y}(t_k))}{\partial \boldsymbol{y}^{(i)}} \right)^2 + \epsilon}} + \mathcal{O}(h^3). \tag{10}$$

We firstly do the Taylor expansions of $\frac{\partial f(\boldsymbol{x}(t_k), \boldsymbol{y}(t_k))}{\partial \boldsymbol{x}^{(j)}}$ as follows:

$$\frac{\partial f(\boldsymbol{x}(t_k), \boldsymbol{y}(t_k))}{\partial \boldsymbol{x}^{(j)}} = \frac{\partial f(\boldsymbol{x}(t_n), \boldsymbol{y}(t_n))}{\partial \boldsymbol{x}^{(j)}} + \sum_{s=1}^{d_1} \frac{\partial^2 f(\boldsymbol{x}(t_n), \boldsymbol{y}(t_n))}{\partial \boldsymbol{x}^{(j)} \partial \boldsymbol{x}^{(s)}} \left( \boldsymbol{x}^{(s)}(t_k) - \boldsymbol{x}^{(s)}(t_n) \right)$$

$$+ \sum_{s=1}^{d_2} \frac{\partial^2 f(\boldsymbol{x}(t_n), \boldsymbol{y}(t_n))}{\partial \boldsymbol{x}^{(j)} \partial \boldsymbol{y}^{(s)}} \left( \boldsymbol{y}^{(s)}(t_k) - \boldsymbol{y}^{(s)}(t_n) \right) + \mathcal{O}\left( h^2 \right)$$

$$= \frac{\partial f(\boldsymbol{x}(t_n), \boldsymbol{y}(t_n))}{\partial \boldsymbol{x}^{(j)}} - \sum_{s=1}^{d_1} \frac{\partial^2 f(\boldsymbol{x}(t_n), \boldsymbol{y}(t_n))}{\partial \boldsymbol{x}^{(j)} \partial \boldsymbol{x}^{(s)}} \dot{\boldsymbol{x}}^{(s)}(t_n^+) (n-k) h$$

$$- \sum_{s=1}^{d_2} \frac{\partial^2 f(\boldsymbol{x}(t_n), \boldsymbol{y}(t_n))}{\partial \boldsymbol{x}^{(j)} \partial \boldsymbol{y}^{(s)}} \dot{\boldsymbol{y}}^{(s)}(t_n^+) (n-k) h + \mathcal{O}\left( h^2 \right),$$

$$= \frac{\partial f(\boldsymbol{x}(t_n), \boldsymbol{y}(t_n))}{\partial \boldsymbol{x}^{(j)}} + \sum_{s=1}^{d_1} \frac{\partial^2 f(\boldsymbol{x}(t_n), \boldsymbol{y}(t_n))}{\partial \boldsymbol{x}^{(j)} \partial \boldsymbol{x}^{(s)}} \frac{\partial G_n(\boldsymbol{x}(t_n), \boldsymbol{y}(t_n))}{\partial \boldsymbol{x}^{(s)}} (n-k) h$$

$$- \sum_{s=1}^{d_2} \frac{\partial^2 f(\boldsymbol{x}(t_n), \boldsymbol{y}(t_n))}{\partial \boldsymbol{x}^{(j)} \partial \boldsymbol{y}^{(s)}} \frac{\partial F_n(\boldsymbol{x}(t_n), \boldsymbol{y}(t_n))}{\partial \boldsymbol{y}^{(s)}} (n-k) h + \mathcal{O}\left( h^2 \right), \tag{11}$$

then we obtain that

$$\sum_{k=0}^{n} \frac{\beta^{n-k}(1-\beta)}{1-\beta^{n+1}} \frac{\partial f(\boldsymbol{x}(t_k), \boldsymbol{y}(t_k))}{\partial \boldsymbol{x}^{(j)}} = \sum_{k=0}^{n} \frac{\beta^{n-k}(1-\beta)}{1-\beta^{n+1}} \frac{\partial f(\boldsymbol{x}(t_n), \boldsymbol{y}(t_n))}{\partial \boldsymbol{x}^{(j)}}$$

$$+ h \sum_{k=0}^{n} \frac{\beta^{n-k}(1-\beta)(n-k)}{1-\beta^{n+1}} \sum_{s=1}^{d_1} \frac{\partial^2 f(\boldsymbol{x}(t_n), \boldsymbol{y}(t_n))}{\partial \boldsymbol{x}^{(j)} \partial \boldsymbol{x}^{(s)}} \frac{\partial G_n(\boldsymbol{x}(t_n), \boldsymbol{y}(t_n))}{\partial \boldsymbol{x}^{(s)}}$$

$$- h \sum_{k=0}^{n} \frac{\beta^{n-k}(1-\beta)(n-k)}{1-\beta^{n+1}} \sum_{s=1}^{d_2} \frac{\partial^2 f(\boldsymbol{x}(t_n), \boldsymbol{y}(t_n))}{\partial \boldsymbol{x}^{(j)} \partial \boldsymbol{y}^{(s)}} \frac{\partial F_n(\boldsymbol{x}(t_n), \boldsymbol{y}(t_n))}{\partial \boldsymbol{y}^{(s)}} + \mathcal{O}\left(h^2\right),$$

$$(12)$$

Executing square operation in the both sides of (11), we can obtain

$$\left( \frac{\partial f(\boldsymbol{x}(t_k), \boldsymbol{y}(t_k))}{\partial \boldsymbol{x}^{(j)}} \right)^2 = \left( \frac{\partial f(\boldsymbol{x}(t_n), \boldsymbol{y}(t_n))}{\partial \boldsymbol{x}^{(j)}} \right)^2$$

$$+ 2h \sum_{s=1}^{d_1} (n-k) \frac{\partial^2 f(\boldsymbol{x}(t_n), \boldsymbol{y}(t_n))}{\partial \boldsymbol{x}^{(j)} \partial \boldsymbol{x}^{(s)}} \frac{\partial f(\boldsymbol{x}(t_n), \boldsymbol{y}(t_n))}{\partial \boldsymbol{x}^{(j)}} \frac{\partial G_n(\boldsymbol{x}(t_n), \boldsymbol{y}(t_n))}{\partial \boldsymbol{x}^{(s)}}$$

$$- 2h \sum_{s=1}^{d_2} (n-k) \frac{\partial^2 f(\boldsymbol{x}(t_n), \boldsymbol{y}(t_n))}{\partial \boldsymbol{x}^{(j)} \partial \boldsymbol{y}^{(s)}} \frac{\partial f(\boldsymbol{x}(t_n), \boldsymbol{y}(t_n))}{\partial \boldsymbol{x}^{(j)}} \frac{\partial F_n(\boldsymbol{x}(t_n), \boldsymbol{y}(t_n))}{\partial \boldsymbol{y}^{(s)}} + \mathcal{O}\left(h^2\right).$$

$$(13)$$

Then we can get

$$\sum_{k=0}^{n} \frac{\rho^{n-k}(1-\rho)}{1-\rho^{n+1}} \left( \frac{\partial f(\boldsymbol{x}(t_k), \boldsymbol{y}(t_k))}{\partial \boldsymbol{x}^{(j)}} \right)^2$$

$$= \sum_{k=0}^{n} \frac{\rho^{n-k}(1-\rho)}{1-\rho^{n+1}} \left( \frac{\partial f(\boldsymbol{x}(t_n), \boldsymbol{y}(t_n))}{\partial \boldsymbol{x}^{(j)}} \right)^2$$

$$+ 2h \sum_{k=0}^{n} \frac{\rho^{n-k}(1-\rho)(n-k)}{1-\rho^{n+1}} \sum_{s=1}^{d_1} \frac{\partial^2 f(\boldsymbol{x}(t_n), \boldsymbol{y}(t_n))}{\partial \boldsymbol{x}^{(j)} \partial \boldsymbol{x}^{(s)}} \frac{\partial f(\boldsymbol{x}(t_n), \boldsymbol{y}(t_n))}{\partial \boldsymbol{x}^{(j)}} \frac{\partial G_n(\boldsymbol{x}(t_n), \boldsymbol{y}(t_n))}{\partial \boldsymbol{x}^{(s)}}$$

$$- 2h \sum_{k=0}^{n} \frac{\rho^{n-k}(1-\rho)(n-k)}{1-\rho^{n+1}} \sum_{s=1}^{d_2} \frac{\partial^2 f(\boldsymbol{x}(t_n), \boldsymbol{y}(t_n))}{\partial \boldsymbol{x}^{(j)} \partial \boldsymbol{y}^{(s)}} \frac{\partial f(\boldsymbol{x}(t_n), \boldsymbol{y}(t_n))}{\partial \boldsymbol{x}^{(j)}} \frac{\partial F_n(\boldsymbol{x}(t_n), \boldsymbol{y}(t_n))}{\partial \boldsymbol{y}^{(s)}}$$

$$+ \mathcal{O}\left(h^2\right).$$

$$(14)$$

Then we use the fact that $\left( \sum_{k=0}^{n} a_k h^k \right)^{-\frac{1}{2}} = \frac{1}{\sqrt{a_0}} - \frac{a_1}{2(\sqrt{a_0})^3} h + O(h^2)$ to get

$$\frac{1}{\sqrt{\sum_{k=0}^{n} \frac{\rho^{n-k}(1-\rho)}{1-\rho^{n+1}} \left( \frac{\partial f(\boldsymbol{x}(t_k), \boldsymbol{y}(t_k))}{\partial \boldsymbol{x}^{(j)}} \right)^2 + \epsilon}} = \frac{1}{D_n^{\boldsymbol{x}^{(j)}}} + \frac{h S_n^{\boldsymbol{x}^{(j)}}}{\left( D_n^{\boldsymbol{x}^{(j)}} \right)^3} + \mathcal{O}\left(h^2\right), \qquad (15)$$

where

$$S_n^{\boldsymbol{x}^{(j)}} = \sum_{k=0}^{n} \frac{\rho^{n-k}(1-\rho)\,(n-k)}{1-\rho^{n+1}} \left( \sum_{s=1}^{d_1} \frac{\partial^2 f(\boldsymbol{x}(t_n),\boldsymbol{y}(t_n))}{\partial \boldsymbol{x}^{(j)}\partial \boldsymbol{x}^{(s)}} \frac{\partial f(\boldsymbol{x}(t_n),\boldsymbol{y}(t_n))}{\partial \boldsymbol{x}^{(j)}} \frac{\partial G_n\left(\boldsymbol{x}(t_n),\boldsymbol{y}(t_n)\right)}{\partial \boldsymbol{x}^{(s)}} \right.$$

$$\left. - \sum_{s=1}^{d_2} \frac{\partial^2 f(\boldsymbol{x}(t_k),\boldsymbol{y}(t_k))}{\partial \boldsymbol{x}^{(j)}\partial \boldsymbol{y}^{(s)}} \frac{\partial f(\boldsymbol{x}(t_k),\boldsymbol{y}(t_k))}{\partial \boldsymbol{x}^{(j)}} \frac{\partial F_n\left(\boldsymbol{x}(t_n),\boldsymbol{y}(t_n)\right)}{\partial \boldsymbol{y}^{(s)}} \right),$$

and we have used the equation

$$\sum_{k=0}^{n} \frac{\rho^{n-k}(1-\rho)}{1-\rho^{n+1}} \left( \frac{\partial f(\boldsymbol{x}(t_n),\boldsymbol{y}(t_n))}{\partial \boldsymbol{x}^{(j)}} \right)^2 = \left( \frac{\partial f(\boldsymbol{x}(t_n),\boldsymbol{y}(t_n))}{\partial \boldsymbol{x}^{(j)}} \right)^2.$$

Firstly, we solve $\frac{\partial G_n(\boldsymbol{x},\boldsymbol{y})}{\partial \boldsymbol{x}^{(j)}}$ and $\frac{\partial F_n(\boldsymbol{x},\boldsymbol{y})}{\partial \boldsymbol{y}^{(i)}}$. Substituting (12) and (15) into (9), we can get

$$\boldsymbol{x}^{(j)}(t_{n+1}) - \boldsymbol{x}^{(j)}(t_n) = -h\frac{N_n^{\boldsymbol{x}^{(j)}}}{D_n^{\boldsymbol{x}^{(j)}}} + \mathcal{O}(h^2) \tag{16}$$

Since $\dot{\boldsymbol{x}}^{(j)}(t_n^+)$, $\ddot{\boldsymbol{x}}^{(j)}(t_n^+)$ and $\dddot{\boldsymbol{x}}^{(j)}(t_n^+)$ are bounded independent of $h$ (by Lemma B.1), we can obtain the Taylor expansion at time $t_n$ (recall that $t_n = nh$)

$$\boldsymbol{x}^{(j)}(t_{n+1}) - \boldsymbol{x}^{(j)}(t_n) = \boldsymbol{x}^{(j)}(t_n + h) - \boldsymbol{x}^{(j)}(t_n)$$

$$= h\dot{\boldsymbol{x}}^{(j)}(t_n^+) + \frac{h^2}{2}\ddot{\boldsymbol{x}}^{(j)}(t_n^+) + \mathcal{O}(h^3), \tag{17}$$

where $\dot{\boldsymbol{x}}^{(j)}(t_n^+)$ (resp. $\ddot{\boldsymbol{x}}^{(j)}(t_n^+)$) is the right first-order derivative (resp. the right second-order derivative) at time $t_n$.

Applying the chain rule for (7), we have

$$\ddot{\boldsymbol{x}}^{(j)}(t_n^+)$$

$$= -\sum_{s=1}^{d_1} \frac{\partial^2 G_n(\boldsymbol{x}(t_n),\boldsymbol{y}(t_n))}{\partial \boldsymbol{x}^{(s)}\partial \boldsymbol{x}^{(j)}} \dot{\boldsymbol{x}}^{(s)}(t_n^+) - \sum_{s=1}^{d_2} \frac{\partial^2 G_n(\boldsymbol{x}(t_n),\boldsymbol{y}(t_n))}{\partial \boldsymbol{x}^{(j)}\partial \boldsymbol{y}^{(s)}} \dot{\boldsymbol{y}}^{(s)}(t_n^+) + \mathcal{O}(h)$$

$$= \sum_{s=1}^{d_1} \frac{\partial^2 G_n(\boldsymbol{x}(t_n),\boldsymbol{y}(t_n))}{\partial \boldsymbol{x}^{(s)}\partial \boldsymbol{x}^{(j)}} \frac{\partial G_n(\boldsymbol{x}(t_n),\boldsymbol{y}(t_n))}{\partial \boldsymbol{x}^{(s)}} \tag{18}$$

$$- \sum_{s=1}^{d_2} \frac{\partial^2 G_n(\boldsymbol{x}(t_n),\boldsymbol{y}(t_n))}{\partial \boldsymbol{x}^{(j)}\partial \boldsymbol{y}^{(s)}} \frac{\partial F_n(\boldsymbol{x}(t_n),\boldsymbol{y}(t_n))}{\partial \boldsymbol{y}^{(s)}} + \mathcal{O}(h).$$

Also note that from (7), we have

$$\dot{\boldsymbol{x}}^{(j)}(t_n^+) = -\frac{\partial G_n(\boldsymbol{x}(t_n),\boldsymbol{y}(t_n))}{\partial \boldsymbol{x}^{(j)}} + hA_n^{(j)}(\boldsymbol{x}(t_n),\boldsymbol{y}(t_n)), \tag{19}$$

Substituting (18) and (19) into (17), we can get

$$\boldsymbol{x}^{(j)}(t_{n+1}) - \boldsymbol{x}^{(j)}(t_n) = -h\frac{\partial G_n(\boldsymbol{x}(t_n),\boldsymbol{y}(t_n))}{\partial \boldsymbol{x}^{(j)}} + h^2 A_n^{(j)}(\boldsymbol{x}(t_n),\boldsymbol{y}(t_n))$$

$$+ \frac{h^2}{2}\sum_{s=1}^{d_1} \frac{\partial^2 G_n(\boldsymbol{x}(t_n),\boldsymbol{y}(t_n))}{\partial \boldsymbol{x}^{(s)}\partial \boldsymbol{x}^{(j)}} \frac{\partial G_n(\boldsymbol{x}(t_n),\boldsymbol{y}(t_n))}{\partial \boldsymbol{x}^{(s)}} \tag{20}$$

$$- \frac{h^2}{2}\sum_{s=1}^{d_2} \frac{\partial^2 G_n(\boldsymbol{x}(t_n),\boldsymbol{y}(t_n))}{\partial \boldsymbol{x}^{(j)}\partial \boldsymbol{y}^{(s)}} \frac{\partial F_n(\boldsymbol{x}(t_n),\boldsymbol{y}(t_n))}{\partial \boldsymbol{y}^{(s)}} + \mathcal{O}(h^3).$$

Comparing the coefficients of term $h$ in (16) and (20), we can get

$$\frac{\partial G_n(\boldsymbol{x}(t_n), \boldsymbol{y}(t_n))}{\partial \boldsymbol{x}^{(j)}} = \frac{N_n^{\boldsymbol{x}^{(j)}}}{D_n^{\boldsymbol{x}^{(j)}}}. \tag{21}$$

Repeating the similar argument for $y$-player's equation, we can get

$$\frac{\partial F_n(\boldsymbol{x}(t_n), \boldsymbol{y}(t_n))}{\partial \boldsymbol{y}^{(i)}} = \frac{N_n^{\boldsymbol{y}^{(i)}}}{D_n^{\boldsymbol{y}^{(i)}}}. \tag{22}$$

With (21) and (22) in hand, we can substitute (12) and (15) into (9) to get

$$\boldsymbol{x}^{(j)}(t_{n+1}) - \boldsymbol{x}^{(j)}(t_n) = -h\frac{N_n^{\boldsymbol{x}^{(j)}}}{D_n^{\boldsymbol{x}^{(j)}}} - h^2 \left( \frac{Q_n^{\boldsymbol{x}^{(j)}}}{D_n^{\boldsymbol{x}^{(j)}}} - \frac{P_n^{\boldsymbol{x}^{(j)}} N_n^{\boldsymbol{x}^{(j)}}}{\left(D_n^{\boldsymbol{x}^{(j)}}\right)^3} \right) + \mathcal{O}(h^3) \tag{23}$$

Comparing the coefficients of term $h^2$ in (23) and (20), we can get

$A_n^{(j)}(\boldsymbol{x}(t_n), \boldsymbol{y}(t_n))$

$$= \frac{1}{2} \sum_{s=1}^{d_2} \frac{\partial^2 G_n(\boldsymbol{x}(t_n), \boldsymbol{y}(t_n))}{\partial \boldsymbol{x}^{(j)} \partial \boldsymbol{y}^{(s)}} \frac{\partial F_n(\boldsymbol{x}(t_n), \boldsymbol{y}(t_n))}{\partial \boldsymbol{y}^{(s)}}$$

$$- \frac{1}{2} \sum_{s=1}^{d_1} \frac{\partial^2 G_n(\boldsymbol{x}(t_n), \boldsymbol{y}(t_n))}{\partial \boldsymbol{x}^{(s)} \partial \boldsymbol{x}^{(j)}} \frac{\partial G_n(\boldsymbol{x}(t_n), \boldsymbol{y}(t_n))}{\partial \boldsymbol{x}^{(s)}} - \left( \frac{Q_n^{\boldsymbol{x}^{(j)}}}{D_n^{\boldsymbol{x}^{(j)}}} - \frac{P_n^{\boldsymbol{x}^{(j)}} N_n^{\boldsymbol{x}^{(j)}}}{\left(D_n^{\boldsymbol{x}^{(j)}}\right)^3} \right)$$

$$= -\left( \sum_{s=1}^{d_1} \frac{\frac{\partial N_n^{\boldsymbol{x}^{(j)}}}{\partial \boldsymbol{x}^{(s)}} D_n^{\boldsymbol{x}^{(j)}} - \frac{\partial D_n^{\boldsymbol{x}^{(j)}}}{\partial \boldsymbol{x}^{(s)}} N_n^{\boldsymbol{x}^{(j)}}}{2 \left(D_n^{\boldsymbol{x}^{(j)}}\right)^2} \frac{N_n^{\boldsymbol{x}^{(s)}}}{D_n^{\boldsymbol{x}^{(s)}}} - \sum_{s=1}^{d_2} \frac{\frac{\partial N_n^{\boldsymbol{x}^{(j)}}}{\partial \boldsymbol{y}^{(s)}} D_n^{\boldsymbol{x}^{(j)}} - \frac{\partial D_n^{\boldsymbol{x}^{(j)}}}{\partial \boldsymbol{y}^{(s)}} N_n^{\boldsymbol{x}^{(j)}}}{2 \left(D_n^{\boldsymbol{x}^{(j)}}\right)^2} \frac{N_n^{\boldsymbol{y}^{(s)}}}{D_n^{\boldsymbol{y}^{(s)}}} \right)$$

$$- \left( \frac{Q_n^{\boldsymbol{x}^{(j)}}}{D_n^{\boldsymbol{x}^{(j)}}} - \frac{P_n^{\boldsymbol{x}^{(j)}} N_n^{\boldsymbol{x}^{(j)}}}{\left(D_n^{\boldsymbol{x}^{(j)}}\right)^3} \right). \tag{24}$$

Substituting $D_n^{\boldsymbol{x}^{(j)}}$, $N_n^{\boldsymbol{x}^{(j)}}$, $P_n^{\boldsymbol{x}^{(j)}}$ and $Q_n^{\boldsymbol{x}^{(j)}}$ in (6) into (24), we can get the desired equation.

Repeating the above similar argument for $y$-player's equation, we can get

$$\frac{\partial F_n(\boldsymbol{x}, \boldsymbol{y})}{\partial \boldsymbol{y}^{(i)}} = \frac{N_n^{\boldsymbol{y}^{(i)}}}{D_n^{\boldsymbol{y}^{(i)}}}, \tag{25}$$

$$B_n^{(i)}(\boldsymbol{x}(t_n), \boldsymbol{y}(t_n)) = \sum_{s=1}^{d_1} \frac{\frac{\partial N_n^{\boldsymbol{y}^{(i)}}}{\partial \boldsymbol{x}^{(s)}} D_n^{\boldsymbol{y}^{(i)}} - \frac{\partial D_n^{\boldsymbol{y}^{(i)}}}{\partial \boldsymbol{x}^{(s)}} N_n^{\boldsymbol{y}^{(i)}}}{2 \left(D_n^{\boldsymbol{y}^{(i)}}\right)^2} \frac{N_n^{\boldsymbol{x}^{(s)}}}{D_n^{\boldsymbol{x}^{(s)}}}$$

$$- \sum_{s=1}^{d_2} \frac{\frac{\partial N_n^{\boldsymbol{y}^{(i)}}}{\partial \boldsymbol{y}^{(s)}} D_n^{\boldsymbol{y}^{(i)}} - \frac{\partial D_n^{\boldsymbol{y}^{(i)}}}{\partial \boldsymbol{y}^{(s)}} N_n^{\boldsymbol{y}^{(i)}}}{2 \left(D_n^{\boldsymbol{y}^{(i)}}\right)^2} \frac{N_n^{\boldsymbol{y}^{(s)}}}{D_n^{\boldsymbol{y}^{(s)}}} + \left( \frac{Q_n^{\boldsymbol{y}^{(i)}}}{D_n^{\boldsymbol{y}^{(i)}}} - \frac{P_n^{\boldsymbol{y}^{(i)}} N_n^{\boldsymbol{y}^{(i)}}}{\left(D_n^{\boldsymbol{y}^{(i)}}\right)^3} \right). \tag{26}$$

Substituting $D_n^{\boldsymbol{y}^{(i)}}$, $N_n^{\boldsymbol{y}^{(i)}}$, $P_n^{\boldsymbol{y}^{(i)}}$ and $Q_n^{\boldsymbol{y}^{(i)}}$ in (6) into (25) and (26), we can get the desired equation. $\square$

**Corollary B.3.** *When $n$ tends to infinity, the ODEs become*

$$\frac{d\boldsymbol{x}^{(j)}(t)}{dt} = -\frac{\frac{\partial f(\boldsymbol{x},\boldsymbol{y})}{\partial \boldsymbol{x}^{(j)}}}{\sqrt{\left(\frac{\partial f(\boldsymbol{x},\boldsymbol{y})}{\partial \boldsymbol{x}^{(j)}}\right)^2 + \epsilon}} - \frac{h}{2\sqrt{\left(\frac{\partial f(\boldsymbol{x},\boldsymbol{y})}{\partial \boldsymbol{x}^{(j)}}\right)^2 + \epsilon}} \times$$

$$\left(\frac{1+\beta}{1-\beta} - \frac{1+\rho}{1-\rho} + \frac{\epsilon}{\left(\frac{\partial f(\boldsymbol{x},\boldsymbol{y})}{\partial \boldsymbol{x}^{(j)}}\right)^2 + \epsilon}\frac{1+\rho}{1-\rho}\right)\left(\frac{\partial}{\partial \boldsymbol{x}^{(j)}}\|\nabla_x f(\boldsymbol{x},\boldsymbol{y})\|_{1,\epsilon} - \frac{\partial}{\partial \boldsymbol{x}^{(j)}}\|\nabla_y f(\boldsymbol{x},\boldsymbol{y})\|_{1,\epsilon}\right),$$

$$\frac{d\boldsymbol{y}^{(i)}(t)}{dt} = \frac{\frac{\partial f(\boldsymbol{x},\boldsymbol{y})}{\partial \boldsymbol{y}^{(i)}}}{\sqrt{\left(\frac{\partial f(\boldsymbol{x},\boldsymbol{y})}{\partial \boldsymbol{y}^{(i)}}\right)^2 + \epsilon}} + \frac{h}{2\sqrt{\left(\frac{\partial f(\boldsymbol{x},\boldsymbol{y})}{\partial \boldsymbol{y}^{(i)}}\right)^2 + \epsilon}} \times$$

$$\left(\frac{1+\beta}{1-\beta} - \frac{1+\rho}{1-\rho} + \frac{\epsilon}{\left(\frac{\partial f(\boldsymbol{x},\boldsymbol{y})}{\partial \boldsymbol{y}^{(i)}}\right)^2 + \epsilon}\frac{1+\rho}{1-\rho}\right)\left(\frac{\partial}{\partial \boldsymbol{y}^{(i)}}\|\nabla_x f(\boldsymbol{x},\boldsymbol{y})\|_{1,\epsilon} - \frac{\partial}{\partial \boldsymbol{y}^{(i)}}\|\nabla_y f(\boldsymbol{x},\boldsymbol{y})\|_{1,\epsilon}\right).$$

**Remark.** Recall that the derivation of ODEs in Appendix B.2 is written by coordinate-wise and we transform it into matrix form in Section 3.

**Lemma B.4.** *Suppose that $f(\boldsymbol{x})$ has bounded derivatives up to the fourth order. Let $T_0$ be a fixed time interval. For all $t \in [0, T_0]$. Let $(\tilde{\boldsymbol{x}}(t), \tilde{\boldsymbol{y}}(t))$ be the solution trajectory of ODEs defined in Lemma B.2 and $(\boldsymbol{x}(t), \boldsymbol{y}(t))$ be the solution trajectory of ODEs for Continuous Adam-DA defined in Corollary B.3. Suppose that at time $t_n = nh$, we have*

$$(\tilde{\boldsymbol{x}}(t), \tilde{\boldsymbol{y}}(t)) = (\boldsymbol{x}(t), \boldsymbol{y}(t)).$$

*If we select $n > \max\{\frac{2\log h}{\log|\beta|}, \frac{2\log h}{\log \rho}\}$, then at time $t_{n+1} = (n+1)h$, we have*

$$\|(\tilde{\boldsymbol{x}}(t_{n+1}), \tilde{\boldsymbol{y}}(t_{n+1})) - (\boldsymbol{x}(t_{n+1}), \boldsymbol{y}(t_{n+1}))\| = \mathcal{O}(h^3).$$

*Proof.* We only need to prove

$$|\tilde{\boldsymbol{x}}^{(j)}(t_{n+1}) - \boldsymbol{x}^{(j)}(t_{n+1})| = \mathcal{O}(h^3), \ \ j = 1, 2, \cdots, d_1,$$
$$|\tilde{\boldsymbol{y}}^{(i)}(t_{n+1}) - \boldsymbol{y}^{(i)}(t_{n+1})| = \mathcal{O}(h^3), \ \ i = 1, 2, \cdots, d_2.$$

Firstly, we prove $|\tilde{\boldsymbol{x}}^{(j)}(t_{n+1}) - \boldsymbol{x}^{(j)}(t_{n+1})| = \mathcal{O}(h^3)$. By the Taylor expansion, we have

$$\tilde{\boldsymbol{x}}^{(j)}(t_{n+1}) = \tilde{\boldsymbol{x}}^{(j)}(nh + h) = \tilde{\boldsymbol{x}}^{(j)}(nh) + h\dot{\tilde{\boldsymbol{x}}}^{(j)}(nh^+) + \frac{h^2}{2}\ddot{\tilde{\boldsymbol{x}}}^{(j)}(nh^+) + \mathcal{O}(h^3),$$

$$\boldsymbol{x}^{(j)}(t_{n+1}) = \boldsymbol{x}^{(j)}(nh + h) = \boldsymbol{x}^{(j)}(nh) + h\dot{\boldsymbol{x}}^{(j)}(nh) + \frac{h^2}{2}\ddot{\boldsymbol{x}}^{(j)}(nh) + \mathcal{O}(h^3),$$

Recall that $(\boldsymbol{x}(t_n), \boldsymbol{y}(t_n)) = (\tilde{\boldsymbol{x}}(t_n), \tilde{\boldsymbol{y}}(t_n))$, then we can get

$$|\tilde{\boldsymbol{x}}^{(j)}(t_{n+1}) - \boldsymbol{x}^{(j)}(t_{n+1})| = h|\dot{\tilde{\boldsymbol{x}}}^{(j)}(nh^+) - \dot{\boldsymbol{x}}^{(j)}(nh)| + \frac{h^2}{2}|\ddot{\tilde{\boldsymbol{x}}}^{(j)}(nh^+) - \ddot{\boldsymbol{x}}^{(j)}(nh)| + \mathcal{O}(h^3).$$

By Lemma B.2 and Corollary B.3, we have

$$|h\dot{\tilde{\boldsymbol{x}}}^{(j)}(nh^+) - h\dot{\boldsymbol{x}}^{(j)}(nh)|$$

$$= h^2 \left| \frac{\epsilon}{\left(\frac{\partial f(\boldsymbol{x}(t), \boldsymbol{y}(t))}{\partial \boldsymbol{y}^{(i)}}\right)^2 + \epsilon} \left(-\frac{(n+1)\rho^{n+1}}{1 - \rho^{n+1}}\right) - \frac{(n+1)\beta^{n+1}}{1 - \beta^{n+1}} + \frac{(n+1)\rho^{n+1}}{1 - \rho^{n+1}} \right|$$

$$\times \left| \frac{\partial}{\partial \boldsymbol{x}^{(j)}} \|\nabla_x f(x(t), y(t))\|_{1,\epsilon} - \frac{\partial}{\partial \boldsymbol{x}^{(j)}} \|\nabla_y f(x(t), y(t))\|_{1,\epsilon} \right|$$

$$\leq h^2 \left( \left| \frac{\epsilon}{\left(\frac{\partial f(x(t), y(t))}{\partial \boldsymbol{y}^{(i)}}\right)^2 + \epsilon} \left(-\frac{(n+1)\rho^{n+1}}{1 - \rho^{n+1}}\right) \right| + \left| -\frac{(n+1)\beta^{n+1}}{1 - \beta^{n+1}} \right| + \left| \frac{(n+1)\rho^{n+1}}{1 - \rho^{n+1}} \right| \right)$$

$$\times \left| \frac{\partial}{\partial \boldsymbol{x}^{(j)}} \|\nabla_x f(x(t), y(t))\|_{1,\epsilon} - \frac{\partial}{\partial \boldsymbol{x}^{(j)}} \|\nabla_y f(x(t), y(t))\|_{1,\epsilon} \right|$$

$$\leq h^2 \left( \left| \frac{(n+1)\rho^{n+1}}{1 - \rho} \right| + \left| \frac{(n+1)\beta^{n+1}}{1 - \beta} \right| + \left| \frac{(n+1)\rho^{n+1}}{1 - \rho} \right| \right)$$

$$\times \left| \frac{\partial}{\partial \boldsymbol{x}^{(j)}} \|\nabla_x f(x(t), y(t))\|_{1,\epsilon} - \frac{\partial}{\partial \boldsymbol{x}^{(j)}} \|\nabla_y f(x(t), y(t))\|_{1,\epsilon} \right|$$

Actually, the term

$$\left| \frac{\partial}{\partial \boldsymbol{x}^{(j)}} \|\nabla_x f(\boldsymbol{x}(t), \boldsymbol{y}(t))\|_{1,\epsilon} - \frac{\partial}{\partial \boldsymbol{x}^{(j)}} \|\nabla_y f(\boldsymbol{x}(t), \boldsymbol{y}(t))\|_{1,\epsilon} \right|$$

is bounded since the first and second derivatives of $f$ are bounded. It suffices to prove

$$\left| \frac{(n+1)\rho^{n+1}}{1 - \rho} \right| + \left| \frac{(n+1)\beta^{n+1}}{1 - \beta} \right| + \left| \frac{(n+1)\rho^{n+1}}{1 - \rho} \right| = \mathcal{O}(h).$$

If we select $n > \max\{\frac{2\log h}{\log|\beta|}, \frac{2\log h}{\log \rho}\}$, i.e., $|\beta|^n < h^2$ and $|\rho|^n < h^2$, then we have

$$|(n+1)\beta^{n+1}| < \frac{T_0}{h}|\beta|^{n+1} < T_0|\beta|h, \quad |(n+1)\rho^{n+1}| < \frac{T_0}{h}|\rho|^{n+1} < T_0|\rho|h.$$

Repeating the similar argument, we can prove

$$h^2|\ddot{\tilde{\boldsymbol{x}}}^{(j)}(nh^+) - \ddot{\boldsymbol{x}}^{(j)}(nh)| = \mathcal{O}(h^3)$$

if $n > \max\{\frac{2\log h}{\log|\beta|}, \frac{2\log h}{\log \rho}\}$. We can also prove $|\tilde{\boldsymbol{y}}^{(i)}(t_{n+1}) - \boldsymbol{y}^{(i)}(t_{n+1})| = \mathcal{O}(h^3)$ through similarly arguments. $\qquad\square$

The proof of Theorem 3.1 follows from Lemma B.2, Corollary B.3 and Lemma B.4.

## C  ADDITIONAL MATERIALS FOR SECTION 4.

### C.1  PROOF OF PROPOSITION 4.1.

**Lemma C.1.** *We have*

- $\nabla_x \|\nabla_x f(\boldsymbol{x}, \boldsymbol{y})\|_{1,\epsilon} = \nabla_x^2 f(\boldsymbol{x}, \boldsymbol{y}) \cdot \mu_\epsilon(\boldsymbol{x}, \boldsymbol{y}) \cdot \nabla_x f(\boldsymbol{x}, \boldsymbol{y})$

- $\nabla_y \|\nabla_x f(\boldsymbol{x}, \boldsymbol{y})\|_{1,\epsilon} = \nabla_{yx} f(\boldsymbol{x}, \boldsymbol{y}) \cdot \mu_\epsilon(x, y) \cdot \nabla_x f(\boldsymbol{x}, \boldsymbol{y})$

- $\nabla_y \|\nabla_y f(\boldsymbol{x}, \boldsymbol{y})\|_{1,\epsilon} = \nabla_y^2 f(\boldsymbol{x}, \boldsymbol{y}) \cdot \nu_\epsilon(x, y) \cdot \nabla_y f(\boldsymbol{x}, \boldsymbol{y})$

- $\nabla_x \|\nabla_y f(\boldsymbol{x}, \boldsymbol{y})\|_{1,\epsilon} = \nabla_{xy} f(\boldsymbol{x}, \boldsymbol{y}) \cdot \nu_\epsilon(x, y) \cdot \nabla_y f(\boldsymbol{x}, \boldsymbol{y})$

*Proof.* Here we only calculate $\nabla_x \|\nabla_x f(\boldsymbol{x}, \boldsymbol{y})\|_{1,\epsilon}$, other items are similar.

Let $g(\boldsymbol{x}, \boldsymbol{y}) = \nabla_x f(\boldsymbol{x}, \boldsymbol{y})$, then we have

$$\nabla_x \|\nabla_x f(\boldsymbol{x}, \boldsymbol{y})\|_{1,\epsilon} = \nabla_x \|g(\boldsymbol{x}, \boldsymbol{y})\|_{1,\epsilon} = \nabla_x g(\boldsymbol{x}, \boldsymbol{y}) \cdot \nabla_g \|g(\boldsymbol{x}, \boldsymbol{y})\|_{1,\epsilon} = \nabla_x^2 f(\boldsymbol{x}, \boldsymbol{y}) \cdot \nabla_g \|g(\boldsymbol{x}, \boldsymbol{y})\|_{1,\epsilon}$$

For the term $\nabla_g \|g(\boldsymbol{x}, \boldsymbol{y})\|_{1,\epsilon}$, we have

$$\nabla_g \|g(\boldsymbol{x}, \boldsymbol{y})\|_{1,\epsilon} = \nabla_g \left( \sum_{i=1}^n \sqrt{g_i^2 + \epsilon} \right) = \left( \frac{g_i}{\sqrt{g_i^2 + \epsilon}} \right)_i = \mu_\epsilon(\boldsymbol{x}, \boldsymbol{y}) \cdot \nabla_x f(\boldsymbol{x}, \boldsymbol{y})$$

Thus combine above, we get

$$\nabla_x \|\nabla_x f(\boldsymbol{x}, \boldsymbol{y})\|_{1,\epsilon} = \nabla_x^2 f(\boldsymbol{x}, \boldsymbol{y}) \cdot \mu_\epsilon(\boldsymbol{x}, \boldsymbol{y}) \cdot \nabla_x f(\boldsymbol{x}, \boldsymbol{y})$$

$\square$

**Lemma C.2.** *For Continuous Adam-DA, let $(\boldsymbol{x}^*, \boldsymbol{y}^*)$ be a local Nash equilibrium. Denote the constant*

$$\gamma := \frac{h(1 + \beta)}{2\sqrt{\epsilon}(1 - \beta)},$$

*then we have*

$$\nabla_x \left( \frac{d\boldsymbol{x}}{dt} \right) \Big|_{(\boldsymbol{x}^*, \boldsymbol{y}^*)} = -\frac{1}{\sqrt{\epsilon}} \left( \nabla_x^2 f(\boldsymbol{x}^*, \boldsymbol{y}^*) + \gamma \left( \nabla_x^2 f \boldsymbol{x}^*, \boldsymbol{y}^*) \cdot \nabla_x^2 f(\boldsymbol{x}^*, \boldsymbol{y}^*) - \nabla_{xy} f(\boldsymbol{x}^*, \boldsymbol{y}^*) \cdot \nabla_{yx} f(\boldsymbol{x}^*, \boldsymbol{y}^*) \right) \right)$$

$$\nabla_y \left( \frac{d\boldsymbol{x}}{dt} \right) \Big|_{(\boldsymbol{x}^*, \boldsymbol{y}^*)} = -\frac{1}{\sqrt{\epsilon}} \left( \nabla_{xy} f(\boldsymbol{x}^*, \boldsymbol{y}^*) + \gamma \left( \nabla_x^2 f(\boldsymbol{x}^*, \boldsymbol{y}^*) \cdot \nabla_{xy} f(\boldsymbol{x}^*, \boldsymbol{y}^*) - \nabla_{xy} f(\boldsymbol{x}^*, \boldsymbol{y}^*) \cdot \nabla_y^2 f(\boldsymbol{x}^*, \boldsymbol{y}^*) \right) \right)$$

$$\nabla_x \left( \frac{d\boldsymbol{y}}{dt} \right) \Big|_{(\boldsymbol{x}^*, \boldsymbol{y}^*)} = \frac{1}{\sqrt{\epsilon}} \left( \nabla_{yx} f(\boldsymbol{x}^*, \boldsymbol{y}^*) + \gamma \left( \nabla_{yx} f(\boldsymbol{x}^*, \boldsymbol{y}^*) \cdot \nabla_x^2 f(\boldsymbol{x}^*, \boldsymbol{y}^*) - \nabla_y^2 f(\boldsymbol{x}^*, \boldsymbol{y}^*) \cdot \nabla_{yx} f(\boldsymbol{x}^*, \boldsymbol{y}^*) \right) \right)$$

$$\nabla_y \left( \frac{d\boldsymbol{y}}{dt} \right) \Big|_{(\boldsymbol{x}^*, \boldsymbol{y}^*)} = \frac{1}{\sqrt{\epsilon}} \left( \nabla_y^2 f(\boldsymbol{x}^*, \boldsymbol{y}^*) + \gamma \left( \nabla_{yx} f(\boldsymbol{x}^*, \boldsymbol{y}^*) \cdot \nabla_{xy} f(\boldsymbol{x}^*, \boldsymbol{y}^*) - \nabla_y^2 f(\boldsymbol{x}^*, \boldsymbol{y}^*) \cdot \nabla_y^2 f(\boldsymbol{x}^*, \boldsymbol{y}^*) \right) \right)$$

*Proof.* Here we only present the detailed proof for $\nabla_x \left( \frac{d\boldsymbol{x}}{dt} \right) \Big|_{(\boldsymbol{x}^*, \boldsymbol{y}^*)}$, other terms can be proved through a similar calculation.

Recall that in Continuous Adam-DA, we have

$$\dot{\boldsymbol{x}}(t) = -\mu_\epsilon(\boldsymbol{x}, \boldsymbol{y}) \left( \nabla_x f(\boldsymbol{x}, \boldsymbol{y}) + \frac{h}{2} \mathcal{M}_{\beta, \rho, \epsilon}^\mu(\boldsymbol{x}, \boldsymbol{y}) \cdot \nabla_x \left( \|\nabla_x f(\boldsymbol{x}, \boldsymbol{y})\|_{1,\epsilon} - \|\nabla_y f(\boldsymbol{x}, \boldsymbol{y})\|_{1,\epsilon} \right) \right),$$

thus by the product rule, we have

$$\nabla_x \left( \frac{d\boldsymbol{x}}{dt} \right) \Big|_{(\boldsymbol{x}^*, \boldsymbol{y}^*)} = -\nabla_x \mu_\epsilon(\boldsymbol{x}^*, \boldsymbol{y}^*) \cdot \left( \nabla_x f(\boldsymbol{x}^*, \boldsymbol{y}^*) + \frac{h}{2} \mathcal{M}_{\beta, \rho, \epsilon}^\mu(\boldsymbol{x}^*, \boldsymbol{y}^*) \cdot \nabla_x \left( \|\nabla_x f(\boldsymbol{x}^*, \boldsymbol{y}^*)\|_{1,\epsilon} - \|\nabla_y f(\boldsymbol{x}^*, \boldsymbol{y}^*)\|_{1,\epsilon} \right) \right)$$

$$- \mu_\epsilon(\boldsymbol{x}^*, \boldsymbol{y}^*) \cdot \nabla_x \left( \nabla_x f(\boldsymbol{x}^*, \boldsymbol{y}^*) + \frac{h}{2} \mathcal{M}_{\beta, \rho, \epsilon}^\mu(\boldsymbol{x}^*, \boldsymbol{y}^*) \cdot \nabla_x \left( \|\nabla_x f(\boldsymbol{x}^*, \boldsymbol{y}^*)\|_{1,\epsilon} - \|\nabla_y f(\boldsymbol{x}^*, \boldsymbol{y}^*)\|_{1,\epsilon} \right) \right)$$

$$\tag{27}$$

By the definition of local Nash equilibrium, we have

$$\nabla_x f(\boldsymbol{x}^*, \boldsymbol{y}^*) = \boldsymbol{0}, \quad \nabla_y f(\boldsymbol{x}^*, \boldsymbol{y}^*) = \boldsymbol{0}$$

and

$$\mu_\epsilon(\boldsymbol{x}^*, \boldsymbol{y}^*) := \begin{bmatrix} \frac{1}{\sqrt{\epsilon}} & & & \\ & \frac{1}{\sqrt{\epsilon}} & & \\ & & \ddots & \\ & & & \frac{1}{\sqrt{\epsilon}} \end{bmatrix}_{d_1 \times d_1}, \quad \mathcal{M}^\mu_{\beta,\rho,\epsilon}(\boldsymbol{x}^*, \boldsymbol{y}^*) = \mathcal{K}(\beta, \rho)\mathcal{I}_{d_1} + \frac{\epsilon(1+\rho)}{1-\rho}\mu_\epsilon^2(\boldsymbol{x}^*, \boldsymbol{y}^*).$$

Moreover, future simplification gives

$$\mathcal{M}^\mu_{\beta,\rho,\epsilon}(\boldsymbol{x}^*, \boldsymbol{y}^*) = \mathcal{K}(\beta, \rho)\mathcal{I}_{d_1} + \frac{\epsilon(1+\rho)}{1-\rho}\mu_\epsilon^2(\boldsymbol{x}^*, \boldsymbol{y}^*)$$

$$= \mathcal{K}(\beta, \rho)\mathcal{I}_{d_1} + \frac{1+\rho}{1-\rho}\mathcal{I}_{d_1}$$

$$= \frac{1+\beta}{1-\beta}\mathcal{I}_{d_1}, \quad \text{since } \mathcal{K}(\beta, \rho) = (1+\beta)/(1-\beta) + (1+\rho)/(1-\rho).$$

Thus the first term on the right hand side of equation 27 is $\boldsymbol{0}$, and we have

$$\nabla_x \left( \frac{d\boldsymbol{x}}{dt} \right)\big|_{(\boldsymbol{x}^*, \boldsymbol{y}^*)}$$

$$= -\mu_\epsilon(\boldsymbol{x}^*, \boldsymbol{y}^*) \cdot \nabla_x \left( \nabla_x f(\boldsymbol{x}^*, \boldsymbol{y}^*) + \frac{h}{2}\mathcal{M}^\mu_{\beta,\rho,\epsilon}(\boldsymbol{x}^*, \boldsymbol{y}^*) \cdot \nabla_x \left( \|\nabla_x f(\boldsymbol{x}^*, \boldsymbol{y}^*)\|_{1,\epsilon} - \|\nabla_y f(\boldsymbol{x}^*, \boldsymbol{y}^*)\|_{1,\epsilon} \right) \right)$$

$$= -\frac{1}{\sqrt{\epsilon}} \cdot \left( \nabla_x^2 f(\boldsymbol{x}^*, \boldsymbol{y}^*) + \frac{h(1+\beta)}{2(1-\beta)} \cdot \nabla_x^2 \left( \|\nabla_x f(\boldsymbol{x}^*, \boldsymbol{y}^*)\|_{1,\epsilon} - \|\nabla_y f(\boldsymbol{x}^*, \boldsymbol{y}^*)\|_{1,\epsilon} \right) \right) \quad (28)$$

Recall from Lemma C.1, we have

$$\nabla_x^2 \|\nabla_x f(\boldsymbol{x}, \boldsymbol{y})\|_{1,\epsilon}$$
$$= \nabla_x \left( \nabla_x^2 f(\boldsymbol{x}, \boldsymbol{y}) \cdot \mu_\epsilon(\boldsymbol{x}, \boldsymbol{y}) \right) \cdot \nabla_x f(\boldsymbol{x}, \boldsymbol{y}) + \nabla_x^2 f(\boldsymbol{x}, \boldsymbol{y}) \cdot \mu_\epsilon(\boldsymbol{x}, \boldsymbol{y}) \cdot \nabla_x^2 f(\boldsymbol{x}, \boldsymbol{y})$$

and

$$\nabla_x^2 \|\nabla_y f(\boldsymbol{x}, \boldsymbol{y})\|_{1,\epsilon}$$
$$= \nabla_x \left( \nabla_{xy} f(\boldsymbol{x}, \boldsymbol{y}) \cdot \nu_\epsilon(\boldsymbol{x}, \boldsymbol{y}) \right) \cdot \nabla_y f(\boldsymbol{x}, \boldsymbol{y}) + \nabla_{xy} f(\boldsymbol{x}, \boldsymbol{y}) \cdot \mu_\epsilon(\boldsymbol{x}, \boldsymbol{y}) \cdot \nabla_{yx} f(\boldsymbol{x}, \boldsymbol{y})$$

Take $(\boldsymbol{x}^*, \boldsymbol{y}^*)$ into above two equalities, and use the fact that $\nabla_y f(\boldsymbol{x}^*, \boldsymbol{y}^*), \nabla_y f(\boldsymbol{x}^*, \boldsymbol{y}^*) = \boldsymbol{0}$, we get

$$\nabla_x^2 \|\nabla_x f(\boldsymbol{x}, \boldsymbol{y})\|_{1,\epsilon} = \nabla_x^2 f(\boldsymbol{x}^*, \boldsymbol{y}^*) \cdot \mu_\epsilon(\boldsymbol{x}^*, \boldsymbol{y}^*) \cdot \nabla_x^2 f(\boldsymbol{x}^*, \boldsymbol{y}^*)$$

$$= \frac{1}{\sqrt{\epsilon}} \nabla_x^2 f(\boldsymbol{x}^*, \boldsymbol{y}^*) \cdot \nabla_x^2 f(\boldsymbol{x}^*, \boldsymbol{y}^*)$$

and

$$\nabla_x^2 \|\nabla_y f(\boldsymbol{x}, \boldsymbol{y})\|_{1,\epsilon} = \nabla_{xy} f(\boldsymbol{x}^*, \boldsymbol{y}^*) \cdot \mu_\epsilon(\boldsymbol{x}^*, \boldsymbol{y}^*) \cdot \nabla_{yx} f(\boldsymbol{x}^*, \boldsymbol{y}^*)$$

$$= \frac{1}{\sqrt{\epsilon}} \nabla_{xy} f(\boldsymbol{x}^*, \boldsymbol{y}^*) \cdot \nabla_{yx} f(\boldsymbol{x}^*, \boldsymbol{y}^*)$$

Take above two equalities into equation 28, we get

$$\nabla_x \left( \frac{d\boldsymbol{x}}{dt} \right)\big|_{(\boldsymbol{x}^*, \boldsymbol{y}^*)} = -\frac{1}{\sqrt{\epsilon}} \left( \nabla_x^2 f(\boldsymbol{x}^*, \boldsymbol{y}^*) + \gamma \left( \nabla_x^2 f(\boldsymbol{x}^*, \boldsymbol{y}^*) \cdot \nabla_x^2 f(\boldsymbol{x}^*, \boldsymbol{y}^*) - \nabla_{xy} f(\boldsymbol{x}^*, \boldsymbol{y}^*) \cdot \nabla_{yx} f(\boldsymbol{x}^*, \boldsymbol{y}^*) \right) \right),$$

this completes the proof for the terms $\nabla_x \left( \frac{d\boldsymbol{x}}{dt} \right)\big|_{(\boldsymbol{x}^*, \boldsymbol{y}^*)}$. $\qquad\square$

Now we are ready to proof Proposition 4.1.

*Proof of Proposition 4.1 .* By definition, the Jacobian $\mathcal{J}_{\text{Adam}}$ of Continuous Adam-DA at $(\boldsymbol{x}^*, \boldsymbol{y}^*)$ is written as

$$\mathcal{J}_{\text{Adam}} = \begin{bmatrix} \nabla_x \left( \frac{d\boldsymbol{x}}{dt} \right) |_{(\boldsymbol{x}^*, \boldsymbol{y}^*)} & \nabla_y \left( \frac{d\boldsymbol{x}}{dt} \right) |_{(\boldsymbol{x}^*, \boldsymbol{y}^*)} \\ \nabla_x \left( \frac{d\boldsymbol{y}}{dt} \right) |_{(\boldsymbol{x}^*, \boldsymbol{y}^*)} & \nabla_y \left( \frac{d\boldsymbol{y}}{dt} \right) |_{(\boldsymbol{x}^*, \boldsymbol{y}^*)} \end{bmatrix}$$

Take the terms of $\nabla_x \left( \frac{d\boldsymbol{x}}{dt} \right) |_{(\boldsymbol{x}^*, \boldsymbol{y}^*)}, \nabla_y \left( \frac{d\boldsymbol{x}}{dt} \right) |_{(\boldsymbol{x}^*, \boldsymbol{y}^*)}, \nabla_x \left( \frac{d\boldsymbol{y}}{dt} \right) |_{(\boldsymbol{x}^*, \boldsymbol{y}^*)}$ and $\nabla_y \left( \frac{d\boldsymbol{y}}{dt} \right) |_{(\boldsymbol{x}^*, \boldsymbol{y}^*)}$ in Lemma C.2 into above, we get

$$\mathcal{J}_{\text{Adam}} = \frac{1}{\sqrt{\epsilon}} \left( \mathcal{I}_{d_1} - \frac{h(1+\beta)}{2\sqrt{\epsilon}(1-\beta)} \mathcal{J} \right) \mathcal{J}, \ \mathcal{J} = \begin{bmatrix} -\nabla_x^2 f(\boldsymbol{x}^*, \boldsymbol{y}^*) & -\nabla_{xy} f(\boldsymbol{x}^*, \boldsymbol{y}^*) \\ \nabla_{yx} f(\boldsymbol{x}^*, \boldsymbol{y}^*) & \nabla_y^2 f(\boldsymbol{x}^*, \boldsymbol{y}^*) \end{bmatrix}.$$

$\square$

### C.2 PROOF OF THEOREM 4.3.

We first introduce the following lemma, which describe the eigenvalues of matrix polynomial

**Lemma C.3.** *If $p(x)$ is a polynomial and $A \in \mathbb{R}^{n \times n}$, then every eigenvalue of $p(A)$ can be represented by $p(\lambda)$, where $\lambda \in \text{Sp}(\mathcal{J})$.*

*Proof.* The following proof comes from Graham (2018). A key point here is that we are considering eigenvalues of matrices over complex number field $\mathbb{C}$, and every $n$-degree polynomial over $\mathbb{C}$ has $n$ roots. Let $\mu \in \mathbb{C}$ be an eigenvalue of $p(A)$, and we consider the polynomial $p(x) - \mu$. Over the complex number field $\mathbb{C}$, the polynomial $p(x) - \mu$ can be factored as

$$p(x) - \mu = a \prod_{i=1}^{n} (x - \lambda_i) \tag{29}$$

Moreover, since $\mu$ is an eigenvalue of $p(A)$, we have $\det(p(A) - \mu\mathcal{I}) = 0$. Thus, from (29), as least one of $A - \lambda_i \mathcal{I}$ is not invertible, i.e., $\exists v \in \mathbb{C}^n$, such that $(A - \lambda_i \mathcal{I}) v = 0$. This implies $\mu = p(\lambda_i)$ and finishes the proof. $\square$

In the following, we state a corollary of Assumption 4.2, which is proved in (Wang & Chizat, 2024).

**Lemma C.4.** *[Theorem 2.1 in Wang & Chizat (2024)] Under Assumption 4.2, we have $\Re(\lambda) < 0$ for every $\lambda \in \text{Sp}(\mathcal{J})$.*

*Proof of Theorem 4.3.* According to Proposition 4.1 and Lemma C.3, every eigenvalue of $\mathcal{J}_{\text{Adam}}$ can be represented by a quadratic polynomial

$$\frac{1}{\sqrt{\epsilon}} \left( 1 - \frac{h(1+\beta)}{2\sqrt{\epsilon}(1-\beta)} \lambda \right) \lambda, \tag{30}$$

where $\lambda \in \text{Sp}(\mathcal{J})$ is an eigenvalue of Jacobian, and any number represented by equation 30 is an eigenvalue of $\mathcal{J}_{\text{Adam}}$.

Thus, to ensure the local convergence of Continuous Adam-DA, we need

$$\Re \left( \left[ 1 - \frac{h(1+\beta)}{2\sqrt{\epsilon}(1-\beta)} \lambda \right] \cdot \lambda \right)$$

$$= \Re(\lambda) - \frac{h(1+\beta)}{2\sqrt{\epsilon}(1-\beta)} \left( \Re(\lambda)^2 - \Im(\lambda)^2 \right) < 0, \ \forall \lambda \in \text{Sp}(\mathcal{J}). \tag{31}$$

From Lemma C.4, $\Re(\lambda)$ is a negative number, and from Assumption 4.2, the term $\Re(\lambda)^2 - \Im(\lambda)^2$ in equation 31 is a negative number. Thus, for some fixed parameter $\beta, \epsilon$ and $\forall \lambda \in \mathrm{Sp}(\mathcal{J})$, we need the step size $h$ in equation 31 satisfies

$$h < \min_{\lambda \in \mathrm{Sp}(\mathcal{J})} \frac{2\sqrt{\epsilon}(1-\beta)}{(1+\beta)} \frac{|\Re(\lambda)|}{(\Im(\lambda)^2 - \Re(\lambda)^2)}, \tag{32}$$

and this finishes the proof of Theorem 4.3. $\qquad\square$

## C.3  PROOF OF THEOREM 4.4.

**Additional Notations.**  Denote the Jacobian matrix of any matrix $A$ by $\mathrm{Jac}(A)$. Denote the spectral radius of any matrix $A$ by $\varrho(A)$.

Define $\boldsymbol{z}_n = (\tilde{\boldsymbol{m}}_n, \tilde{\boldsymbol{v}}_n, \boldsymbol{x}_n, \hat{\boldsymbol{m}}_n, \hat{\boldsymbol{v}}_n, \boldsymbol{y}_n)^\top$. We can rewrite Adam-DA as a time-dependent discrete-time dynamical system $\boldsymbol{z}_{n+1} = T(n, \boldsymbol{z}_n)$, which can be split into an autonomous dynamical system $\bar{T}(\boldsymbol{z}_n)$ and a non-autonomous one $\mathcal{R}(n, \boldsymbol{z}_n)$ in the following form

$$\boldsymbol{z}_{n+1} = T(n, \boldsymbol{z}_n) = \bar{T}(\boldsymbol{z}_n) + \mathcal{R}(n, \boldsymbol{z}_n), \qquad \text{(Non-Autonomous System)}$$

where

$$T(n, \boldsymbol{z}_n) = \begin{bmatrix} \beta \tilde{\boldsymbol{m}}_n + (1-\beta)\nabla_x f(\boldsymbol{x}_n, \boldsymbol{y}_n) \\[2mm] \rho \tilde{\boldsymbol{v}}_n + (1-\rho)\left(\nabla_x f(\boldsymbol{x}_n, \boldsymbol{y}_n)\right)^2 \\[2mm] \boldsymbol{x}_n - h \frac{\tilde{\boldsymbol{m}}_{n+1}/(1-\beta^{n+1})}{\sqrt{\tilde{\boldsymbol{v}}_{n+1}/(1-\rho^{n+1})} + \epsilon} \\[2mm] \beta \hat{\boldsymbol{m}}_n + (1-\beta)\nabla_y f(\boldsymbol{x}_n, \boldsymbol{y}_n) \\[2mm] \rho \hat{\boldsymbol{v}}_n + (1-\rho)\left(\nabla_y f(\boldsymbol{x}_n, \boldsymbol{y}_n)\right)^2 \\[2mm] \boldsymbol{y}_n + h \frac{\hat{\boldsymbol{m}}_{n+1}/(1-\beta^{n+1})}{\sqrt{\hat{\boldsymbol{v}}_{n+1}/(1-\rho^{n+1})} + \epsilon} \end{bmatrix}, \ \bar{T}(\boldsymbol{z}_n) = \begin{bmatrix} \beta \tilde{\boldsymbol{m}}_n + (1-\beta)\nabla_x f(\boldsymbol{x}_n, \boldsymbol{y}_n) \\[2mm] \rho \tilde{\boldsymbol{v}}_n + (1-\rho)\left(\nabla_x f(\boldsymbol{x}_n, \boldsymbol{y}_n)\right)^2 \\[2mm] \boldsymbol{x}_n - h \frac{\tilde{\boldsymbol{m}}_{n+1}}{\sqrt{\tilde{\boldsymbol{v}}_{n+1}} + \epsilon} \\[2mm] \beta \hat{\boldsymbol{m}}_n + (1-\beta)\nabla_y f(\boldsymbol{x}_n, \boldsymbol{y}_n) \\[2mm] \rho \hat{\boldsymbol{v}}_n + (1-\rho)\left(\nabla_y f(\boldsymbol{x}_n, \boldsymbol{y}_n)\right)^2 \\[2mm] \boldsymbol{y}_n + h \frac{\hat{\boldsymbol{m}}_{n+1}}{\sqrt{\hat{\boldsymbol{v}}_{n+1}} + \epsilon} \end{bmatrix},$$

and

$$\mathcal{R}(n, \boldsymbol{z}_n) = \begin{bmatrix} \boldsymbol{0} \\[2mm] \boldsymbol{0} \\[2mm] \frac{h\tilde{\boldsymbol{m}}_{n+1}}{\sqrt{\tilde{\boldsymbol{v}}_{n+1}} + \epsilon} - \frac{h\tilde{\boldsymbol{m}}_{n+1}/(1-\beta^{n+1})}{\sqrt{\tilde{\boldsymbol{v}}_{n+1}/(1-\rho^{n+1})} + \epsilon} \\[2mm] \boldsymbol{0} \\[2mm] \boldsymbol{0} \\[2mm] \frac{-h\hat{\boldsymbol{m}}_{n+1}}{\sqrt{\hat{\boldsymbol{v}}_{n+1}} + \epsilon} + \frac{h\hat{\boldsymbol{m}}_{n+1}/(1-\beta^{n+1})}{\sqrt{\hat{\boldsymbol{v}}_{n+1}/(1-\rho^{n+1})} + \epsilon} \end{bmatrix} = \begin{bmatrix} \boldsymbol{0} \\[2mm] \boldsymbol{0} \\[2mm] \frac{h\tilde{\boldsymbol{m}}_{n+1}}{\sqrt{\tilde{\boldsymbol{v}}_{n+1}} + \epsilon} - \frac{h\sqrt{1-\rho^{n+1}}}{1-\beta^{n+1}} \frac{\tilde{\boldsymbol{m}}_{n+1}}{\sqrt{\tilde{\boldsymbol{v}}_{n+1}} + (1-\rho^{n+1})\epsilon} \\[2mm] \boldsymbol{0} \\[2mm] \boldsymbol{0} \\[2mm] \frac{-h\hat{\boldsymbol{m}}_{n+1}}{\sqrt{\hat{\boldsymbol{v}}_{n+1}} + \epsilon} + \frac{h\sqrt{1-\rho^{n+1}}}{1-\beta^{n+1}} \frac{\hat{\boldsymbol{m}}_{n+1}}{\sqrt{\hat{\boldsymbol{v}}_{n+1}} + (1-\rho^{n+1})\epsilon} \end{bmatrix}.$$

With this split in hand, the proof of local convergence of Adam-DA is transformed into the proof of local convergence of $\bar{T}(\boldsymbol{z}_n)$ and $\mathcal{R}(n, \boldsymbol{z}_n)$. The sketched proof of Theorem 4.4 follows by two steps:

- Step 1: We start by proving that finding the Local Nash Equilibrium $(\boldsymbol{x}^*, \boldsymbol{y}^*)$ is equivalent to finding the fixed point $(\boldsymbol{0}, \boldsymbol{0}, \boldsymbol{x}^*, \boldsymbol{0}, \boldsymbol{0}, \boldsymbol{y}^*)$ of $\bar{T}$ in Lemma C.5. We next compute the characteristic polynomial of the Jacobian matrix of $\bar{T}$ at the fixed point. Then by Lemma C.7, we can select the parameters $h$, $\beta$ and $\epsilon$ to ensure that the spectral radius of the Jacobian matrix of $\bar{T}$ is less than 1. Therefore, we can conclude $\bar{T}$ converges locally near the local Nash equilibrium by Lemma C.6.

- Step 2: We prove that $\|\mathcal{R}(n, \boldsymbol{z}_n)\|$, the perturbation term of Non-Autonomous System, converges locally with an exponential rate by direct algebra computation, i.e., the perturbation term of Non-Autonomous System vanish sufficiently fast with an exponential rate. Intuitively, the exponentially vanishing perturbation hardly influence the local convergence of Non-Autonomous System.

- Step 3: We prove that (Non-Autonomous System) consists of an autonomous system $\bar{T}$ with $\varrho\left(\text{Jac}(\bar{T})\right) < 1$ and an exponentially vanishing perturbation will locally converge with an exponential rate in Lemma C.9.

**Lemma C.5.** $(\boldsymbol{x}^*, \boldsymbol{y}^*)$ *is a local Nash equilibrium if and only if* $\boldsymbol{z}^* = (\boldsymbol{0}, \boldsymbol{0}, \boldsymbol{x}^*, \boldsymbol{0}, \boldsymbol{0}, \boldsymbol{y}^*)^\top$ *is the fixed point of* $T$ *and* $\bar{T}$, *and* $\nabla_x^2 f(\boldsymbol{x}^*, \boldsymbol{y}^*) \succeq \boldsymbol{0}$, $\nabla_y^2 f(\boldsymbol{x}^*, \boldsymbol{y}^*) \preceq \boldsymbol{0}$.

*Proof.* On the one hand, if $\boldsymbol{z}^* = (\boldsymbol{0}, \boldsymbol{0}, \boldsymbol{x}^*, \boldsymbol{0}, \boldsymbol{0}, \boldsymbol{y}^*)$ is the fixed point of $\bar{T}$, consider the following equation

$$
\bar{T}\left((\boldsymbol{0}, \boldsymbol{0}, \boldsymbol{x}^*, \boldsymbol{0}, \boldsymbol{0}, \boldsymbol{y}^*)^\top\right) =
\begin{bmatrix}
(1-\beta)\nabla_x f(\boldsymbol{x}^*, \boldsymbol{y}^*) \\
(1-\rho)\left(\nabla_x f(\boldsymbol{x}^*, \boldsymbol{y}^*)\right)^2 \\
\boldsymbol{x}^* \\
(1-\beta)\nabla_y f(\boldsymbol{x}^*, \boldsymbol{y}^*) \\
(1-\rho)\left(\nabla_y f(\boldsymbol{x}^*, \boldsymbol{y}^*)\right)^2 \\
\boldsymbol{y}^*
\end{bmatrix}
=
\begin{bmatrix}
\boldsymbol{0} \\
\boldsymbol{0} \\
\boldsymbol{x}^* \\
\boldsymbol{0} \\
\boldsymbol{0} \\
\boldsymbol{y}^*
\end{bmatrix},
$$

we can get $\nabla_x f(\boldsymbol{x}^*, \boldsymbol{y}^*) = \boldsymbol{0}$ and $\nabla_y f(\boldsymbol{x}^*, \boldsymbol{y}^*) = \boldsymbol{0}$, i.e., $(\boldsymbol{x}^*, \boldsymbol{y}^*)$ is a local Nash equilibrium.

On the other hand, if $\nabla_x f(\boldsymbol{x}^*, \boldsymbol{y}^*) = \boldsymbol{0}$ and $\nabla_y f(\boldsymbol{x}^*, \boldsymbol{y}^*) = \boldsymbol{0}$, we have

$$
\bar{T}\left((\tilde{\boldsymbol{m}}^*, \tilde{\boldsymbol{v}}^*, \boldsymbol{x}^*, \hat{\boldsymbol{m}}^*, \hat{\boldsymbol{v}}^*, \boldsymbol{y}^*)^\top\right) =
\begin{bmatrix}
\beta\tilde{\boldsymbol{m}}^* \\
\rho\tilde{\boldsymbol{v}}^* \\
\boldsymbol{x}^* - h\frac{\tilde{\boldsymbol{m}}^*}{\sqrt{\tilde{\boldsymbol{v}}^* + \epsilon}} \\
\beta\hat{\boldsymbol{m}}^* \\
\rho\hat{\boldsymbol{v}}^* \\
\boldsymbol{y}^* + h\frac{\hat{\boldsymbol{m}}^*}{\sqrt{\hat{\boldsymbol{v}}^* + \epsilon}}
\end{bmatrix}
=
\begin{bmatrix}
\tilde{\boldsymbol{m}}^* \\
\tilde{\boldsymbol{v}}^* \\
\boldsymbol{x}^* \\
\hat{\boldsymbol{m}}^* \\
\hat{\boldsymbol{v}}^* \\
\boldsymbol{y}^*
\end{bmatrix},
$$

solve this fixed point equation, we can get $\tilde{\boldsymbol{m}}^* = \boldsymbol{0}$, $\tilde{\boldsymbol{v}}^* = \boldsymbol{0}$, $\hat{\boldsymbol{m}}^* = \boldsymbol{0}$ and $\hat{\boldsymbol{v}}^* = \boldsymbol{0}$, which means $(\boldsymbol{0}, \boldsymbol{0}, \boldsymbol{x}^*, \boldsymbol{0}, \boldsymbol{0}, \boldsymbol{y}^*)$ is the fixed point of $\bar{T}$. We can implement the similar argument for $T$. □

**Lemma C.6.** *[Corollary 4.35 in Elaydi (2005) and Theorem II.1 in Bock & Weiß (2021)] Consider* $\bar{T} : M \to M$ *with a fixed point* $w^*$ *and* $\bar{T}$ *continuously differentiable in an open disk* $B_\delta(w^*) \subset M$ *with radius* $\delta$. *Assume*

$$\varrho(Jac(\bar{T}_{w^*})) < 1,$$

*then there exists* $0 < \delta_0 < \delta$ *and* $0 \le c < 1$ *such that for all* $w_0$ *with* $\|w_0 - w^*\| < \delta_0$ *and for all* $t \in \mathbb{N}$,

$$\|w(t; w_0) - w^*\| \le c^t \|w_0 - w^*\|.$$

**Lemma C.7.** *[Theorem 6.8(b) in Henrici (1974)] For a 2rd-order polynomial* $p(\lambda) = \lambda^2 + a\lambda + b$, *where* $a, b \in \mathbb{C}$, *its roots all lie within the open unit disk of the complex plane if and only if*

$$|b| < 1$$

*and*

$$|a - b\bar{a}| < 1 - |b|^2,$$

*where* $\bar{a}$ *is the complex conjugate of* $a$.

Lemma C.8, which describes how to construct an equivalent norm of matrix $J$ satisfying the contraction property when $\varrho(J) < 1$, is the key to proving Lemma C.9.

**Lemma C.8** (Equivalent norm construction). *Suppose that the matrix $J$ satisfies $\varrho(J) < 1$. Then there exists a positive definite matrix $P$ defined by*

$$P = \sum_{k=0}^{\infty} (J^\top)^k J^k,$$

*such that the induced norm*

$$\|\boldsymbol{x}\|_P = \sqrt{\boldsymbol{x}^\top P \boldsymbol{x}}$$

*satisfies*

$$\|J\boldsymbol{x}\|_P \leq \gamma \|\boldsymbol{x}\|_P, \quad \forall \boldsymbol{x} \in \mathbb{R}^d$$

*for some $\gamma \in (0, 1)$. In other words, $J$ is a contraction in the $\|\cdot\|_P$-norm.*

*Proof.* The following proof comes from standard techniques in matrix calculation, e.g., Elaydi (2005). We include it here for the completeness of the proof.

The key to proof is verifying $P = \sum_{k=0}^{\infty} (J^\top)^k J^k$ is well defined, i.e., $P = \sum_{k=0}^{\infty} (J^\top)^k J^k$, converges in component-wise. The proof of $P = \sum_{k=0}^{\infty} (J^\top)^k J^k$ converging in component-wise can be decomposed into the following 5 steps:

**Step 1. Make a Jordan decomposition for $J$.** Over complex field $\mathbb{C}$, there exists an invertible matrix $V$ such that

$$J = V \, \text{diag}(J_1, \ldots, J_t) \, V^{-1},$$

where each $J_i$ with size $r_i$ is a Jordan block of the form

$$J_i = \lambda_i \mathcal{I}_{r_i} + N_i,$$

with $N_i$ strictly upper triangular and $N_i^{r_i} = \boldsymbol{0}$. Let

$$m := \max_i r_i$$

be the size of the largest Jordan block. Then

$$\|J^k\| \leq \|V\| \, \|V^{-1}\| \cdot \max_i \|J_i^k\|.$$

**Step 2. Make binomial expansion of each single Jordan block.** Fix one block $J_i = \lambda_i \mathcal{I}_{r_i} + N_i$. By the binomial expansion,

$$J_i^k = \sum_{s=0}^{r_i-1} \binom{k}{s} \lambda_i^{k-s} N_i^s, \qquad k \geq 0,$$

since $N_i^{r_i} = 0$. Hence

$$\|J_i^k\| \leq \sum_{s=0}^{r_i-1} \binom{k}{s} |\lambda_i|^{k-s} \|N_i\|^s.$$

**Step 3. Estimate the combinatorial numbers.** First, $\binom{k}{s} \leq k^s/s!$ for $0 \leq s \leq k$. Second, choose $\mu$ with $\rho(J) < \mu < 1$. Then $|\lambda_i| \leq \rho(J) < \mu$, so

$$|\lambda_i|^{k-s} \leq \mu^{k-s} = \mu^k \mu^{-s}.$$

Therefore,

$$\|J_i^k\| \leq \mu^k \sum_{s=0}^{r_i-1} \frac{k^s}{s!} (\mu^{-1}\|N_i\|)^s.$$

**Step 4. Extract the polynomial factors.** Since $s \leq r_i - 1$, for $k \geq 1$ we have $k^s \leq k^{r_i-1}$. Hence

$$\|J_i^k\| \leq \mu^k k^{r_i-1} \sum_{s=0}^{r_i-1} \frac{(\mu^{-1}\|N_i\|)^s}{s!}.$$

Define the constant

$$C_i := \sum_{s=0}^{r_i-1} \frac{(\mu^{-1}\|N_i\|)^s}{s!}.$$

Thus,

$$\|J_i^k\| \leq C_i \, k^{r_i-1} \, \mu^k.$$

**Step 5. Combine all blocks.** Taking the maximum over all Jordan blocks yields (Recall that $m = \max_i r_i$.)

$$\|J^k\| \leq \|V\| \, \|V^{-1}\| \cdot \max_i \|J_i^k\| \leq \left( \|V\| \, \|V^{-1}\| \, \max_i C_i \right) k^{m-1} \mu^k = C_{\max} \, k^{m-1} \, \mu^k,$$

where we define

$$C_{\max} = \|V\| \, \|V^{-1}\| \, \max_i C_i.$$

Therefore, we have

$$\sum_{k=0}^{\infty} \|(J^\top)^k J^k\| \leq \sum_{k=0}^{\infty} \|J^k\|^2 \leq \sum_{k=0}^{\infty} C_{\max}^2 k^{2(m-1)} \mu^{2k} < \infty,$$

i.e., $P = \sum_{k=0}^{\infty} (J^\top)^k J^k$ converges absolutely, thus $P$ converges in component-wise, which means $P$ is well defined. We have proved that $P = \sum_{k=0}^{\infty} (J^\top)^k J^k$ is well defined. Next, we procced to complete the remaining proof of Lemma C.8.

Note that for all $\boldsymbol{x} \in \mathbb{R}^d$,

$$\boldsymbol{x}^\top P \boldsymbol{x} = \sum_{k=0}^{\infty} \boldsymbol{x}^\top (J^\top)^k J^k \boldsymbol{x} = \sum_{k=0}^{\infty} \|J^k x\|^2,$$

we can easily get $P$ is positive definite.

Besides, we can verify

$$J^\top P J = J^\top \left( \sum_{k=0}^{\infty} (J^\top)^k J^k \right) J = \sum_{k=0}^{\infty} (J^\top)^{k+1} J^{k+1} = \sum_{k=1}^{\infty} (J^\top)^k J^k.$$

Then we have

$$P - J^\top P J = \sum_{k=0}^{\infty} (J^\top)^k J^k - \sum_{k=1}^{\infty} (J^\top)^k J^k = (J^\top)^0 J^0 = \mathcal{I}. \tag{33}$$

From (33), for all $\mathbb{R}^d$, we have

$$\boldsymbol{x}^\top J^\top P J \boldsymbol{x} = \boldsymbol{x}^\top (P - \mathcal{I}) \boldsymbol{x} = \boldsymbol{x}^\top P \boldsymbol{x} - \|\boldsymbol{x}\|^2.$$

Suppose $\lambda_{\min}$ is the smallest eigenvalue of $P^{-1}$, then we have

$$\begin{aligned}
\boldsymbol{x}^\top J^\top P J \boldsymbol{x} &= \boldsymbol{x}^\top P \boldsymbol{x} - \|\boldsymbol{x}\|^2 \\
&= \boldsymbol{x}^\top P \boldsymbol{x} - \boldsymbol{x}^\top \mathcal{I} \boldsymbol{x} \\
&= \boldsymbol{x}^\top P \boldsymbol{x} - \boldsymbol{x}^\top P^{\frac{1}{2}} P^{-1} P^{\frac{1}{2}} \boldsymbol{x} \\
&\leq \boldsymbol{x}^\top P \boldsymbol{x} - \lambda_{\min} \boldsymbol{x}^\top P \boldsymbol{x} \\
&= (1 - \lambda_{\min}) \boldsymbol{x}^\top P \boldsymbol{x}.
\end{aligned}$$

Therefore

$$J^\top P J \preceq (1 - \lambda_{\min}) P.$$

Since $P$ is positive definite, then $J^\top P J$ is also positive definite, thus $1 - \lambda_{\min} > 0$. Also, $P^{-1}$ is positive definite implies $\lambda_{\min} > 0$, thus $0 < 1 - \lambda_{\min} < 1$.

Then the operator norm induced by the matrix $P$, $\|\cdot\|_P$ will satisfy (Denote $\gamma^2 = 1 - \lambda_{\min} \in (0,1)$)

$$\|J\|_P^2 := \sup_{\boldsymbol{x} \neq 0} \frac{\|J\boldsymbol{x}\|_P}{\|\boldsymbol{x}\|_P} = \sup_{\boldsymbol{x} \neq 0} \frac{\boldsymbol{x}^\top J^\top P J \boldsymbol{x}}{\boldsymbol{x}^\top P \boldsymbol{x}} \leq 1 - \lambda_{\min} = \gamma^2,$$

i.e.,

$$\|J\boldsymbol{x}\|_P \leq \gamma \|\boldsymbol{x}\|_P, \quad \forall \boldsymbol{x} \in \mathbb{R}^d.$$

$\square$

**Lemma C.9.** *Consider ([Non-Autonomous System](#)). Let $\boldsymbol{z}^*$ be a fixed point of $\bar{T}$, and suppose:*

> *(i) The Jacobian $Jac(\bar{T}(\boldsymbol{z}^*))$ satisfies $\varrho\left(Jac(\bar{T}(\boldsymbol{z}^*))\right) < 1$.*
>
> *(ii) The perturbation satisfies $\|\mathcal{R}(n, \boldsymbol{z})\| \leq C'\rho^n\|\boldsymbol{z} - \boldsymbol{z}^*\|$ for some $C' > 0$, $\rho \in (0,1)$, uniformly for $\boldsymbol{z}$ in a neighborhood of $\boldsymbol{z}^*$.*

*Then there exists a neighborhood $U$ of $z^*$ and constant $0 < \tilde{\gamma} < 1$ such that for any $z_0 \in U$, the iterates of $\{\boldsymbol{z}_n\}$ satisfy*

$$\|\boldsymbol{z}_n - \boldsymbol{z}^*\|_2 = \mathcal{O}\left(\tilde{\gamma}^n\|\boldsymbol{z}_1 - \boldsymbol{z}^*\|_2\right).$$

*Proof.* By Lemma [C.8](#), there exists a norm $\|\cdot\|_P$ and $\gamma \in (0,1)$ such that

$$\|\bar{T}(\boldsymbol{z}) - \boldsymbol{z}^*\|_P = \|\bar{T}(\boldsymbol{z}) - \bar{T}(\boldsymbol{z}^*)\|_P = \|\bar{T}(\boldsymbol{z} - \boldsymbol{z}^*)\|_P \leq \gamma\|\boldsymbol{z} - \boldsymbol{z}^*\|_P$$

for $\boldsymbol{z}$ close to $\boldsymbol{z}^*$. Now write the iteration as

$$\boldsymbol{z}_{n+1} - \boldsymbol{z}^* = \bar{T}(\boldsymbol{z}_n) - \bar{T}(\boldsymbol{z}^*) + \mathcal{R}(n, \boldsymbol{z}_n).$$

Taking the $\|\cdot\|_P$ norm,

$$\|\boldsymbol{z}_{n+1} - \boldsymbol{z}^*\|_P \leq \gamma\|\boldsymbol{z}_n - \boldsymbol{z}^*\|_P + \|\mathcal{R}(n, \boldsymbol{z}_n)\|_P.$$

By assumption, $\|R(n, z_n)\|_P \leq C'\rho^n\|z_n - z^*\|_P$ for some $C' > 0$. Thus

$$\|\boldsymbol{z}_{n+1} - \boldsymbol{z}^*\|_P \leq (\gamma + C'\rho^n)\|\boldsymbol{z}_n - \boldsymbol{z}^*\|_P.$$

Since $\rho^n \to 0$, for sufficiently large $n$ we have $\gamma + C'\rho^n < \tilde{\gamma} < 1$. Therefore the sequence contracts at rate $\tilde{\gamma} < 1$, implying local convergence:

$$\|\boldsymbol{z}_n - \boldsymbol{z}^*\|_P \leq \tilde{\gamma}^{n-1}\|\boldsymbol{z}_1 - \boldsymbol{z}^*\|_P.$$

By the Equivalence of norms in finite dimensions, we have there exists $c_1, c_2 > 0$ such that

$$c_1\|\boldsymbol{z}\|_2 \leq \|\boldsymbol{z}\|_P \leq c_2\|\boldsymbol{z}\|_2, \quad \forall \boldsymbol{z}.$$

This gives the local convergence with an exponential rate in the Euclidean norm as well, i.e.,

$$\|\boldsymbol{z}_n - \boldsymbol{z}^*\|_2 \leq \frac{c_2}{c_1}\tilde{\gamma}^{n-1}\|\boldsymbol{z}_1 - \boldsymbol{z}^*\|_2.$$

$\square$

**Lemma C.10.** *Suppose that $f(\boldsymbol{x}, \boldsymbol{y})$ satisfies Assumption [4.2](#). Let $\beta \in (-1, 1)$ and $0 < \rho < 1$. Set $h$, $\epsilon$ and $\beta$ such that*

$$h < \min_{\lambda \in \mathrm{Sp}(\mathcal{J})} \frac{2\sqrt{\epsilon}(1 - \beta^2)|\Re(\lambda)|}{(1 + \beta^2)|\lambda|^2 + 2\beta\left(|\Im(\lambda)|^2 - |\Re(\lambda)|^2\right)}.$$

*Then $\bar{T}$ converges locally with an exponential rate.*

*Proof.* According to Lemma [C.6](#), we complete the proof by proving $\varrho\left(Jac(\bar{T})\right) < 1$.

By direct computation, we can get the Jacobian matrix of $\bar{T}$ at $\boldsymbol{z}^* = (\boldsymbol{0}, \boldsymbol{0}, \boldsymbol{x}^*, \boldsymbol{0}, \boldsymbol{0}, \boldsymbol{y}^*)$ is

$$\mathcal{M}_{\mathcal{S}} = \begin{bmatrix} \beta\mathcal{I} & \boldsymbol{0} & (1-\beta)\nabla^2_{\boldsymbol{x}}f(\boldsymbol{x}^*, \boldsymbol{y}^*) & \boldsymbol{0} & \boldsymbol{0} & (1-\beta)\nabla_{xy}f(\boldsymbol{x}^*, \boldsymbol{y}^*) \\ \boldsymbol{0} & \rho\mathcal{I} & \boldsymbol{0} & \boldsymbol{0} & \boldsymbol{0} & \boldsymbol{0} \\ -\frac{h\beta}{\sqrt{\epsilon}}\mathcal{I} & \boldsymbol{0} & \mathcal{I} - \frac{h(1-\beta)}{\sqrt{\epsilon}}\nabla^2_{\boldsymbol{x}}f(\boldsymbol{x}^*, \boldsymbol{y}^*) & \boldsymbol{0} & \boldsymbol{0} & -\frac{h(1-\beta)}{\sqrt{\epsilon}}\nabla_{xy}f(\boldsymbol{x}^*, \boldsymbol{y}^*) \\ \boldsymbol{0} & \boldsymbol{0} & (1-\beta)\nabla_{yx}f(\boldsymbol{x}^*, \boldsymbol{y}^*) & \beta\mathcal{I} & \boldsymbol{0} & (1-\beta)\nabla^2_{\boldsymbol{y}}f(\boldsymbol{x}^*, \boldsymbol{y}^*) \\ \boldsymbol{0} & \boldsymbol{0} & \boldsymbol{0} & \boldsymbol{0} & \rho\mathcal{I} & \boldsymbol{0} \\ \boldsymbol{0} & \boldsymbol{0} & \frac{h(1-\beta)}{\sqrt{\epsilon}}\nabla_{yx}f(\boldsymbol{x}^*, \boldsymbol{y}^*) & \frac{h\beta}{\sqrt{\epsilon}}\mathcal{I} & \boldsymbol{0} & \mathcal{I} + \frac{h(1-\beta)}{\sqrt{\epsilon}}\nabla^2_{\boldsymbol{y}}f(\boldsymbol{x}^*, \boldsymbol{y}^*) \end{bmatrix}.$$

Obviously, $\mathcal{M}_S$ has $d_1 + d_2$ eigenvalues $\rho$. Next, we consider the following matrix

$$
\mathcal{M}_{\mathcal{S}}^{(1)} = \begin{bmatrix}
\beta\mathcal{I} & (1-\beta)\nabla_{\boldsymbol{x}}^2 f(\boldsymbol{x}^*, \boldsymbol{y}^*) & \mathbf{0} & (1-\beta)\nabla_{xy} f(\boldsymbol{x}^*, \boldsymbol{y}^*) \\
-\frac{h\beta}{\sqrt{\epsilon}}\mathcal{I} & \mathcal{I} - \frac{h(1-\beta)}{\sqrt{\epsilon}}\nabla_{\boldsymbol{x}}^2 f(\boldsymbol{x}^*, \boldsymbol{y}^*) & \mathbf{0} & -\frac{h(1-\beta)}{\sqrt{\epsilon}}\nabla_{xy} f(\boldsymbol{x}^*, \boldsymbol{y}^*) \\
\mathbf{0} & (1-\beta)\nabla_{yx} f(\boldsymbol{x}^*, \boldsymbol{y}^*) & \beta\mathcal{I} & (1-\beta)\nabla_{\boldsymbol{y}}^2 f(\boldsymbol{x}^*, \boldsymbol{y}^*) \\
\mathbf{0} & \frac{h(1-\beta)}{\sqrt{\epsilon}}\nabla_{yx} f(\boldsymbol{x}^*, \boldsymbol{y}^*) & \frac{h\beta}{\sqrt{\epsilon}}\mathcal{I} & \mathcal{I} + \frac{h(1-\beta)}{\sqrt{\epsilon}}\nabla_{\boldsymbol{y}}^2 f(\boldsymbol{x}^*, \boldsymbol{y}^*)
\end{bmatrix}.
$$

Since $0 < \rho < 1$, if we want to prove $\varrho\left(\mathrm{Jac}(\bar{T})\right) < 1$, we only need to ensure $\varrho\left(\mathcal{M}_{\mathcal{S}}^{(1)}\right) < 1$.

Exchange the 1st and 4th rows, as well as 1st and 4th columns of $\mathcal{M}_{\mathcal{S}}^{(1)}$, we can get

$$
\mathcal{M}_{\mathcal{S}}^{(2)} = \begin{bmatrix}
\mathcal{I} + \frac{h(1-\beta)}{\sqrt{\epsilon}}\nabla_y^2 f(\boldsymbol{x}^*, \boldsymbol{y}^*) & \frac{h(1-\beta)}{\sqrt{\epsilon}}\nabla_{yx} f(\boldsymbol{x}^*, \boldsymbol{y}^*) & \frac{h\beta}{\sqrt{\epsilon}}\mathcal{I} & \mathbf{0} \\
-\frac{h(1-\beta)}{\sqrt{\epsilon}}\nabla_{xy} f(\boldsymbol{x}^*, \boldsymbol{y}^*) & \mathcal{I} - \frac{h(1-\beta)}{\sqrt{\epsilon}}\nabla_x^2 f(\boldsymbol{x}^*, \boldsymbol{y}^*) & \mathbf{0} & -\frac{h\beta}{\sqrt{\epsilon}}\mathcal{I} \\
(1-\beta)\nabla_y^2 f(\boldsymbol{x}^*, \boldsymbol{y}^*) & (1-\beta)\nabla_{yx} f(\boldsymbol{x}^*, \boldsymbol{y}^*) & \beta\mathcal{I} & \mathbf{0} \\
(1-\beta)\nabla_{xy} f(\boldsymbol{x}^*, \boldsymbol{y}^*) & (1-\beta)\nabla_x^2 f(\boldsymbol{x}^*, \boldsymbol{y}^*) & \mathbf{0} & \beta\mathcal{I}
\end{bmatrix}.
$$

Then we only need to ensure $\varrho\left(\mathcal{M}_{\mathcal{S}}^{(2)}\right) < 1$. Calculate its characteristic polynomial:

$$
\det(\hat{\mu}\mathcal{I} - \mathcal{M}_{\mathcal{S}}^{(2)})
$$
$$
= \det\left(\begin{bmatrix}
(\hat{\mu} - 1)\mathcal{I} - \frac{h(1-\beta)}{\sqrt{\epsilon}}\nabla_y^2 f(\boldsymbol{x}^*, \boldsymbol{y}^*) & -\frac{h(1-\beta)}{\sqrt{\epsilon}}\nabla_{yx} f(\boldsymbol{x}^*, \boldsymbol{y}^*) & -\frac{h\beta}{\sqrt{\epsilon}}\mathcal{I} & \mathbf{0} \\
\frac{h(1-\beta)}{\sqrt{\epsilon}}\nabla_{xy} f(\boldsymbol{x}^*, \boldsymbol{y}^*) & (\hat{\mu} - 1)\mathcal{I} + \frac{h(1-\beta)}{\sqrt{\epsilon}}\nabla_x^2 f(\boldsymbol{x}^*, \boldsymbol{y}^*) & \mathbf{0} & \frac{h\beta}{\sqrt{\epsilon}}\mathcal{I} \\
-(1-\beta)\nabla_y^2 f(\boldsymbol{x}^*, \boldsymbol{y}^*) & -(1-\beta)\nabla_{yx} f(\boldsymbol{x}^*, \boldsymbol{y}^*) & (\hat{\mu} - \beta)\mathcal{I} & \mathbf{0} \\
-(1-\beta)\nabla_{xy} f(\boldsymbol{x}^*, \boldsymbol{y}^*) & -(1-\beta)\nabla_x^2 f(\boldsymbol{x}^*, \boldsymbol{y}^*) & \mathbf{0} & (\hat{\mu} - \beta)\mathcal{I}
\end{bmatrix}\right).
$$

Let

$$
A = \begin{bmatrix}
(\hat{\mu} - 1)\mathcal{I} - \frac{h(1-\beta)}{\sqrt{\epsilon}}\nabla_y^2 f(\boldsymbol{x}^*, \boldsymbol{y}^*) & -\frac{h(1-\beta)}{\sqrt{\epsilon}}\nabla_{yx} f(\boldsymbol{x}^*, \boldsymbol{y}^*) \\
\frac{h(1-\beta)}{\sqrt{\epsilon}}\nabla_{xy} f(\boldsymbol{x}^*, \boldsymbol{y}^*) & (\hat{\mu} - 1)\mathcal{I} + \frac{h(1-\beta)}{\sqrt{\epsilon}}\nabla_x^2 f(\boldsymbol{x}^*, \boldsymbol{y}^*)
\end{bmatrix}, B = \begin{bmatrix}
-\frac{h\beta}{\sqrt{\epsilon}}\mathcal{I} & \mathbf{0} \\
\mathbf{0} & \frac{h\beta}{\sqrt{\epsilon}}\mathcal{I}
\end{bmatrix},
$$

and

$$
C = \begin{bmatrix}
-(1-\beta)\nabla_y^2 f(\boldsymbol{x}^*, \boldsymbol{y}^*) & -(1-\beta)\nabla_{yx} f(\boldsymbol{x}^*, \boldsymbol{y}^*) \\
-(1-\beta)\nabla_{xy} f(\boldsymbol{x}^*, \boldsymbol{y}^*) & -(1-\beta)\nabla_x^2 f(\boldsymbol{x}^*, \boldsymbol{y}^*)
\end{bmatrix}, \quad D = \begin{bmatrix}
(\hat{\mu} - \beta)\mathcal{I} & \mathbf{0} \\
\mathbf{0} & (\hat{\mu} - \beta)\mathcal{I}
\end{bmatrix}.
$$

Without loss of generality, we assume $\hat{\mu} \neq \beta$. (If $\hat{\mu} = \beta$, we can easily verify $\beta$ is the unique eigenvalue of $\mathcal{M}_{\mathcal{S}}^{(2)}$. We only need to set $\beta \in (-1, 1)$ to ensure $\varrho\left(\mathcal{M}_{\mathcal{S}}^{(2)}\right) < 1$.)

We state a fact that if $D$ is invertible, then $\det\left(\begin{bmatrix} A & B \\ C & D \end{bmatrix}\right) = \det(D)\det(A - BD^{-1}C)$.

According to this fact, we can get

$$\det(\hat{\mu}\mathcal{I} - \mathcal{M}_{\mathcal{S}}^{(2)})$$

$$= (\hat{\mu} - \beta)^{d_1+d_2} \det \left( \begin{bmatrix} (\hat{\mu}-1)\mathcal{I} - \frac{h(1-\beta)}{\sqrt{\epsilon}}\nabla_y^2 f(\boldsymbol{x}^*, \boldsymbol{y}^*) & -\frac{h(1-\beta)}{\sqrt{\epsilon}}\nabla_{yx} f(\boldsymbol{x}^*, \boldsymbol{y}^*) \\\\ \frac{h(1-\beta)}{\sqrt{\epsilon}}\nabla_{xy} f(\boldsymbol{x}^*, \boldsymbol{y}^*) & (\hat{\mu}-1)\mathcal{I} + \frac{h(1-\beta)}{\sqrt{\epsilon}}\nabla_x^2 f(\boldsymbol{x}^*, \boldsymbol{y}^*) \end{bmatrix} \right.$$

$$\left. - \begin{bmatrix} -\frac{h\beta}{\sqrt{\epsilon}}\mathcal{I} & \mathbf{0} \\\\ \mathbf{0} & \frac{h\beta}{\sqrt{\epsilon}}\mathcal{I} \end{bmatrix} \begin{bmatrix} \frac{1}{\hat{\mu}-\beta}\mathcal{I} & \mathbf{0} \\\\ \mathbf{0} & \frac{1}{\hat{\mu}-\beta}\mathcal{I} \end{bmatrix} \begin{bmatrix} -(1-\beta)\nabla_y^2 f(\boldsymbol{x}^*, \boldsymbol{y}^*) & -(1-\beta)\nabla_{yx}^2 f(\boldsymbol{x}^*, \boldsymbol{y}^*) \\\\ -(1-\beta)\nabla_{xy}^2 f(\boldsymbol{x}^*, \boldsymbol{y}^*) & -(1-\beta)\nabla_x^2 f(\boldsymbol{x}^*, \boldsymbol{y}^*) \end{bmatrix} \right)$$

$$= (\hat{\mu} - \beta)^{d_1+d_2} \det \left( \begin{bmatrix} (\hat{\mu}-1)\mathcal{I} - \frac{h(1-\beta)\hat{\mu}}{\sqrt{\epsilon}(\hat{\mu}-\beta)}\nabla_y^2 f(\boldsymbol{x}^*, \boldsymbol{y}^*) & -\frac{h(1-\beta)\hat{\mu}}{\sqrt{\epsilon}(\hat{\mu}-\beta)}\nabla_{yx} f(\boldsymbol{x}^*, \boldsymbol{y}^*) \\\\ \frac{h(1-\beta)\hat{\mu}}{\sqrt{\epsilon}(\hat{\mu}-\beta)}\nabla_{xy} f(\boldsymbol{x}^*, \boldsymbol{y}^*) & (\hat{\mu}-1)\mathcal{I} + \frac{h(1-\beta)\hat{\mu}}{\sqrt{\epsilon}(\hat{\mu}-\beta)}\nabla_x^2 f(\boldsymbol{x}^*, \boldsymbol{y}^*) \end{bmatrix} \right)$$

$$= (\hat{\mu} - \beta)^{d_1+d_2} \det \left( \begin{bmatrix} \hat{\mu}\mathcal{I} & \mathbf{0} \\\\ \mathbf{0} & \hat{\mu}\mathcal{I} \end{bmatrix} - \left( \begin{bmatrix} \mathcal{I} & \mathbf{0} \\\\ \mathbf{0} & \mathcal{I} \end{bmatrix} - \frac{h(1-\beta)\hat{\mu}}{\sqrt{\epsilon}(\hat{\mu}-\beta)} \begin{bmatrix} -\nabla_y^2 f(\boldsymbol{x}^*, \boldsymbol{y}^*) & -\nabla_{yx} f(\boldsymbol{x}^*, \boldsymbol{y}^*) \\\\ \nabla_{xy} f(\boldsymbol{x}^*, \boldsymbol{y}^*) & \nabla_x^2 f(\boldsymbol{x}^*, \boldsymbol{y}^*) \end{bmatrix} \right) \right).$$

Let $\det(\hat{\mu}\mathcal{I} - \mathcal{M}_{\mathcal{S}}^{(2)}) = 0$. Obviously, $\beta \in (-1, 1)$ have ensured that the $d_1 + d_2$ roots $\hat{\mu} = \beta$ lie in the unit disk, we only need to consider

$$\det \left( \begin{bmatrix} \hat{\mu}\mathcal{I} & \mathbf{0} \\\\ \mathbf{0} & \hat{\mu}\mathcal{I} \end{bmatrix} - \left( \begin{bmatrix} \mathcal{I} & \mathbf{0} \\\\ \mathbf{0} & \mathcal{I} \end{bmatrix} - \frac{h(1-\beta)\hat{\mu}}{\sqrt{\epsilon}(\hat{\mu}-\beta)} \begin{bmatrix} -\nabla_y^2 f(\boldsymbol{x}^*, \boldsymbol{y}^*) & -\nabla_{yx} f(\boldsymbol{x}^*, \boldsymbol{y}^*) \\\\ \nabla_{xy} f(\boldsymbol{x}^*, \boldsymbol{y}^*) & \nabla_x^2 f(\boldsymbol{x}^*, \boldsymbol{y}^*) \end{bmatrix} \right) \right) = \mathbf{0},$$

which means $\hat{\mu}$ is the eigenvalue of the following matrix

$$N := \begin{bmatrix} \mathcal{I} & \mathbf{0} \\\\ \mathbf{0} & \mathcal{I} \end{bmatrix} - \frac{h(1-\beta)\hat{\mu}}{\sqrt{\epsilon}(\hat{\mu}-\beta)} \begin{bmatrix} -\nabla_y^2 f(\boldsymbol{x}^*, \boldsymbol{y}^*) & -\nabla_{yx} f(\boldsymbol{x}^*, \boldsymbol{y}^*) \\\\ \nabla_{xy} f(\boldsymbol{x}^*, \boldsymbol{y}^*) & \nabla_x^2 f(\boldsymbol{x}^*, \boldsymbol{y}^*) \end{bmatrix}.$$

Note that

$$\mathrm{Sp} \left( \begin{bmatrix} -\nabla_y^2 f(\boldsymbol{x}^*, \boldsymbol{y}^*) & -\nabla_{yx} f(\boldsymbol{x}^*, \boldsymbol{y}^*) \\\\ \nabla_{xy} f(\boldsymbol{x}^*, \boldsymbol{y}^*) & \nabla_x^2 f(\boldsymbol{x}^*, \boldsymbol{y}^*) \end{bmatrix} \right) = \mathrm{Sp} \left( \begin{bmatrix} \nabla_x^2 f(\boldsymbol{x}^*, \boldsymbol{y}^*) & \nabla_{xy} f(\boldsymbol{x}^*, \boldsymbol{y}^*) \\\\ -\nabla_{yx} f(\boldsymbol{x}^*, \boldsymbol{y}^*) & -\nabla_y^2 f(\boldsymbol{x}^*, \boldsymbol{y}^*) \end{bmatrix} \right)$$

$$= -\mathrm{Sp} \left( \begin{bmatrix} -\nabla_x^2 f(\boldsymbol{x}^*, \boldsymbol{y}^*) & -\nabla_{xy} f(\boldsymbol{x}^*, \boldsymbol{y}^*) \\\\ \nabla_{yx} f(\boldsymbol{x}^*, \boldsymbol{y}^*) & \nabla_y^2 f(\boldsymbol{x}^*, \boldsymbol{y}^*) \end{bmatrix} \right) = -\mathrm{Sp}(\mathcal{J}).$$

Let $\{\lambda_i\}_{i=1}^{d_1+d_2}$ be the eigenvalues of the matrix $\mathcal{J}$. Then $\left\{ 1 - \frac{h(1-\beta)\hat{\mu}}{\sqrt{\epsilon}(\hat{\mu}-\beta)}(-\lambda_i) \right\}_{i=1}^{d_1+d_2}$ are the eigenvalues of $N$. Then we can get

$$\hat{\mu} = 1 - \frac{h(1-\beta)\hat{\mu}}{\sqrt{\epsilon}(\hat{\mu}-\beta)}(-\lambda_i), \quad i = 1, 2, \cdots, d_1 + d_2. \tag{34}$$

Solving the equation (34), we can get all the eigenvalues of $\mathcal{M}_{\mathcal{S}}^{(2)}$.

Rewriting (34), we can get

$$\hat{\mu}^2 - \left(\beta + 1 + \frac{h(1-\beta)}{\sqrt{\epsilon}}\lambda_i\right)\hat{\mu} + \beta = 0. \tag{35}$$

Applying Lemma C.7 for (35) with

$$a = -\left(\beta + 1 - \frac{h(1-\beta)}{\sqrt{\epsilon}}\lambda_i\right), \quad b = \beta,$$

and solving the resulting inequalities, we can get

$$h < \min_i \frac{-2\sqrt{\epsilon}(1-\beta^2)\Re(\lambda_i)}{(1+\beta^2)|\lambda_i|^2 + 2\beta\left(|\Im(\lambda_i)|^2 - |\Re(\lambda_i)|^2\right)} \tag{36}$$

$$= \min_{\lambda \in \mathrm{Sp}(\mathcal{J})} \frac{2\sqrt{\epsilon}(1-\beta^2)|\Re(\lambda)|}{(1+\beta^2)|\lambda|^2 + 2\beta\left(|\Im(\lambda)|^2 - |\Re(\lambda)|^2\right)}, \tag{37}$$

where we use the fact that Assumption 4.2 implies $\Re(\lambda_i) < 0$ in the equality.

This means all roots of (35) lie within the open unit disk of the complex plane, which means the spectral radius of $\bar{T}$ is less than 1, i.e., $\varrho(\mathrm{Jac}(\bar{T})) < 1$. $\qquad\square$

**Lemma C.11.** *Assume $f(x)$ is $C^2$. Then the non-autonomous system $\mathcal{R}(n, \boldsymbol{z}_n)$ converges locally with an exponential rate, i.e., there exists $0 < Q < 1$ such that*

$$\|\mathcal{R}(n, \boldsymbol{z}_n)\| = \mathcal{O}(Q^n\|\boldsymbol{z}_n - \boldsymbol{z}^*\|).$$

*Proof.* For the non-autonomous system, we have

$$\|\mathcal{R}(n, \boldsymbol{z}_n)\|^2 = h^2 \underbrace{\left\|\frac{\tilde{\boldsymbol{m}}_{n+1}}{\sqrt{\tilde{\boldsymbol{v}}_{n+1}} + \epsilon} - \frac{\sqrt{1-\rho^{n+1}}}{1-\beta^{n+1}}\frac{\tilde{\boldsymbol{m}}_{n+1}}{\sqrt{\tilde{\boldsymbol{v}}_{n+1} + (1-\rho^{n+1})\epsilon}}\right\|^2}_{\text{Term1}}$$

$$+ h^2 \underbrace{\left\|\frac{\hat{\boldsymbol{m}}_{n+1}}{\sqrt{\hat{\boldsymbol{v}}_{n+1}} + \epsilon} - \frac{\sqrt{1-\rho^{n+1}}}{1-\beta^{n+1}}\frac{\hat{\boldsymbol{m}}_{n+1}}{\sqrt{\hat{\boldsymbol{v}}_{n+1} + (1-\rho^{n+1})\epsilon}}\right\|^2}_{\text{Term2}}.$$

For Term 1, we have

Term1

$$= h^2 \left\|\frac{\tilde{\boldsymbol{m}}_{n+1}}{\sqrt{\tilde{\boldsymbol{v}}_{n+1}} + \epsilon} - \frac{\sqrt{1-\rho^{n+1}}}{1-\beta^{n+1}}\frac{\tilde{\boldsymbol{m}}_{n+1}}{\sqrt{\tilde{\boldsymbol{v}}_{n+1}} + \epsilon} + \frac{\sqrt{1-\rho^{n+1}}}{1-\beta^{n+1}}\left(\frac{\tilde{\boldsymbol{m}}_{n+1}}{\sqrt{\tilde{\boldsymbol{v}}_{n+1}} + \epsilon} - \frac{\tilde{\boldsymbol{m}}_{n+1}}{\sqrt{\tilde{\boldsymbol{v}}_{n+1} + (1-\rho^{n+1})\epsilon}}\right)\right\|^2$$

$$\leq 2h^2 \left\|\frac{\tilde{\boldsymbol{m}}_{n+1}}{\sqrt{\tilde{\boldsymbol{v}}_{n+1}} + \epsilon} - \frac{\sqrt{1-\rho^{n+1}}}{1-\beta^{n+1}}\frac{\tilde{\boldsymbol{m}}_{n+1}}{\sqrt{\tilde{\boldsymbol{v}}_{n+1}} + \epsilon}\right\|^2 + 2h^2 \left\|\frac{\sqrt{1-\rho^{n+1}}}{1-\beta^{n+1}}\left(\frac{\tilde{\boldsymbol{m}}_{n+1}}{\sqrt{\tilde{\boldsymbol{v}}_{n+1}} + \epsilon} - \frac{\tilde{\boldsymbol{m}}_{n+1}}{\sqrt{\tilde{\boldsymbol{v}}_{n+1} + (1-\rho^{n+1})\epsilon}}\right)\right\|^2$$

$$= 2h^2 \left(1 - \frac{\sqrt{1-\rho^{n+1}}}{1-\beta^{n+1}}\right)^2 \frac{\|\tilde{\boldsymbol{m}}_{n+1}\|^2}{\tilde{\boldsymbol{v}}_{n+1} + \epsilon} + 2h^2 \left(\frac{\sqrt{1-\rho^{n+1}}}{1-\beta^{n+1}}\right)^2 \left\|\frac{\tilde{\boldsymbol{m}}_{n+1}}{\sqrt{\tilde{\boldsymbol{v}}_{n+1}} + \epsilon} - \frac{\tilde{\boldsymbol{m}}_{n+1}}{\sqrt{\tilde{\boldsymbol{v}}_{n+1} + (1-\rho^{n+1})\epsilon}}\right\|^2$$

$$\leq \frac{2h^2\|\tilde{\boldsymbol{m}}_{n+1}\|^2}{\epsilon}\left(\frac{\sqrt{1-\rho^{n+1}} - (1-\beta^{n+1})}{1-\beta^{n+1}}\right)^2 + \frac{2h^2\|\tilde{\boldsymbol{m}}_{n+1}\|^2}{(1-\beta)^2}\left(\frac{1}{\sqrt{\tilde{\boldsymbol{v}}_{n+1}} + \epsilon} - \frac{1}{\sqrt{\tilde{\boldsymbol{v}}_{n+1} + (1-\rho^{n+1})\epsilon}}\right)^2,$$

where we use the fact $\|\boldsymbol{u} + \boldsymbol{v}\|^2 \leq 2(\|\boldsymbol{u}\|^2 + \|\boldsymbol{v}\|^2)$ in the first inequality.

Direct computation yields

$$\left( \frac{\sqrt{1 - \rho^{n+1}} - (1 - \beta^{n+1})}{1 - \beta^{n+1}} \right)^2 = \left( \frac{1 - \rho^{n+1} - (1 - \beta^{n+1})^2}{(1 - \beta^{n+1}) \left( \sqrt{1 - \rho^{n+1}} + (1 - \beta^{n+1}) \right)} \right)^2$$

$$\leq \left( \frac{-\rho^{n+1} + 2\beta^{n+1} - (\rho^2)^{n+1}}{(1 - \beta) (\sqrt{1 - \rho} + 1 - \beta)} \right)^2$$

$$\leq \left( \frac{|-\rho^{n+1}| + 2|\beta^{n+1}| + |-(\rho^2)^{n+1}|}{(1 - \beta) (\sqrt{1 - \rho} + 1 - \beta)} \right)^2$$

$$\leq \left( \frac{4Q^{n+1}}{(1 - \beta) (\sqrt{1 - \rho} + 1 - \beta)} \right)^2 = C_1 Q^{2n+2}$$

Here

$$Q = \max\{|\rho|, |\beta|^2, |\rho^2|\}, \quad C_1 = \frac{16}{(1 - \beta)^2 (\sqrt{1 - \rho} + 1 - \beta)^2},$$

and we use the fact

$$|a + b + c|^2 \leq (|a| + |b| + |c|)^2$$

in the second inequality.

Simple algebra operation yields

$$\left( \frac{1}{\sqrt{\tilde{\boldsymbol{v}}_{n+1} + \epsilon}} - \frac{1}{\sqrt{\tilde{\boldsymbol{v}}_{n+1} + (1 - \rho^{n+1})\epsilon}} \right)^2$$

$$= \left( \frac{\rho^{n+1}\epsilon}{\sqrt{\tilde{\boldsymbol{v}}_{n+1} + \epsilon} \sqrt{\tilde{\boldsymbol{v}}_{n+1} + (1 - \rho^{n+1})\epsilon} \left( \sqrt{\tilde{\boldsymbol{v}}_{n+1} + \epsilon} + \sqrt{\tilde{\boldsymbol{v}}_{n+1} + (1 - \rho^{n+1})\epsilon} \right)} \right)^2$$

$$\leq \left( \frac{\rho^{n+1}\epsilon}{\sqrt{\epsilon} \sqrt{(1 - \rho)\epsilon} \left( \sqrt{\epsilon} + \sqrt{(1 - \rho)\epsilon} \right)} \right)^2$$

$$= \left( \frac{\rho^{n+1}}{\sqrt{\epsilon} \sqrt{(1 - \rho)} \left( 1 + \sqrt{1 - \rho} \right)} \right)^2 = C_2 \rho^{2n+2},$$

where

$$C_2 = \frac{1}{\epsilon(1 - \rho)(1 + \sqrt{1 - \rho})^2}.$$

Since $f(x)$ is $C^2$, $\|\nabla^2 f(\boldsymbol{x}, \boldsymbol{y})\|$ is bounded in any bounded set, which means $\nabla_x f(\boldsymbol{x}, \boldsymbol{y})$ is locally Lipschitz in any bounded set, i.e., there exists $L > 0$ such that for all $(\boldsymbol{x}_1, \boldsymbol{y}_1), (\boldsymbol{x}_2, \boldsymbol{y}_2)$ in the neighborhood $B(\boldsymbol{x}^*, \boldsymbol{y}^*)$ of $(\boldsymbol{x}^*, \boldsymbol{y}^*)$,

$$\|\nabla_x f(\boldsymbol{x}_1, \boldsymbol{y}_1) - \nabla_x f(\boldsymbol{x}_2, \boldsymbol{y}_2)\| \leq L \|(\boldsymbol{x}_1, \boldsymbol{y}_1) - (\boldsymbol{x}_2, \boldsymbol{y}_2)\|.$$

Recall that the fixed point $(\tilde{\boldsymbol{m}}^*, \tilde{\boldsymbol{v}}^*, \boldsymbol{x}^*, \hat{\boldsymbol{m}}^*, \hat{\boldsymbol{v}}^*, \boldsymbol{y}^*) = (\boldsymbol{0}, \boldsymbol{0}, \boldsymbol{x}^*, \boldsymbol{0}, \boldsymbol{0}, \boldsymbol{y}^*)$.

$$\|\tilde{\boldsymbol{m}}_{n+1}\| = \|\tilde{\boldsymbol{m}}_{n+1} - \tilde{\boldsymbol{m}}^*\|$$

$$\leq \beta \|\tilde{\boldsymbol{m}}_n - \tilde{\boldsymbol{m}}^*\| + (1 - \beta) \|\nabla_x f^{(n)}(\boldsymbol{x}_n, \boldsymbol{y}_n) - \nabla_x f^{(n)}(\boldsymbol{x}^*, \boldsymbol{y}^*)\| \quad (38)$$

$$\leq \beta \|\tilde{\boldsymbol{m}}_n - \tilde{\boldsymbol{m}}^*\| + (1 - \beta) L \|(\boldsymbol{x}_n, \boldsymbol{y}_n) - (\boldsymbol{x}^*, \boldsymbol{y}^*)\|.$$

In fact,
$$\beta\|\tilde{\boldsymbol{m}} - \tilde{\boldsymbol{m}}^*\| + (1-\beta)L\|(\tilde{\boldsymbol{x}}, \tilde{\boldsymbol{y}}) - (\boldsymbol{x}^*, \boldsymbol{y}^*)\|$$
corresponds to a norm
$$\|(\tilde{\boldsymbol{m}}, \tilde{\boldsymbol{x}}, \tilde{\boldsymbol{y}})\|_* := \beta\|\tilde{\boldsymbol{m}}\| + (1-\beta)L\|(\tilde{\boldsymbol{x}}, \tilde{\boldsymbol{y}})\|. \tag{39}$$

Next, we verify that $\|\cdot\|_*$ is exactly a norm:

- **Positive definiteness.** Obviously, we have
$$\|(\tilde{\boldsymbol{m}}, \tilde{\boldsymbol{x}}, \tilde{\boldsymbol{y}})\|_* = \beta\,\|\tilde{\boldsymbol{m}}\| + (1-\beta)L\,\|(\tilde{\boldsymbol{x}}, \tilde{\boldsymbol{y}})\| \geq 0$$
  and $\|(\tilde{\boldsymbol{m}}, \tilde{\boldsymbol{x}}, \tilde{\boldsymbol{y}})\|_* = \boldsymbol{0}$ iff $(\tilde{\boldsymbol{m}}, \tilde{\boldsymbol{x}}, \tilde{\boldsymbol{y}}) = (\boldsymbol{0}, \boldsymbol{0}, \boldsymbol{0})$, since $0 < \beta < 1$ and $L > 0$.

- **Absolute homogeneity.** For all $a$ and all $(\tilde{\boldsymbol{m}}, \tilde{\boldsymbol{x}}, \tilde{\boldsymbol{y}})$, we have
$$\begin{aligned}
\|a\,(\tilde{\boldsymbol{m}}, \tilde{\boldsymbol{x}}, \tilde{\boldsymbol{y}})\|_* &= \beta\,\||a|\,\tilde{\boldsymbol{m}}\| + (1-\beta)L\,\||a|\,(\tilde{\boldsymbol{x}}, \tilde{\boldsymbol{y}})\| \\
&= |a|\,\beta\,\|\tilde{\boldsymbol{m}}\| + |a|\,(1-\beta)L\,\|(\tilde{\boldsymbol{x}}, \tilde{\boldsymbol{y}})\| \\
&= |a|\,(\beta\,\|\tilde{\boldsymbol{m}}\| + (1-\beta)L\,\|(\tilde{\boldsymbol{x}}, \tilde{\boldsymbol{y}})\|) \\
&= |a|\,\|(\tilde{\boldsymbol{m}}, \tilde{\boldsymbol{x}}, \tilde{\boldsymbol{y}})\|_*
\end{aligned}$$

- **Triangle inequality.** For all $(\tilde{\boldsymbol{m}}_1, \tilde{\boldsymbol{x}}_1, \tilde{\boldsymbol{y}}_1)$ and $(\tilde{\boldsymbol{m}}_2, \tilde{\boldsymbol{x}}_2, \tilde{\boldsymbol{y}}_2)$, we have
$$\begin{aligned}
&\|(\tilde{\boldsymbol{m}}_1, \tilde{\boldsymbol{x}}_1, \tilde{\boldsymbol{y}}_1) + (\tilde{\boldsymbol{m}}_2, \tilde{\boldsymbol{x}}_2, \tilde{\boldsymbol{y}}_2)\|_* \\
&= \|(\tilde{\boldsymbol{m}}_1 + \tilde{\boldsymbol{m}}_2, \tilde{\boldsymbol{x}}_1 + \tilde{\boldsymbol{x}}_2, \tilde{\boldsymbol{y}}_1 + \tilde{\boldsymbol{y}}_2)\|_* \\
&= \beta\,\|\tilde{\boldsymbol{m}}_1 + \tilde{\boldsymbol{m}}_2\| + (1-\beta)L\,\|(\tilde{\boldsymbol{x}}_1 + \tilde{\boldsymbol{x}}_2, \tilde{\boldsymbol{y}}_1 + \tilde{\boldsymbol{y}}_2)\| \\
&\leq \beta\,(\|\tilde{\boldsymbol{m}}_1\| + \|\tilde{\boldsymbol{m}}_2\|) + (1-\beta)L\,(\|(\tilde{\boldsymbol{x}}_1, \tilde{\boldsymbol{y}}_1)\| + \|(\tilde{\boldsymbol{x}}_2, \tilde{\boldsymbol{y}}_2)\|) \\
&= \beta\,\|\tilde{\boldsymbol{m}}_1\| + (1-\beta)L\,\|(\tilde{\boldsymbol{x}}_1, \tilde{\boldsymbol{y}}_1)\| + \beta\,\|\tilde{\boldsymbol{m}}_2\| + (1-\beta)L\,\|(\tilde{\boldsymbol{x}}_2, \tilde{\boldsymbol{y}}_2)\| \\
&= \|(\tilde{\boldsymbol{m}}_1, \tilde{\boldsymbol{x}}_1, \tilde{\boldsymbol{y}}_1)\|_* + \|(\tilde{\boldsymbol{m}}_2, \tilde{\boldsymbol{x}}_2, \tilde{\boldsymbol{y}}_2)\|_*.
\end{aligned}$$

We have verified $\|\cdot\|_*$ is exactly a norm.

By the equivalence of norms, there exists $\tilde{C} > 0$ such that
$$\|(\tilde{\boldsymbol{m}}, \tilde{\boldsymbol{x}}, \tilde{\boldsymbol{y}})\|_* \leq \tilde{C}\|(\tilde{\boldsymbol{m}}, \tilde{\boldsymbol{x}}, \tilde{\boldsymbol{y}})\|. \tag{40}$$

Combining (38), (39) and (40), we have
$$\|\tilde{\boldsymbol{m}}^{(n+1)}\| = \|\tilde{\boldsymbol{m}}^{(n+1)} - \tilde{\boldsymbol{m}}^*\| \leq \tilde{C}\|(\tilde{\boldsymbol{m}}_n - \tilde{\boldsymbol{m}}^*, \boldsymbol{x}_n - \boldsymbol{x}^*, \boldsymbol{y}_n - \boldsymbol{y}^*)\| \leq \tilde{C}\|\boldsymbol{z}_n - \boldsymbol{z}^*\|,$$
since $\tilde{\boldsymbol{m}}^* = \boldsymbol{0}$ according to Lemma C.5.

Putting all the above facts together, we have
$$\text{Term 1} \leq \frac{2h^2 C_1 \tilde{C}^2 \|\boldsymbol{z}_n - \boldsymbol{z}^*\|^2}{\epsilon} Q^{2n+2} + \frac{2h^2 C_2 \tilde{C}^2 \|\boldsymbol{z}_n - \boldsymbol{z}^*\|^2}{(1-\beta)^2} \rho^{2n+2}$$

$$\leq \left( \frac{2h^2 C_1 \tilde{C}^2 \|\boldsymbol{z}_n - \boldsymbol{z}^*\|^2}{\epsilon} + \frac{2h^2 C_2 \tilde{C}^2 \|\boldsymbol{z}_n - \boldsymbol{z}^*\|^2}{(1-\beta)^2} \right) Q^{2n+2}$$

$$= D_1 Q^{2n+2} \|\boldsymbol{z}_n - \boldsymbol{z}^*\|^2,$$

where
$$D_1 = \frac{2h^2 C_1 \tilde{C}^2}{\epsilon} + \frac{2h^2 C_2 \tilde{C}^2}{(1-\beta)^2}.$$

As for Term 2, by a similar argument of Term 1, we have
$$\text{Term 2} \leq D_2 Q^{2n+2} \|\boldsymbol{z}_n - \boldsymbol{z}^*\|^2.$$

Eventually, we can get $\|\mathcal{R}(n, \boldsymbol{z}_n)\| = \mathcal{O}(Q^n \|\boldsymbol{z}_n - \boldsymbol{z}^*\|)$, where $Q < 1$. $\qquad\square$

Lastly, putting Lemma C.10, Lemma C.11 together with Lemma C.9, we can get Adam-DA converges locally with an exponential rate and this completes the proof of Theorem 4.4.

## C.4 FUTURE DETAILS OF FIGURE 2.

The details of Figure 2 are:

- For the Adam-DA and Continuous Adam-DA, the test function is defined as

$$\min_x \max_y 0.2x^2 - xy + 0.2y^2,$$

  which is a convex-concave function. We fix $\epsilon = 10^{-3}$.

- For Adam in minimization problem, the test function is defined as

$$\min_{(x,y)} x^2 + y^2$$

  which is a convex function. We fix $\epsilon = 10^{-4}$.

In Figure 7, we present how the range of $h$ changes with $\epsilon$; the x-axis represents $\epsilon$ and the y-axis represents $h$. We can observe that larger $\epsilon$ values expand the convergence range for all three methods, supporting Theorem 4.3 and 4.4.

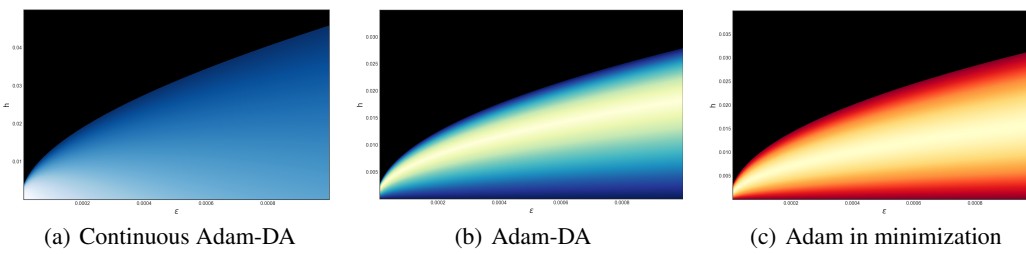

(a) Continuous Adam-DA      (b) Adam-DA      (c) Adam in minimization

Figure 7: *Effect of $\epsilon$.*

# D    ADDITIONAL MATERIALS FOR SECTION 5

## D.1    EVOLUTION OF INCEPTION SCORES

In Figure 8, we present the evolution of the inception scores for models training by different parameters of algorithms as reported in Figure 4, Table 1 and Table 2.

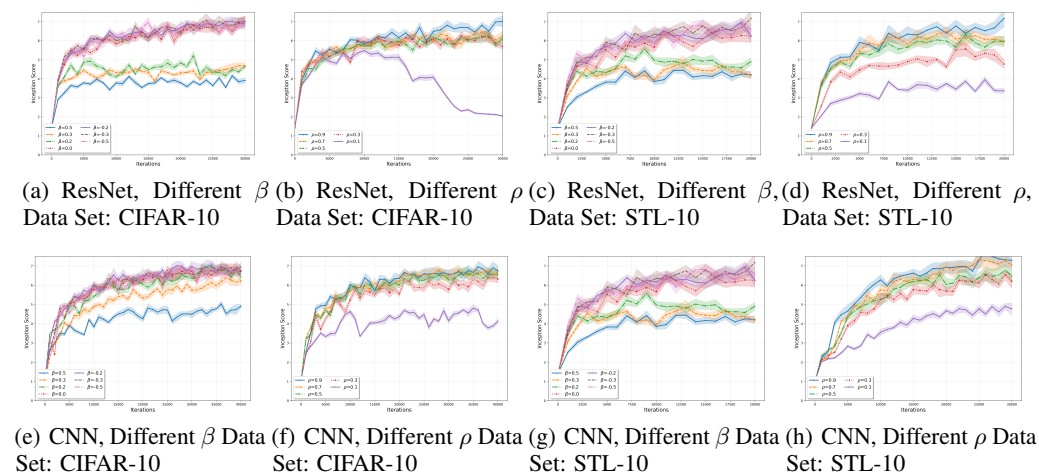

(a) ResNet, Different $\beta$ Data Set: CIFAR-10

(b) ResNet, Different $\rho$ Data Set: CIFAR-10

(c) ResNet, Different $\beta$, Data Set: STL-10

(d) ResNet, Different $\rho$, Data Set: STL-10

(e) CNN, Different $\beta$ Data Set: CIFAR-10

(f) CNN, Different $\rho$ Data Set: CIFAR-10

(g) CNN, Different $\beta$ Data Set: STL-10

(h) CNN, Different $\rho$ Data Set: STL-10

Figure 8: *Inception Score for the corresponding experimental settings in Figure 4. A high score implies better training performance. Training algorithms with lower $\ell_1$ gradient norms also have better performance.*

Interestingly, by comparing Figure 8 with Figure 4 (reproduced below), we observe that the evolutionary trend of the inception score closely aligns with that of the cumulative average gradient norm: as the cumulative average gradient norm decreases, the inception score increases. This finding supports the existence of a strong correlation between the training performance under a given algorithm and the flatness of the training trajectories associated with that algorithm.

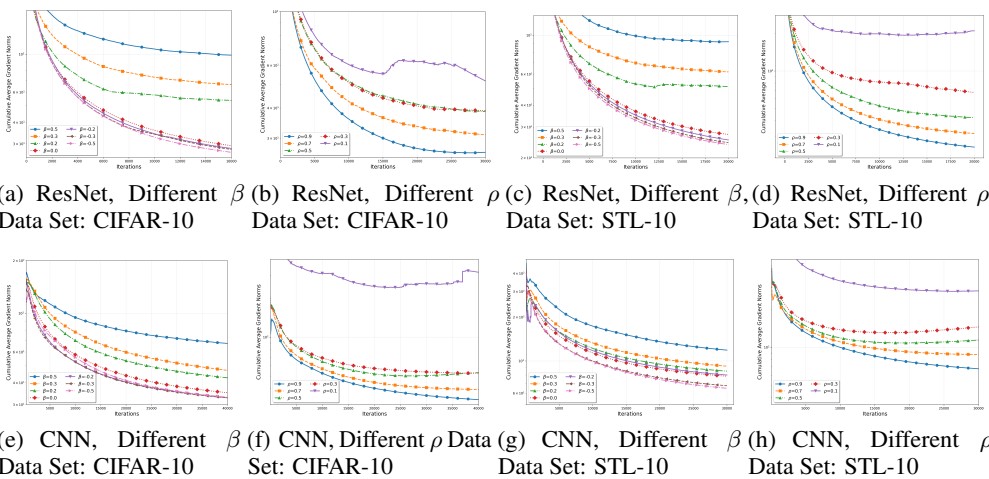

(a) ResNet, Different $\beta$ Data Set: CIFAR-10

(b) ResNet, Different $\rho$ Data Set: CIFAR-10

(c) ResNet, Different $\beta$, Data Set: STL-10

(d) ResNet, Different $\rho$, Data Set: STL-10

(e) CNN, Different $\beta$ Data Set: CIFAR-10

(f) CNN, Different $\rho$ Data Set: CIFAR-10

(g) CNN, Different $\beta$ Data Set: STL-10

(h) CNN, Different $\rho$ Data Set: STL-10

## D.2 Evolution of Gradient Norms

In the main text, we use cumulative average gradient norms to improve the visualization; here, we present the evolution of gradient norms in each iteration.

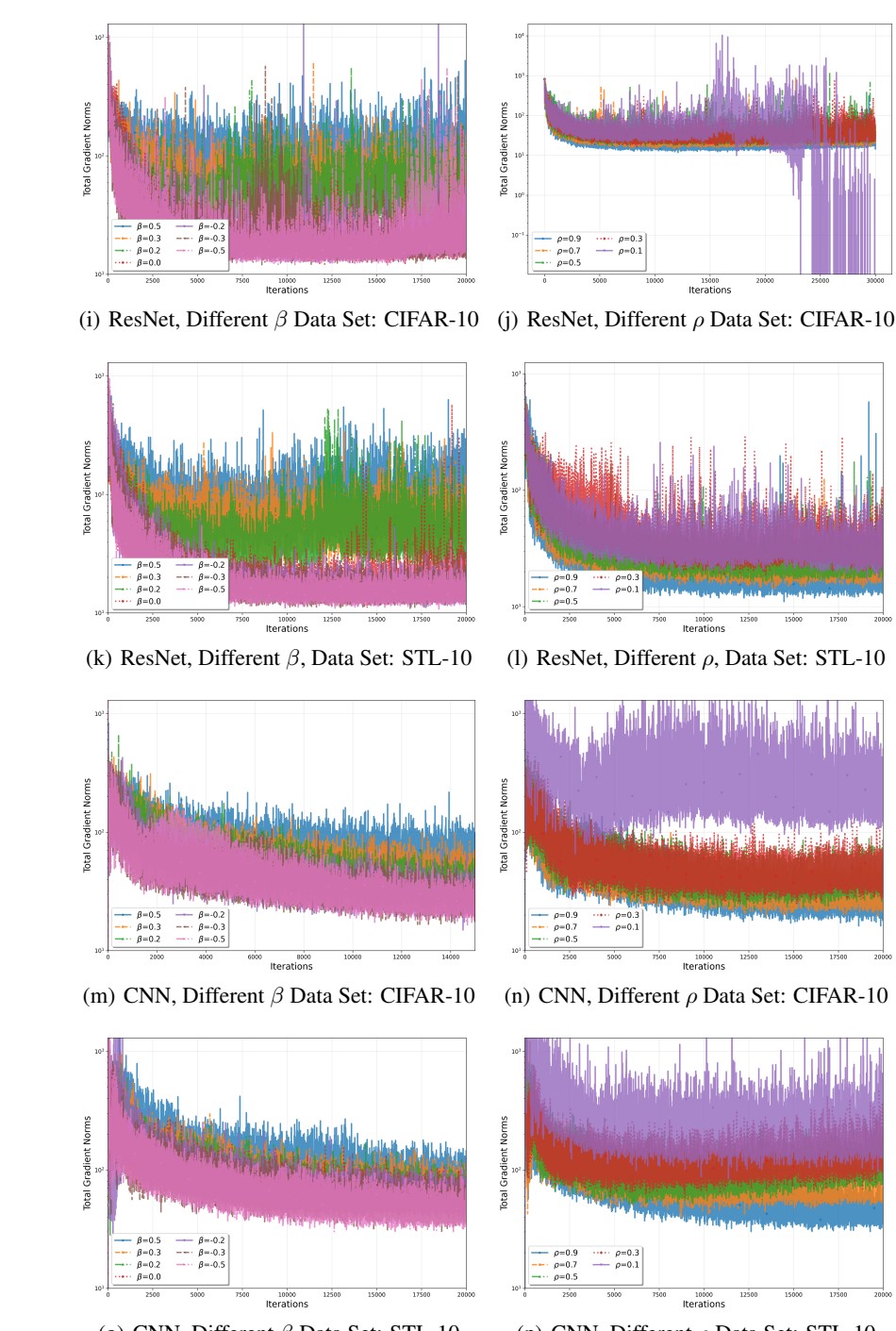

(i) ResNet, Different $\beta$ Data Set: CIFAR-10 (j) ResNet, Different $\rho$ Data Set: CIFAR-10

(k) ResNet, Different $\beta$, Data Set: STL-10 (l) ResNet, Different $\rho$, Data Set: STL-10

(m) CNN, Different $\beta$ Data Set: CIFAR-10 (n) CNN, Different $\rho$ Data Set: CIFAR-10

(o) CNN, Different $\beta$ Data Set: STL-10 (p) CNN, Different $\rho$ Data Set: STL-10

### D.3 SAMPLE IMAGES

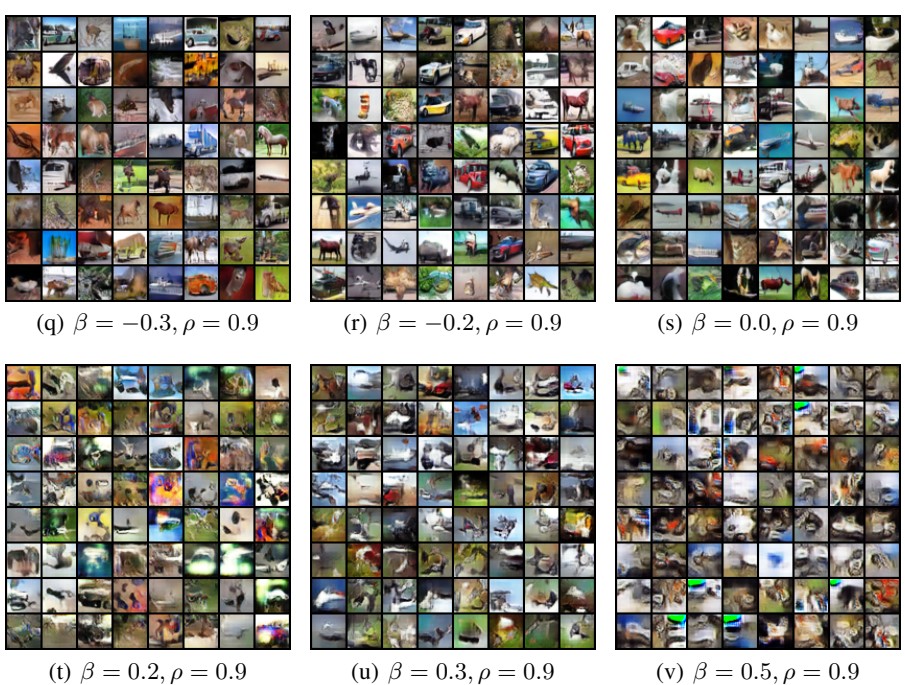

(q) $\beta = -0.3, \rho = 0.9$    (r) $\beta = -0.2, \rho = 0.9$    (s) $\beta = 0.0, \rho = 0.9$

(t) $\beta = 0.2, \rho = 0.9$    (u) $\beta = 0.3, \rho = 0.9$    (v) $\beta = 0.5, \rho = 0.9$

Figure 9: *Sample images for different $\beta$. Architecture: ResNet. Data Set: CIFAR-10.*

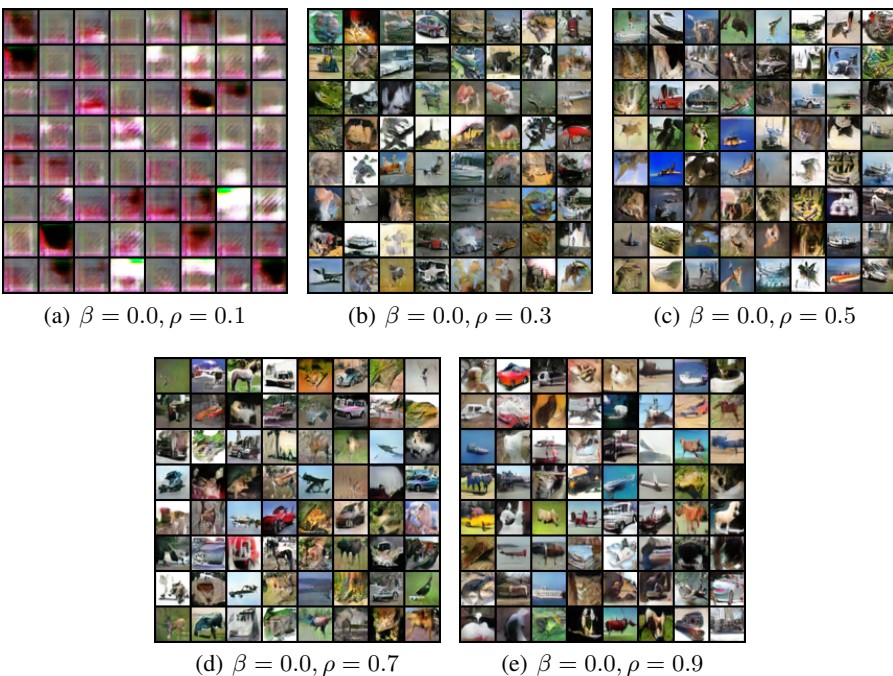

(a) $\beta = 0.0, \rho = 0.1$    (b) $\beta = 0.0, \rho = 0.3$    (c) $\beta = 0.0, \rho = 0.5$

(d) $\beta = 0.0, \rho = 0.7$    (e) $\beta = 0.0, \rho = 0.9$

Figure 10: *Sample images for different $\rho$. Architecture: ResNet. Data Set: CIFAR-10.*

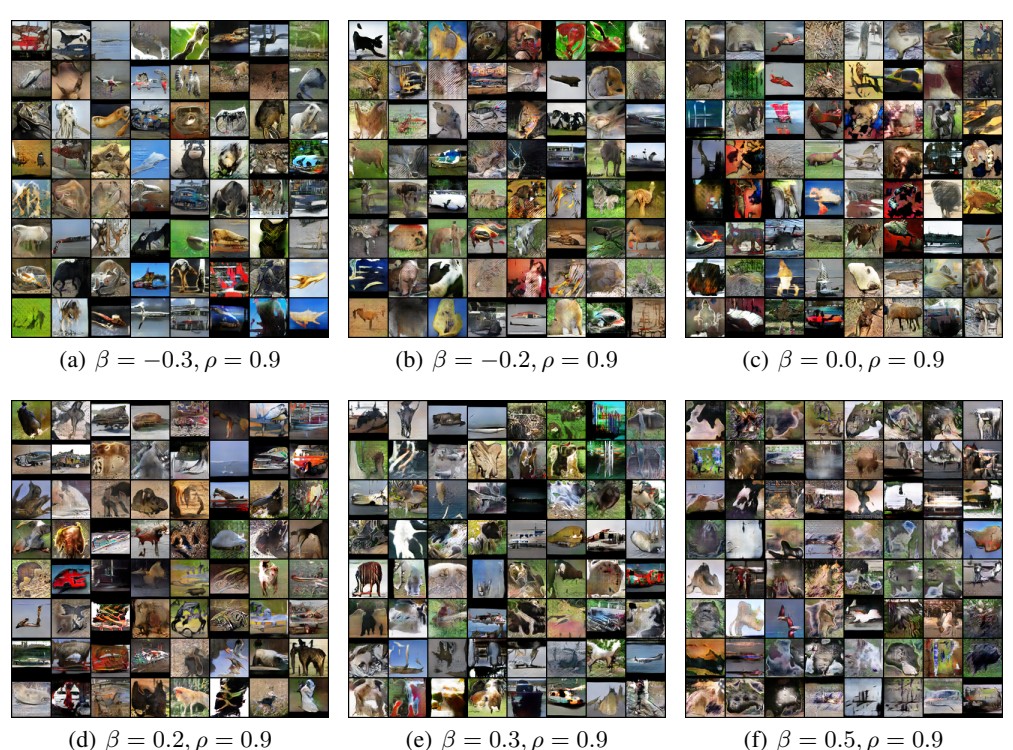

(a) $\beta = -0.3, \rho = 0.9$     (b) $\beta = -0.2, \rho = 0.9$     (c) $\beta = 0.0, \rho = 0.9$

(d) $\beta = 0.2, \rho = 0.9$     (e) $\beta = 0.3, \rho = 0.9$     (f) $\beta = 0.5, \rho = 0.9$

Figure 11: *Sample images for different $\beta$. Architecture: CNN. Data Set: STL-10.*

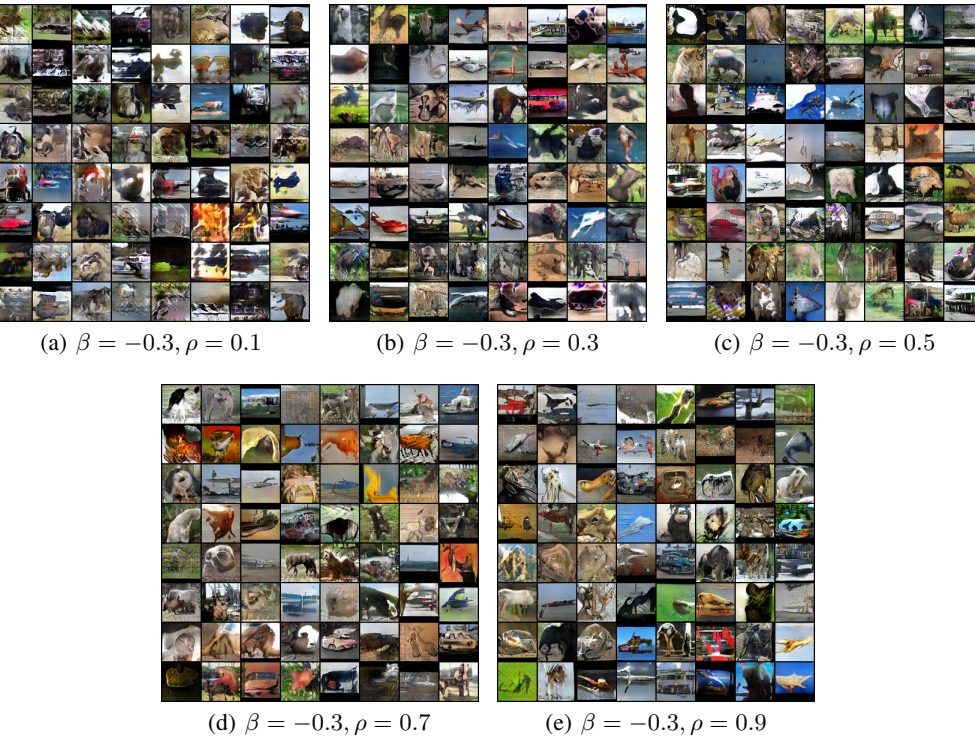

(a) $\beta = -0.3, \rho = 0.1$     (b) $\beta = -0.3, \rho = 0.3$     (c) $\beta = -0.3, \rho = 0.5$

(d) $\beta = -0.3, \rho = 0.7$     (e) $\beta = -0.3, \rho = 0.9$

Figure 12: *Sample images for different $\rho$. Architecture: CNN. Data Set: STL-10.*

