# OpenReview forum: "Continuous-time Analysis of Adam in Min-Max Games: Local Convergence and Implicit Gradient Regularization"
_ICLR.cc/2026/Conference — Submitted to ICLR 2026_

### Official Review · Reviewer_3qbU · 2025-10-29

**Soundness:** 2
**Presentation:** 3
**Contribution:** 2
**Rating:** 2
**Confidence:** 4

**Summary:**

This paper proposes a continuous time counterpart of Adam for minmax optimization in the deterministic setting. First the paper shows that this continuous-time system approximates well Adam for minmax optimization under sufficient regularity on the objective function (Theorem 3.1).  Using this dynamical system viewpoint, the paper derives local convergence results for both continuous and discrete time systems by studying the Jacobian of the vector field at some local Nash equilibrium (Proposition 4.1) and discusses the influence of the parameters of the algorithm on the convergence behavior (Theorem 4.3-4.4) when local convergence can be shown in a suitable regime of step sizes. It is shown that the hyperparameters affect Adam in the opposite manner as for standard optimization. Simulations illustrate some of the claims of the paper.

**Strengths:**

1. The paper is clearly written and well-organized.
2. Adaptive gradient methods are relatively less explored in minmax optimization compared to their counterpart in minimization. Moreover it is already challenging to analyze Adam in mere minimization and still an active research topic given the popularity of the algorithm. The minmax setting is clearly more challenging due to both the complications due to the game setting (cycling behavior …) and the more involved Adam algorithm.
3. The continuous time approach is an interesting approach in general to get some insights about an algorithm. While this is not new, it is interesting to see that the Jacobian for Adam can be related to that of GDA under some assumptions (Proposition 4.1).

**Weaknesses:**

1. **Difference between optimization and min-max optimization** ‘Adam diverges even for bilinear objectives, regardless of parameter choices (Corollary 4.7). These findings highlight fundamental differences in Adam’s dynamics between min-max games and minimization.’ I think it should be clearly noted that even GDA diverges for min-max games when using the same step sizes. The paper highlights a lot the contrast between minimization and minmax for Adam. It is well known that the behavior of GD for minimization and GDA for min-max are different. Unlike what the paper emphasizes, it is hence not surprising in my opinion that this is also the case for Adam between optimization and minmax. One of the main differences is the use of momentum which might give some hope. Well, it has been shown that negative momentum can help (e.g. Gidel et al. 2019) which this paper seems to confirm. The second momentum (which appears additionally in Adam) is shown to be not so relevant in the deterministic setting for local convergence.

2. The paper seems like an immediate follow-up to the paper of Feng et al. 2025 about continuous-time heavy ball momentum for minmax games, which adopts a similar approach and has similar local convergence results. The difference with Adam is the use of adaptive step sizes but some of the assumptions (bounded gradients for instance, see below) imply that these have somehow little effect. There are some new technical treatments, the main one being probably Lemma B.10. But I think that technical novelty is limited overall. Further discussion and more detailed comparison with that paper would be relevant.

3. **Missing related work**: the discussion is missing a number of relevant works analyzing the local convergence of GDA for minmax and even some results for adaptive methods (similar to Adam) for minmax:

- Several works in the literature have conducted local stability analysis for variants of GDA for minmax optimization with refined local convergence results and even characterization of strict local minmax equilibria:

Chi Jin, Praneeth Netrapalli, and Michael I Jordan. What is local optimality in nonconvex-
nonconcave minimax optimization? ICML 2020.

Fiez, Tanner, et al. "Global convergence to local minmax equilibrium in classes of nonconvex zero-sum games.”NeurIPS 2021.

Fiez, Tanner, and Lillian J. Ratliff. Local convergence analysis of gradient descent ascent with finite timescale separation. ICLR 2021.

Hsieh, Ya-Ping, Panayotis Mertikopoulos, and Volkan Cevher. The limits of min-max optimization algorithms: Convergence to spurious non-critical sets. ICML 2021.

- Even adaptive algorithms for minmax optimization (similar to Adam) have been proposed and analyzed with non-asymptotic guarantees, these are based on adaptively learning a time scale separation (i.e. different step sizes for both variables) which has been shown to lead to convergence (see works above for results for GDA without adaptive steps):

X. Li, J. Yang, and N. He. ‘Tiada: A time-scale adaptive algorithm for nonconvex minimax optimization.’ ICLR 2023.

J. Yang, X. Li, and N. He. Nest your adaptive algorithm for parameter-agnostic nonconvex minimax optimization. NeurIPS 2022.

- The continuous time analysis of adaptive gradient methods (including Adam) for (stochastic) optimization has been initiated in a number of prior works in the literature, preceding even Ma et al. 2022 (which actually acknowledges that in their paper as they reuse the previously proposed dynamical systems):

  Da Silva, A. B., & Gazeau, M. (2020). A general system of differential equations to model first-order adaptive algorithms. Journal of Machine Learning Research, 21(129), 1-42.

  Barakat, A., & Bianchi, P. (2021). Convergence and dynamical behavior of the ADAM algorithm for nonconvex stochastic optimization. SIAM Journal on Optimization, 31(1), 244-274.

Barakat, A., Bianchi, P., Hachem, W., & Schechtman, S. (2021). Stochastic optimization with momentum: convergence, fluctuations, and traps avoidance. Electronic Journal of Statistics, 15(2), 3892-3947.

Gadat, S., & Gavra, I. (2022). Asymptotic study of stochastic adaptive algorithms in non-convex landscape. Journal of Machine Learning Research, 23(228), 1-54.

Not also that the continuous time analysis has a much older tradition in the literature, including in the stochastic setting (see e.g.  Borkar 2008 ‘Stochastic Approximation: A Dynamical Systems Viewpoint’ and the references therein, see also e.g. Benaim 1996 ‘A dynamical system approach to stochastic approximations’, SIAM Journal on Control and Optimization, to name a few references among others).

4. **Assumption 4.2:** While this assumption is used in some prior works cited by the paper, it does not seem to be used in the local convergence analysis of GDA (see all references above). One of the conclusions of the paper regarding the influence of the momentum parameter $\beta$ is also intimately related to this assumption. Indeed, if the antisymmetric part (which is responsible for oscillations) is dominant, it is not so surprising that (positive) momentum will then hurt as its effect will amplify past information influence, and negative momentum might help.

5. **Using prior work for more refined results?** Since the Jacobian of Adam and of GDA are related (Proposition 4.1) and since the Jacobian of GDA has been thoroughly studied (e.g. in the works above), I expect that much more refined results can be established under weaker assumptions (even necessary and sufficient conditions of stability, using time-scale separation and characterization of strict local minmax equilibria) than the ones made in this work (e.g. relaxing Assumption 4.2. which seems quite restrictive).

6. **Local Nash:** It is known that even for some simple functions, local or global Nash equilibria may not exist in nonconvex-nonconcave games (see e.g. Jin et al. 2020, Proposition 6). As $(x^\star, y^star)$ is defined as a local Nash and Theorems 4.3-4.4 show local convergence to that NE, it seems that the results implicitly assume existence of local NE. It would be worth giving precisions on what are the assumptions to guarantee their existence or give concrete examples of functions for which the statement is not vacuous.

7. **Restriction to deterministic setting:** The paper is focused on the deterministic setting which is restrictive in practice given the common use of stochastic gradients. While I understand that the analysis of the deterministic can be a first step, I think that similar conclusions will hardly follow in the stochastic setting. For instance, corollary 4.5 argues that the second-order momentum hyperparameter might not have a role ‘in the presence or absence of local convergence’ for Adam, I think its role will be important in the stochastic setting to adapt the step size and to control noise, perhaps also for variance reduction.

8. **Experiments:** Theoretical results (Theorems 4.3-4.4) are about local convergence to local Nash equilibria and are hence concerned with function values around local Nash equilibria, but the experiments are about average gradient norms. It is not clear how experiments validate or support the theoretical claims, especially in nonconvex-nonconcave settings such as in the experiments.

9. **Theorem 3.1:** The assumption of bounded derivatives excludes even simple quadratic objectives. In addition, under this assumption, the $v_n$ sequences are bounded which means that the adaptive (effective) step sizes are upper and lower bounded, and the behavior is closer to a simple momentum algorithm.

**Typos/Minor comments:**

- Proposition 2.1: This is a classical textbook result in dynamical systems, I do not think it is appropriate to cite some 2021 paper for this.
- Lemma B.3 is an elementary result in linear algebra, again I do not think a citation is required.
- In the proofs, Lemma B.8  is a well-known and standard result in the literature (e.g. in control) but not presented as so, see discrete time Lyapunov equation for Schur stable matrices. I do not think a 2 page proof is needed here, or at least it should be clearly said that this is a well-known result and that a proof is provided for completeness.
- l. 116-117: ‘a local Nash equilibrium. This concept is the standard solution concept in min-max games and plays the same role as a local minimum in minimization problems.’ I do not think it is that standard, see the reference above Jin et al. 2020 precisely about the discussion of which relevant solution concept should be the focus and the limitations of local Nash.

- l. 214: ‘In section 4.1 describes’
- l. 260: ‘We are now present our’
- l. 122, l. 127: should be $m_n$s rather than $v_n$s
- l. 137: spectrum and real part notation used before their definitions.
- l. 348: affect?

**Questions:**

1. **Assumption 4.2:** What’s the exact mathematical definition of $\mathcal{A} >> \mathcal{S}$? Can you provide concrete examples where this assumption is satisfied? Is it satisfied for bilinear games? It seems that it is not needed for GDA in prior work about local convergence, why is it needed here? Does it simplify the analysis? Which part of the algorithm or its continuous time analysis pushes to consider this assumption?

2.  **Critical points of Adam DA:** A first important step in analyzing the dynamical system ‘continuous Adam DA’ is to study its equilibrium (or critical) points. Are there other points than the points for which both gradients (w.r.t. x, y) are zero (i.e. equilibrium points of continuous GDA)? In that case it means that the results only guarantee local convergence to points which are equilibrium points of the GDA dynamics.

3. **Proof of Theorem 4.4:** Why don’t you end up having a similar Jacobian to analyze as for the continuous setting (Theorem 4.3) so that you do not have to deal with all the derivations in Lemma B. 10 in particular? While the discrete version induces some discretization errors (plus non-stationarity terms to control), the analysis should be similar for the main leading terms intuitively.

4. **Local Nash vs vanishing gradients:** Some Hessian assumptions are needed and made in the appendix (Lemma B. 5) to recover local Nash from vanishing gradients. Are these stated in the main part in the statement of the results?

---

> ### Author Response · Authors · 2025-11-21
> **Official Comment by Authors [1/5]**
>
> Thank you for your detailed review and acknowledging that the analysis on Adam in minmax is a challenging task (Strength 2). We sincerely appreciate reviewer's valuable comments.
>
> Regarding the concerns, we are inclined to believe that there might exist misunderstandings on the main results and their applications/motivations.  Please allow us to clarify a bit on the GOAL of this paper:
>
> ***We aim to understand the impact of parameters for Adam in min-max games, regarding local convergence and implicit regularization, in order to guide the  parameter selection in practice.***
>
> To make it concrete, please allow us to summarize the take-away information of all theorems in the paper.
>
> **Theorem 3.1:** Trajectories of Continuous Adam-DA discrete Adam-DA are close to each other, which establishes the theoretical foundation of our methodology.
>
> **Theorem 4.3:** Continuous Adam-DA converges under some assumption.
>
> **Theorem 4.4:** Discrete Adam-DA converges (as we desired)
>
> Aboves are theoretical results. But the following corollaries translate these theoretical results into actionable insight, indicating how the theory informs practical parameter choosing.
>
> **Corollary 4.5:** $\rho$ is **useless** in guaranteeing the local convergence of Adam-DA.
>
> **Corollary 4.6:** $\beta$ is related to **robustness** of Adam-DA, in the sense that smaller $\beta$ allows Adam-DA to bear a larger range of learning rates.
>
> **TL;DR:** Want to choose a larger learning rate $h$? Then better to tune down $\beta$.
>
>
> Corollary 4.7 is a side result that can be derived immediately, it might have independent interest to learning dynamics in games theory.
>
> **Section 5** concerns the parameter tuning beyond convergence, i.e., the quality of outputs generated by GANs.
>
> Simply speaking, although $\rho$ is useless in local convergence, it is crucial in improving the image quality generated by GANs. Larger $\rho$ and smaller $\beta$ help find flatter regions in landscapes and produce better outputs.
>
> **Experiments:** In the end, to what extend, these results on parameter selection can guide a better minmax learning? We use the parameter selection scheme derived from continous and deterministic Adam-DA to **Stochastic Adam-DA** in GANs training. The experimental results show that such parameter selection scheme actually work well on Stochastic Adam-DA.
>
> In summary, the focus of this paper is to investigate the differences between Adam dynamics in minimization and minmax problems, in order to draw the followings:
>
> 1. Corollary 4.5, 4.6 and Section 5, especially the uselessness of $\rho$ in local convergence yet importance in implicit regularization (Cor 4.5 and Sec 5) for min-max games cannot be referred by combining existing papers or "foreseen" by intuition.
> 2. The parameter selection scheme from Cor 4.5,4.6 and Sec 5 are practically useful in real stochastic Adam-DA. The experiments are designed to support this scheme.
>
> ---
>
> In the following, we will provide point-wise responses to your concerns in the weaknesses and questions part of the review.
>
>
> **Weakness Part:**
> ---
>
> 1. Difference between optimization and min-max optimization:
>
>
> >I think it should be clearly noted that even GDA diverges for min-max games when using the same step sizes. It is well known that the behavior of GD for minimization and GDA for min-max are different. ... for local convergence.
>
> We have added a note on the divergence of GDA in the bilinear case below Corollary 4.7 in the revised version.
>
> **The distinction between the convergence properties of Adam in minimization and in min–max games established in our work extends far beyond the bilinear case.** The bilinear setting appears only captures only a small part of the broader differences we identify between minimization and min–max games. For example, Theorems 4.3 and 4.4 establish a quantitative trade-off between the step size $h$ and the momentum parameter $\beta$ in Adam for very general min–max games, **a phenomenon that does not occur in standard minimization**. Moreover, our experimental results in Section 5 show that, in min–max games, smaller $\beta$ and larger $\rho$ guide the trajectories of the algorithms toward flatter regions of the landscape, **which is exactly the opposite of what happens in minimization**. Together, these results go far beyond the classical separation between GDA and GD in the bilinear case.
>
> We highlight the bilinear case in Corollary 4.7 and in the summary of contribution section because it provides the most concise and transparent illustration of our findings. Moreover, even in the bilinear case, we are not aware of any prior proof showing that Adam always diverges before our work. We also do not think this result is obvious from the fact that GDA diverge in bilinear games. For example, previous studies have demonstrated that several GDA variants can converge in the bilinear and other settings, such as the alternating heavy ball momentum method with  same step sizes [9, 10].

---

> ### Author Response · Authors · 2025-11-21
> **Official Comment by Authors [2/5]**
>
> 2. Missing related work:
>
> Thank you for pointing out the missing related works. **We have added these works into "Related work" and Appendix A in the revised version**.
>
> ---
>
> 3. > The paper seems like an immediate follow-up to the paper of Feng et al. 2025 about continuous-time heavy ball momentum for minmax games, which adopts a similar approach and has similar local convergence results.
> 4. >  There are some new technical treatments, the main one being probably Lemma B.10. But I think that technical novelty is limited overall. Further discussion and more detailed comparison with that paper would be relevant.
>
> Our work focuses on understanding the **local convergence** (Section 4) and **implicit gradient regularization** (Section 5) of Adam in min–max games. The mathematical tool tools, such as Jacobian-based local analysis and flatness analysis via implicit regularization in continuous time are commonly used in the literature.
>
> **However, we would like to clarify that extending these tools from the relatively simple non-adaptive Heavy-ball momentum method in (Feng et al., 2025) to Adam is not trivial.** Adam’s adaptive structure substantially increases the complexity of both the continuous-time model and the discrete-time analysis. For instance, deriving an appropriate $\mathcal{O}(h^3)$ local error equation is significantly more involved than in the Heavy-ball case, which in turn makes the Jacobian analysis considerably more difficult.
>
> Importantly, once these tools are successfully generalized to Adam-DA, they yield **new insights that cannot be inferred** from Feng et al. (2025). In particular:
>
> * **Discrete-time vs. continuous-time behavior**: We provide  local convergence result for both the continuous-time model and the discrete-time algorithms. The results show that the continuous-time model matches the algorithms well in usual settings. But when the momentum $\beta$ is very small, this match becomes weak, which highlight the potential failure of continuous-time models that do not appear in previous works.
> * **The role of $\rho$** : Heavy-ball momentum is non-adaptive, so the influence of $\rho$ is entirely absent in Feng et al. (2025). In contrast, we show that in min–max games, larger $\rho$ amplifies Adam’s implicit regularization effect—in sharp contrast to the minimization setting, where [14] shows that smaller $\rho$ strengthens implicit regularization. This is a fundamentally new phenomenon specific to the min–max setting.
> * **The role of $\epsilon$** : Traditionally, $\epsilon$ is viewed merely as a term preventing division by zero and is assumed to be extremely small. In fact, using large $\epsilon$ is often considered inconsistent with the design philosophy of adaptive gradient methods [32]. However, Theorems 4.3 and 4.4 demonstrate that a large $\epsilon$ can enhance local convergence in min–max games—again revealing behavior entirely absent in the Heavy-ball analysis.
>
> Based on these points, we believe that our results provide new insights into the dynamics of Adam in min–max games that go well beyond the findings of Feng et al. (2025). **We have incorporated the above discussion into the Appendix A of revised version**.
>
> ---
>
> 5. > The difference with Adam is the use of adaptive step sizes but some of the assumptions (bounded gradients for instance, see below) imply that these have somehow little effect.
>
>  We would like to clarify several points on the bounded derivative assumption.
>
> * The bounded derivatives condition is standard in the literature on rigorous error analysis between continuous-time equations and discrete-time algorithms, e.g., [10, 13, 14, 15, 17, 18, 19, 20, 28], among many others. In particular, the recent work [14] (Cattaneo et al., ICML 2024) explicitly comments in the conclusion part of their work that
> *“**As far as we know, the assumption similar to (8) is explicitly or implicitly present in all previous work on backward error analysis of gradient-based machine learning algorithms**”*
> where their Assumption (8) is precisely the same as the condition used in Theorem 3.1 of our work.
> * While we assume uniformly bounded gradients, this does not diminish the role of adaptive stepsizes in Adam-DA. Bounded derivative assumption ensures gradients do not explode, but it does not imply that gradient coordinates have similar scales. In min–max games, the coordinate-wise structure can be highly anisotropic due to the interaction between minimization and maximization variables, and the second-moment accumulator still evolves very differently across coordinates as time $n$ increases. The diagonal adaptive preconditioner therefore remains far from identity, and the adaptive stepsize continues to actively shape the trajectory.
> * Moreover, the bounded-derivatives assumption is only required for the backward error analysis in Theorem 3.1, and it is not used in our local convergence result for discrete-time algorithm Adam-DA (Theorem 4.4).

---

> ### Author Response · Authors · 2025-11-21
> **Official Comment by Authors [3/5]**
>
> 6. Local Nash:
>
> >It is known that even for some simple functions, local or global Nash equilibria may not exist in nonconvex-nonconcave games ... which the statement is not vacuous.
>
> We have added a discussion on the non-existence of a local Nash equilibrium in the revised version. In this work, we study the dynamical behavior of Adam under the assumption that there exists a local Nash equilibrium $(x^*,y^*)$, which is the most interesting case from a game-theoretical viewpoint [1, 3, 7, 8, 9, 10, 18].
>
>
> ---
>
> 7. Restriction to deterministic setting:
>
> We agree that the deterministic setting is restrictive in practice. However, we'd like to emphsize that it serves as an important first step toward a theoretical understanding of stochastic case, and deterministic analyses can provide valuable insights into stochastic dynamics, especially in the research direction of implicit regularization, which is a major focus of the current work [11, 12, 17, 18, 22, 29].
>
> Regarding our results, we believe that the findings on implicit gradient regularization of Adam-DA can be extended to the stochastic case, as demonstrated in the experiments in Section 5, where we employ mini-batch Adam-DA to validate our theory. For the local convergence analysis, while the last-iterate convergence may not hold due to stochastic noise, we believe the general conclusion that smaller momentum can help to approach the equilibrium should still apply.
>
> ---
>
> 8. Experiments:
> > Theoretical results are about local convergence to local Nash equilibria ... but the experiments are about average gradient norms.
>
> **These experiments are used to support our results on implicit gradient regularization in Section 5, rather than local convergence**. Especially, we use average gradient norms to improve the visualization on how the gradient norms evolve during the training process. We also provided the gradient curves without averanging in the Appendix D.2 in the submission.
>
> ---
>
> 9. Theorem 3.1
>
> > The assumption of bounded derivatives excludes even simple quadratic objectives. In addition, under this assumption, the  sequences are bounded which means that the adaptive (effective) step sizes are upper and lower bounded, and the behavior is closer to a simple momentum algorithm.
>
> Please refer to our respond to point **5** in **Weakness Part**.
>
> ---
>
> 10. Typos/Minor comments:
>
> We thank the review for pointing out the typos in the submission, we have fixed them in the revised version. Especially, we
> * Change the citation of Proposition 2.1; add a comment on Lemma B.8 to make it clear that this lemma can be derived from standard matrix calculations; remove the description of local Nash, and state its non-existence in certain cases.
> * Fixed other typos.
>
> ---
>
> **Questions Part:**
> ---
> 1. Questions regarding to Assumption 4.2:
>
> >  What’s the exact mathematical definition of $\mathcal{A} \gg \mathcal{S}$?
>
> We use the notation $\mathcal{A} \gg \mathcal{S}$, following [3,10], to indicate that the antisymmetric part of the Jacobian ($\mathcal{A}$) dominates its symmetric part ($\mathcal{S}$) in magnitude. A direct mathematical implication of $\mathcal{A} \gg \mathcal{S}$ is that the absolute value of the imaginary part of the Jacobian’s eigenvalues, $|\Im(\lambda)|$, exceeds the absolute value of the real part, $|\Re(\lambda)|$, which is precisely the condition used in the proofs of Theorems 4.3 and 4.4.
>
> **To avoid potential confusion, we have reformulated Assumption 4.2 in the revised version.**
>
> ---
>
> > Can you provide concrete examples where this assumption is satisfied? Is it satisfied for bilinear games?
>
> Yes, bilinear games satisfy the assumption $\mathcal{A} \gg \mathcal{S}$, since in the bilinear case we have $\mathcal{S} = 0$, while $\mathcal{A}$ corresponds to a block anti-diagonal matrix composed of the payoff matrix. In this case, the eigenvalues of the Jacobian are purely imaginary.
>
> Another class of games that satisfy this assumption is GANs, where the imaginary parts of the eigenvalues of the Jacobian associated with the GDA dynamics are known to be much larger than their real parts [2]. Further examples are perturbed bilinear games $f(x,y) =x^{\top}Ay + \mu_1 \mathcal{H}_1(x) - \mu_2 \mathcal{H}_1(y),$ where $\mu_1,\mu_2 \ge 0$ are small constants and  $\mathcal{H}_1, \mathcal{H}_2$ are strongly-convex regularizers. This class of games has also been widely studied in the literature [21, 23].

---

> ### Author Response · Authors · 2025-11-21
> **Official Comment by Authors [4/5]**
>
> (Continuous of question 1 in **Question Part**)
>
> In the following, we respond to several concerns regarding the restrictiveness and usefulness of Assumption 4.2 raised in the review.
>
> > It seems that it is not needed for GDA in prior work about local convergence, why is it needed here? Does it simplify the analysis? Which part of the algorithm or its continuous time analysis pushes to consider this assumption?
>
> > While this assumption is used in some prior works cited by the paper, it does not seem to be used in the local convergence analysis of GDA ... and negative momentum might help. *（From the weakness part）*
>
> >  Since the Jacobian of Adam and of GDA ...   than the ones made in this work (e.g. relaxing Assumption 4.2. which seems quite restrictive). *（From the weakness part）*
>
> We choose to consider games satisfy "$\mathcal{A} \gg \mathcal{S}$" based on the following two reasons:
>
> 1. $\mathcal{A} \gg \mathcal{S}$ characterizes the most interesting and typical min–max games that **distinguish them from minimization** (with further explanation provided below).
>
> 2. Our motivation in this work is to **understand aspects in which the dynamics of Adam in min–max games fundamentally differ from those in minimization, which can help with practical parameter tuning**. It is therefore natural for us to focus on games that are most likely to reveal such differences—namely, games characterized by $\mathcal{A} \gg \mathcal{S}$. We agree that the condition $\mathcal{A} \gg \mathcal{S}$ is not necessary for the convergence of Adam in min-max games. For example, if the game has the formulation $\min_x \max_y f(x) - g(y)$, which is essentially a minimization problem disguised as a min–max game, then standard convergence results for Adam in minimization directly apply. However, these settings are different from our motivation.
>
>
> **More explanation of reason 1:** The most fundamental characteristic that distinguishes min–max games from standard minimization problems is that, in min–max games, the dynamics of the players are coupled and interact with each other, whereas in minimization, each player independently optimizes their own objective without interaction. From a local analysis perspective, the interaction between players is precisely captured by the matrix $\mathcal{A}$, while the independent component is represented by the matrix $\mathcal{S}$. This viewpoint has been widely adopted in the literature on game dynamics, e.g., [24, 25, 26, 27]. Therefore, the assumption $\mathcal{A} \gg \mathcal{S}$ characterizes the class of min–max games where the interaction between players dominates the dynamics, which is the most interesting and representative case from a game-theoretical perspective.
>
> **Use of $\mathcal{A} \gg \mathcal{S}$ in the literature:** The assumption $\mathcal{A} \gg \mathcal{S}$ is also widely adopted in prior works to study aspects of game dynamics that differ from minimization. For example:
> * Section 3 of [3] assumes this condition to prove that the convergence rate of GDA typically depends on the average eigenvalues of $\mathcal{S}$—a behavior that does not arise in minimization.
> * Section 3 of [1] shows that a large $\mathcal{A}$ relative to $\mathcal{S}$ is necessary for the accelerated local convergence of Optimistic GDA, whereas such acceleration does not appear in minimization.
> * Section 4 of [10] uses the assumption to establish the local convergence of heavy-ball momentum in min–max games, thereby revealing a fundamental difference in the role of momentum between min–max games and minimization.
>
> In addition, this assumption also includes previous studies of game dynamics in the bilinear setting and its perturbations—arguably the most widely studied class of min–max games (e.g., [4, 5, 6, 21, 23], among many others).
>
> **Use of $\mathcal{A} \gg \mathcal{S}$ in current work:** As stated above, the main motivation of the current work is to find aspects that Adam in min-max games are different from minimization. Thus consider min-max games satisfy $\mathcal{A} \gg \mathcal{S}$ is nature for us. Specifically, in Theorem 4.4, we show that the continuous-time Adam-DA converges locally if
> \begin{align*}
>     0< h < \min_{\lambda \in \mathrm{Sp}(\mathcal{J})}\frac{2\sqrt{\epsilon} (1-\beta)\lvert \Re (\lambda) \lvert}{(1+\beta)(\Im(\lambda)^2 - \Re(\lambda)^2)},
> \end{align*}
>
> Assumption 4.2 is necessary to ensure that the denominator term $(\Im(\lambda)^2 - \Re(\lambda)^2)$ remains positive so that the bound above is non-empty. The same condition is also required to guarantee the local convergence of the discrete-time Adam-DA.

---

> ### Author Response · Authors · 2025-11-21
> **Official Comment by Authors [5/5]**
>
> 2. Critical points of Adam DA:
>
> > Are there other points than the points for which both gradients (w.r.t. x, y) are zero (i.e. equilibrium points of continuous GDA)? In that case it means that the results only guarantee local convergence to points which are equilibrium points of the GDA dynamics.
>
> In Theorem 4.3 and 4.4, we consider the setting where the trajectories are close to a local Nash equilibrium, which is the standard setting in the study of local convergence of game dynamics [1, 2, 3, 7, 10].
>
> It is a general fact that certain stable points in game dynamics are not local Nash equilibria [7]. Techniques such as proposed in [8] are designed to avoid such undesirable stationary points. Combining Adam-DA with these approaches presents an interesting direction for future research.
>
> ---
>
> 3. Proof of Theorem 4.4:
>
> > Why don’t you end up having a similar Jacobian to analyze as for the continuous setting (Theorem 4.3) so that you do not have to deal with all the derivations in Lemma B. 10 in particular?
>
> Thank you for pointing out this potential approach.
>
> We choose to directly analyze the Jacobian of the discrete-time Adam-DA to derive an exact formula for the trade-off between step size and momentum. This approach ensures that the bound in Theorem 4.4 serves as a necessary and sufficient condition for local convergence. Such a precise characterization is crucial for comparing the role of momentum in min–max games and standard minimization. For example, neglecting higher-order terms in the Jacobian could destroy the monotonicity decreasing of the upper bound as a function of $\beta$ in Theorem 4.4.
>
> ---
>
> 4. Local Nash vs vanishing gradients:
>
>  > Some Hessian assumptions are needed and made in the appendix (Lemma B. 5) to recover local Nash from vanishing gradients. Are these stated in the main part in the statement of the results?
>
> The Hessian condition in Lemma B.5 corresponds to the second-order optimality condition for a local Nash equilibrium. In the statements of our results, we have clearly specified that the point $(x^*,y^*)$ under consideration is assumed to be a local Nash equilibrium.

---

> ### Author Response · Authors · 2025-11-21
> **Reference**
>
> **Reference**:
>
> [1] Liang & Stokes, Interaction Matters: A Note on Non-asymptotic Local Convergence of Generative Adversarial Networks. AISTATS 2019
>
> [2] Mescheder et al., The Numerics of GANs. NeurIPS 2017
>
> [3] Wang & Chizat, Local Convergence of Gradient Methods for Min-Max Games: Partial Curvature Generically Suffices. NeurIPS 2023
>
> [4] Daskalakis et al., Training GANs with Optimism. ICLR 2018
>
> [5] Zhang & Yu., Convergence of Gradient Methods on Bilinear Zero-Sum Games. ICLR 2020
>
> [6] Bailey et al., Finite Regret and Cycles with Fixed Step-Size via Alternating Gradient Descent-Ascent. COLT 2020
>
> [7] Daskalakis & Panageas, The Limit Points of (Optimistic) Gradient Descent in Min-Max Optimization. NeurIPS 2018
>
> [8] Adolphs et al., Local Saddle Point Optimization: A Curvature Exploitation Approach. AISTATS 2019
>
> [9] Gidel et al., Negative momentum for improved game dynamics. AISTATS 2019
>
> [10] Feng et al., Continuous-Time Analysis of Heavy Ball Momentum in Min-Max Games. ICML 2025
>
> [11] Wang et al., Does Momentum Change the Implicit Regularization on Separable Data? NeurIPS 2022
>
> [12] Zhang et al., The Implicit Bias of Adam on Separable Data. NeurIPS 2024
>
> [13] Ghosh et al., Implicit regularization in Heavy-ball momentum accelerated stochastic gradient descent. ICLR 2023
>
> [14] Cattaneo et al., On the Implicit Bias of Adam. ICML 2024
>
> [15] Ma et al., A Qualitative Study of the Dynamic Behavior for Adaptive Gradient Algorithms. Annual Conference on Mathematical and Scientific Machine Learning, 2021
>
> [16] Barakat & Bianchi, Convergence and Dynamical Behavior of the ADAM Algorithm for Non-Convex Stochastic Optimization. SIAM Journal on Optimization Vol. 31, Iss. 1 (2021)
>
> [17] Barrett & Dherin, Implicit Gradient Regularization. ICLR 2021
>
> [18] Rosca et al., Discretization Drift in Two-Player Games. ICML 2021
>
> [19] Kovachki & Stuart, Continuous Time Analysis of Momentum Methods. Journal of Machine Learning Research 21 (2020) 1-40
>
> [20] Li et al., Stochastic Modified Equations and Dynamics of Stochastic Gradient Algorithms I: Mathematical Foundations. Journal of Machine Learning Research 20 (2019) 1-47
>
> [21] Liu et al., The power of regularization in solving extensive-form games. ICLR 2023
>
> [22] Gunasekar et al., Characterizing Implicit Bias in Terms of Optimization Geometry. ICML 2018
>
> [23] Cen et al., Fast Policy Extragradient Methods for Competitive Games with Entropy Regularization. NeurIPS 2021
>
> [24] Letcher et al., Differentiable Game Mechanics. Journal of Machine Learning Research 20 (2019) 1-40
>
> [25] Chasnov et al., Stability of Gradient Learning Dynamics in Continuous Games: Vector Action Spaces. CDC 2020
>
> [26] Cheung & Tao, Chaos of Learning Beyond Zero-sum and Coordination via Game Decompositions. ICLR 2021
>
> [27]  Legacci et al., A geometric decomposition of finite games: Convergence vs. recurrence under exponential weights. ICML 2024
>
> [28] Lyu et al., Heavy-Ball Momentum Method in Continuous Time and Discretization Error Analysis. NeurIPS 2025
>
> [29] Xie & Li, Implicit Bias of AdamW: ℓ∞-Norm Constrained Optimization. ICML 2024
>
> [30] Cutkosky, Better full-matrix regret via parameter-free online learning. Neurips 2020
>
> [31] Kingma & Ba. Adam: A method for stochastic optimization.  ICLR 2015
>
> [32] Zhang et al., Adam Can Converge Without Any Modification On Update Rules. NeurIPS 2022

---

### Official Review · Reviewer_5mRK · 2025-10-30

**Soundness:** 4
**Presentation:** 3
**Contribution:** 3
**Rating:** 10
**Confidence:** 4

**Summary:**

The paper studies the suitability of Adam to find a saddle point of the from $min_x max_y f(x,y)$ instead of a minimum $min_{x,y} f(x,y)$. This is by no means a sure thing. With a suitable initialization, a gradient ascent/descent procedure can do this when initialized close enough to a regular enough saddle point.  Another popular approach consists in using wildly different step sizes for the minimization and maximization components, at considerable computational cost.  In order to discuss the performance of Adam in that context, the authors define a continuous-time version of Adam (Chizat-style), argue that the continuous-time and discrete-time algorithms are not that far away from each other, and finally show that a substantial momentum does not help the convergence speed of the continuous-time version. They also argue that (contrary to what has been discussed in the case of minimization), momentum does not implicitly regularizes towards low-curvature regions of the cost function (not surprising as saddle points in games often have dramatic shapes). Empirical studies on GANs substantiate the findings.

**Strengths:**

- a thorough analysis with satisfying theory
- a good reminder that min-max problems are much trickier than optimization problems

**Weaknesses:**

- not too surprising to those who know that min-max problems are tricker than optimization.

**Questions:**

- Proposition 2.1. -- Please confirm the regularity assumptions required for this theorem. Min-max problems in games often lead to cost function with rather irregular saddle-points.

---

> ### Author Response · Authors · 2025-11-21
>
> We sincerely appreciate the reviewer's valuable comments.
>
> > Proposition 2.1. -- Please confirm the regularity assumptions required for this theorem. Min-max problems in games often lead to cost function with rather irregular saddle-points.
>
> Proposition 2.1 is the stability of equilibrium points in the dynamical systems literature. It requires that the vector field be continuously differentiable, which is satisfied under our setting.

---

### Official Review · Reviewer_z4QD · 2025-10-31

**Soundness:** 3
**Presentation:** 3
**Contribution:** 3
**Rating:** 4
**Confidence:** 5

**Summary:**

This paper provides a theoretical analysis of Adam in min-max games, revealing that its behavior is precisely the opposite of its behavior in standard minimization. Using continuous-time analysis, the authors derive a high-fidelity model for Adam Descent-Ascent (Adam-DA). The analysis shows that in min-max games, a smaller first-order momentum allows for convergence over a wider range of step-sizes, the reverse of what is optimal for minimization. Furthermore, Adam-DA provably diverges on bilinear games, unlike in minimization. The study finds that a smaller first-order momentum and a larger second-order momentum guide the optimization trajectory towards flatter regions of the loss landscape, which again is the opposite of the effect in minimization. Extensive experiments on GAN training validate these theoretical findings, showing that the identified parameter settings lead to lower gradient norms and improved performance. This work crucially corrects the misplaced intuition of transferring Adam's properties directly from minimization to adversarial settings.

**Strengths:**

1. The paper shows that Adam's momentum parameters have precisely the opposite effect in min-max games compared to minimization. It directly challenges and corrects a common assumption in the field, providing a crucial new theoretical lens for understanding and applying Adam in adversarial settings like GANs.

2. The paper provides a more accurate $O(h^3)$ approximation of Adam's discrete dynamics than prior results.  This theoretical result is complemented by extensive experiments on multiple GAN architectures and datasets, which show a clear correlation between flatter minima and improved performance.

3. Despite the complex theory, the paper presents its findings with simple and actionable guidelines (e.g., "use smaller $\beta$ and larger $rho$").  This bridges a critical gap between theory and practics.

**Weaknesses:**

1. The experimental validation is confined primarily to GANs. The paper's claims about "min-max games" are broad, but it does not demonstrate its findings on other critical applications like adversarial training, leaving the generalizability across the full problem domain uncertain.

2. The empirical comparisons are largely against the standard Adam baseline or its own ablated versions. It would be better to include more results of state-of-the-art algorithms specifically designed for min-max games, to show whether the discovered parameter settings make Adam superior to these specialized methods.

3. The theoretical analysis is presented in a deterministic setting.  However, Adam is almost exclusively used in stochastic optimization.  The paper does not address how stochastic noise, a fundamental aspect of Adam's typical use case, interacts with its conclusions on local convergence and implicit regularization, which is a significant gap.

**Questions:**

1. The theoretical analysis is in the deterministic setting.  How do you expect the presence of stochastic gradient noise, which is intrinsic to the practical use of Adam, to interact with or alter your core findings regarding the opposite roles of $\beta$ and $\rho$ in local convergence and implicit regularization?

2. To enable a more comprehensive evaluation and firmly establish the robustness of the proposed method, it is essential to provide additional experimental results that include more state-of-the-art baselines and other "min-max games" expect for GAN.

---

> ### Author Response · Authors · 2025-11-21
> **Official Comment by Authors [1/2]**
>
> We sincerely appreciate reviewer's valuable comments. Please see our itemized responses below:
>
> 1. Regarding the determinstic setting of Adam
>
>
> > The theoretical analysis is in the deterministic setting. How do you expect the presence of stochastic gradient noise, which is intrinsic to the practical use of Adam, to interact with or alter your core findings regarding the opposite roles of $\beta$ and $\rho$  in local convergence and implicit regularization?
>
> > The theoretical analysis is presented in a deterministic setting. However, Adam is almost exclusively used in stochastic optimization. The paper does not address how stochastic noise, a fundamental aspect of Adam's typical use case, interacts with its conclusions on local convergence and implicit regularization, which is a significant gap.
>
>
> We agree with the reviewer that the deterministic setting studied in the current work may limit the direct applicability of our results to practical scenarios characterized by significant stochastic noise. However, we would like to emphasize the following points:
>
> * **Deterministic analysis is a standard and meaningful first step toward understanding stochastic dynamics.** Even in minimization, many prior works analyze Adam under deterministic dynamics [3, 4, 5, 7, 8], and these results have consistently provided valuable insights into its stochastic behavior. To our knowledge, the current work is the first to establish local convergence and implicit regularization properties of Adam in interaction-dominated min–max games. Given the lack of existing theory in this setting, we believe focusing on deterministic dynamics is a natural and meaningful starting point.
>
>
> * **We believe the insights into Adam's behavior in min-max games from this work can extend to the stochastic setting**. Our reasoning is as follows: although stochasticity may alter convergence rates, the qualitative influence of algorithmic parameters is typically preserved. For instance, in the minimization setting, heavy ball momentum affects the stability of algorithms in a similar way in both deterministic and stochastic settings, even though the resulting convergence rates differ between the two settings [9,10]. Importantly, the GAN experiments in Section 5, conducted in a fully stochastic training regime, already exhibit behavior consistent with our deterministic theoretical predictions. This alignment suggests that the core insights from our deterministic analysis indeed carry over to realistic stochastic settings.
>
> ---
>
> 2. Regarding to the experimental results
>
> > The empirical comparisons are largely against the standard Adam baseline ... whether the discovered parameter settings make Adam superior to these specialized methods.
>
>
> > To enable a more comprehensive evaluation and firmly establish the robustness of the proposed method, it is essential to provide additional experimental results that include more state-of-the-art baselines and other "min-max games" expect for GAN.
>
>
> Thank you for these comments.
>
> We would like to clarify that the goal of the the experiments in Section 5 for Adam in GANs is **not** to propose a new SOTA GANs training method. Instead, the experiments are designed to validate our theoretical findings on Adam’s dynamics in interaction-dominated min–max games. The empirical advantage of using Adam with small momentum in GAN training has already been documented in several prior works, such as [11] and the recent baseline [12].  Our contribution is to provide a theoretical understanding on **aspects in which the dynamics of Adam in interaction-dominated min–max games fundamentally differ from those in minimization.**
>
> In particular, our main founding can be summarized as below:
>
> * **Local Convergence:** Smaller momentum $\beta$ can improve the stability of algorithms by allowing convergence with larger step sizes, while the parameter $\rho$ is not relevent
> * **Implicit Gradient Regularization (IGR):** Smaller $\beta$ and larger $\rho$ bias the dynamics toward flatter regions of the loss landscape
>
> These two aspects help explain the fundamental differences between Adam in interaction-dominated min–max settings and Adam in minimization. In particular, consistent with prior work on IGR [2,6,13,14], our results provide a qualitative characterization rather than a rigorous mathematical guarantee. As is standard in the IGR literature, whether a specific form of implicit regularization actually emerges must be verified through numerical experiments. To verify this qualitative behavior, we choose GANs as our experimental setting because **GAN training dynamics closely satisfy the interaction-dominated condition required for our IGR results**. We therefore believe that the current set of experiments, using Adam on multiple GAN architectures (ResNet, CNNs of various sizes) and on standard datasets (CIFAR-10 at 32×32 resolution and STL-10 at 96×96 resolution), is sufficient to support our claims about IGR.

---

> ### Author Response · Authors · 2025-11-21
> **Official Comment by Authors [2/2]**
>
> 3. Extension to Adversarial training
>
> > The experimental validation is confined primarily to GANs. The paper's claims about "min-max games" are broad, but it does not demonstrate its findings on other critical applications like adversarial training, leaving the generalizability across the full problem domain uncertain.
>
> We would like to clarify that we do not claim our results apply to all min–max games. As explained in Sections 4 and 5, our theoretical findings are specifically derived for the class of **interaction-dominated** games, a class that captures the most challenging and genuinely game-theoretic min–max problems and highlights the fundamental differences from minimization. Importantly, **adversarial training does not belong to this class**.
>
> Below, we explain why we focus on interaction-dominated games, and why adversarial training falls outside this setting.
>
> * ***Why interaction-dominated min-max games?*** As we stated in our response to Point 2, the motivation of this work is to understand **aspects in which the dynamics of Adam in min–max games fundamentally differ from those in minimization**. Such differences cannot hold for arbitrary min–max games, because minimization problems form a subset of min–max problems. For example, games of form $\min_x \max_y f(x) - g(y)$ are essentially unconstrained minimization problems in disguise. Thus, to meaningfully capture dynamics that differ from minimization, we must select a class of problems that is (i) broad enough to include important real-world examples, and (ii) structurally distinct from minimization. Prior works [15,16,17] have identified interaction-dominated games as exactly such a class. They exhibit strong coupling between players, leading to rotational or oscillatory dynamics—phenomena that do not arise in minimization.
>
> * ***Why adversarial training is not interaction-dominated?*** Mathematically, interaction-dominated condition ask for the imaginary parts of the eigenvalues of the Jacobian in the GDA dynamics should be larger in magnitude than the real parts. GAN training is well-known to satisfy this property, exhibiting large imaginary components in the Jacobian spectrum [1]. As a contrasting example, the eigenvalues of the Jacobian of $\min_x \max_y f(x) - g(y)$ are purely real, and there is no interaction at all between the $x$ and $y$ players.  **In the revised version, we added experiments measuring the Jacobian spectrum of adversarial training on CIFAR-10 dataset in Section 5 and found that the eigenvalues are almost purely real**. Consequently, adversarial training behaves much more like minimization than a coupled min–max game, andits implicit regularization properties are better explained by existing analyses of Adam in minimization (e.g., [2]) rather than by our results.
>
> Therefore, **adversarial training is not a counterexample to our theory; rather, it simply does not satisfy the assumptions required for our results**.
>
> **Reference:**
> ---
>
> [1] Mescheder et al., The Numerics of GANs. NeurIPS 2017
>
> [2] Cattaneo et al., On the Implicit Bias of Adam. ICML 2024
>
> [3] Wang et al., Does Momentum Change the Implicit Regularization on Separable Data? NeurIPS 2022
>
> [4] Zhang et al., The Implicit Bias of Adam on Separable Data. NeurIPS 2024
>
> [5] Cohen et al., Understanding Optimization in Deep Learning with Central Flows. ICLR 2025
>
> [6] Rosca et al., Discretization Drift in Two-Player Games. ICML 2021
>
> [7] Gunasekar et al., Characterizing Implicit Bias in Terms of Optimization Geometry. ICML 2018
>
> [8] Xie & Li, Implicit Bias of AdamW: ℓ∞-Norm Constrained Optimization. ICML 2024
>
> [9] Ghadimi & Feyzmahdavian & Johansson, Global convergence of the heavy-ball method for convex optimization. European control conference (ECC) 2015
>
> [10] Liu & Yuan, On almost sure convergence rates of stochastic gradient methods. COLT 2022
>
> [11] Gidel, Negative Momentum for Improved Game Dynamics. AISTATS 2019
>
> [12] Huang et al., The GAN is dead; long live the GAN! A Modern Baseline GAN. NeurIPS 2024
>
> [13] Barrett & Dherin, Implicit gradient regularization. ICLR 2021
>
> [14] Ghosh et al., Implicit regularization in Heavy-ball momentum accelerated stochastic gradient descent. ICLR 2023
>
> [15] Wang & Chizat, Local Convergence of Gradient Methods for Min-Max Games: Partial Curvature Generically Suffices. NeurIPS 2023
>
> [16] Liang & Stokes, Interaction Matters: A Note on Non-asymptotic Local Convergence of Generative Adversarial Networks. AISTATS 2019
>
> [17] Feng et al., Continuous-Time Analysis of Heavy Ball Momentum in Min-Max Games. ICML 2025

---

### Official Review · Reviewer_NpU6 · 2025-10-31

**Soundness:** 2
**Presentation:** 2
**Contribution:** 2
**Rating:** 4
**Confidence:** 3

**Summary:**

This paper investigates the role of Adam in min-max games compared to its role in standard minimization problems. The authors find that these two roles differ, challenging previous assumptions. Specifically, momentum parameters influence Adam’s behavior in min-max games in the opposite way to their effect in minimization. Numerical experiments are provided to support these claims.

**Strengths:**

1. The study raises an interesting question about the differing roles of Adam in min-max games and standard minimization problems.
2. The empirical evidence helps illustrate the observed behavioral differences.

**Weaknesses:**

1. The claim that Adam behaves differently in min-max games compared to standard minimization seems somewhat artificial and potentially misleading, as the authors compare Adam-DA with Adam rather than directly analyzing the original Adam across both settings.
2. The mathematical techniques employed are fairly standard, offering little in terms of novel methodological contribution.
3. The analysis focuses on local convergence and implicit gradient regularization, but the rationale for emphasizing these two perspectives is not clearly justified.

**Questions:**

Can the conclusions be generalized to other optimization algorithms, such as SGD or AdamW?

---

> ### Author Response · Authors · 2025-11-21
> **Official Comment by Authors [1/2]**
>
> We sincerely appreciate reviewer's valuable comments. Please see our itemized responses below:
>
> Weakness Part:
> ---
>
> 1. > The claim that Adam behaves differently in min-max games compared to standard minimization seems somewhat artificial and potentially misleading, as the authors compare Adam-DA with Adam rather than directly analyzing the original Adam across both settings.
>
> We would like to clarify that Adam-DA is the natural counterpart of Adam in the min–max setting. In particular, the relationship between Adam-DA in min–max games and Adam in minimization is directly analogous to the relationship between gradient descent–ascent (GDA) in min–max games and gradient descent (GD) in minimization.
>
> The original Adam is inherently a **descent** algorithm designed for solving minimization problem
> \begin{align*}
> \min_xf(x).
> \end{align*}
> In contrast, a min–max problem takes the form
> \begin{align*}
> \min_x\max_yf(x,y),
> \end{align*}
> where the x-player must perform **descent** to minimize the objective, while the y-player must perform **ascent** to maximize it. This naturally leads to Adam-DA, where the two updates follow descent and ascent directions respectively.
>
> Applying the original Adam directly to a min–max game without modifying the ascent step for the y-player is neither mathematically consistent nor structurally aligned with the nature of the min–max optimization problem.
>
>
> ---
>
> 2. >The mathematical techniques employed are fairly standard, offering little in terms of novel methodological contribution.
>
> The contribution of our work lies in investigating the dynamics of Adam in minimax optimization, which has been missing in existing literature. It should be pointed out that **existing analyses of Adam mainly focus on minimization, the behavior of Adam in min-max optimization problem is still poorly understood**. In this paper, we are the first to provide local convergence and implicit gradient regularization of Adam in min-max problem.
>
> Furthermore, the analysis of Adam in min-max optimization problem is technically nontrivial. Applying standard tools is not straightforward because the **Jacobian structure, coupling between moment estimates, and the interaction between ascent and descent updates** are qualitatively different from the minimization case.
>
> ---
>
> 3. > The analysis focuses on local convergence and implicit gradient regularization, but the rationale for emphasizing these two perspectives is not clearly justified.
>
> There are clear motivations to investigate these two aspects for Adam-DA in min-max games:
>
> * These two aspects are complementary to each other: local convergence takes a local viewpoint and studies the algorithms behaviors when the trajectories are close to the local Nash equilibrium, while implicit gradient regularization takes a global viewpoint to study how the algorithms trajectories interact with the flatness properties of the loss landscape. **Thus we believe combining these two aspects can provide a rather comprehensive understanding on Adam in min-max games.**
>
> * In theoretical analyses of Adam for minimization, the relative scale between the stability constant $\epsilon$ and the gradient norm is known to influence the algorithm’s dynamics in subtle and important ways. For example, [4, 5] show that the implicit regularization of Adam differs depending on whether $\epsilon$ is negligible compared with the gradient norms, and [2] divides Adam’s dynamics into two stages based on this distinction. In our work, the two aspects we study—local convergence and implicit gradient regularization—correspond exactly to these two stages: local convergence captures the regime where $\epsilon$ is not negligible, while implicit regularization captures the regime where $\epsilon$ becomes negligible.
>
> Based on above two points, we believe it is a reasonable choice to investigate the local convergence and implicit regularization aspects of Adam in min-max games.

---

> ### Author Response · Authors · 2025-11-21
> **Official Comment by Authors [2/2]**
>
> Question Part:
> ---
>  > Can the conclusions be generalized to other optimization algorithms, such as SGD or AdamW?
>
> We believe the methodology employed in the current paper can be generalized to other algorithms in min-max games and improve previous analysis.
>
> For the SGD algorithm, previous works of [3] has developed stochastic difference equations to investigate their dynamical behaviors in min-max games. However, their equations enjoys a $\mathcal{O}(h)$ global error (in the sense of weak approximation) to the orginal discrete-time algorithms. We believe using methods in the our works, we can derive new models with an improved $\mathcal{O}(h^2)$ global error, and these models can provide more accurate approximation to the discrete-time algorithms. For AdamW, the recent work of [1] investigated the implciit regularization aspects of AdamW, however, their approach are different from the approach employed in the current paper, especailly, they didn't employ the continuous-time analysis. It would be an interesting future direction to explore whether the continusous-time approach used in the current work and related works such as [2].
>
> ---
>
> **Reference:**
> ---
> [1] Xie & Li, Implicit Bias of AdamW: ℓ∞-Norm Constrained Optimization. ICML 2024
>
> [2] Cattaneo et al., On the Implicit Bias of Adam. ICML 2024
>
> [3] Compagnoni et al., SDEs for Minimax Optimization. AISTATS 2024
>
> [4] Wang et al., Does Momentum Change the Implicit Regularization on Separable Data? NeurIPS 2022
>
> [5] Zhang et al., The Implicit Bias of Adam on Separable Data. NeurIPS 2024
>
> [6] Ghosh et al., Implicit regularization in Heavy-ball momentum accelerated stochastic gradient descent. ICLR 2023

---

### Author Response · Authors · 2025-11-28

We sincerely thank all of our reviewers for the time they spent evaluating our paper and for the helpful feedback provided in their initial reviews. We have uploaded a revised version of the paper that incorporates the reviewers’ comments. All revisions are marked in blue for clarity. Notable updates include:

* Adding more discussion and comparisons with related works.
* Providing additional explanation for Assumption 4.2.
* Including numerical experiments on adversarial training to demonstrate that it does not satisfy the assumptions required for our results.

We have also attempted to address all reviewer comments and questions in our personalized responses. As the discussion period comes to an end, we look forward to seeing whether our responses have helped resolve your concerns.

---

### Meta-Review · Area_Chair_r63L · 2025-12-07

**Summary:**

This paper investigates the dynamical behavior of Adam in min-max games, specifically in comparison to its behavior in standard minimization problems. The authors argue that the momentum parameters in Adam influence its behavior in min-max games in an opposite manner to how they influence minimization. They provide both theoretical results and numerical experiments to support their claims, focusing on local convergence and implicit gradient regularization.

However, the reviewers raised significant concerns regarding the novelty, experimental scope, and generalizability of the findings. Several reviewers expressed that the technical contributions were not sufficiently novel, with some pointing out that the continuous-time approach was not a substantial departure from prior work. Additionally, the empirical results were primarily based on GANs, leading to concerns about whether the conclusions could be generalized to other types of min-max games or practical applications. There were also concerns about the deterministic nature of the analysis, with the reviewers emphasizing the importance of understanding the stochastic setting where Adam is typically applied. Despite the valuable theoretical contributions, these issues led to a recommendation for rejection.

**Reviewer Concerns:**

Novelty and Comparison to Prior Work:

Reviewers (e.g., Reviewer 3qbU and Reviewer NpU6) felt that the contributions were not sufficiently novel, particularly in comparison to existing work (e.g., Feng et al. 2025), and that the paper did not offer a significant new insight.

Rebuttal Addressed: The authors clarified how their work extends existing analyses and emphasized the unique aspects of their approach.

Still Outstanding: The novelty of the work remained in question, as some reviewers did not find enough distinction from prior research.

Generalization of Results:

Concerns were raised about whether the results could be generalized beyond GANs to other types of min-max games or adversarial settings.

Rebuttal Addressed: The authors clarified that the focus was on interaction-dominated games and that their insights could extend beyond GANs, providing additional justification for why adversarial training is not a suitable counterexample.

Still Outstanding: Some reviewers continued to express doubts about the broader applicability of the results.

Deterministic vs. Stochastic Settings:

The paper focused on a deterministic analysis, but many reviewers noted that Adam is typically used in stochastic optimization settings.

Rebuttal Addressed: The authors acknowledged this limitation and emphasized that deterministic analyses can offer useful insights, pointing to their GAN experiments as evidence of applicability to stochastic settings.

Still Outstanding: Reviewers remained concerned that the deterministic analysis did not adequately capture the dynamics of Adam in practical stochastic settings.

Experimental Validation:

Reviewers expressed concern that the experimental results were limited to GANs and did not include comparisons with state-of-the-art algorithms designed specifically for min-max games.

Rebuttal Addressed: The authors clarified that the experiments were meant to validate their theoretical findings, not to propose new state-of-the-art methods.

Still Outstanding: The empirical validation remained a point of contention, with some reviewers feeling that it was too narrow and not sufficiently comprehensive to support the broader claims.

**Reviewer Scores:**

Reviewer 3qbU: The reviewer expressed concerns about the novelty and relevance of the work, ultimately recommending rejection.

Potential Change: Even with the rebuttal, the reviewer likely would have kept their rejection recommendation due to unresolved concerns about novelty and generalizability.

Reviewer NpU6: While acknowledging the interesting theoretical contributions, this reviewer still felt that the work lacked significant novelty and that the empirical results were too limited, leading to a rejection recommendation.

Potential Change: After the rebuttal, this reviewer might have slightly softened their rejection stance but would still lean toward rejection due to the experimental limitations and lack of clear novelty.

Reviewer z4QD: This reviewer had a more positive view of the theoretical contributions but raised concerns about generalizability and experimental validation.

Potential Change: Although the rebuttal clarified some points, this reviewer would likely still lean toward rejection, given the limited experimental validation and concerns about generalizability.

Reviewer 5mRK: Gave a strong accept rating, appreciating the thorough analysis but noted that the results were not surprising to those familiar with the challenges of min-max optimization.

Potential Change: This reviewer may have reconsidered their strong accept recommendation after seeing the responses to the experimental concerns and novelty issues, leaning more toward a reject recommendation.

Reviewer 3qbU (final assessment): With their strong concerns about the lack of novelty, the limited scope of experiments, and the deterministic analysis, this reviewer would likely continue recommending reject.

---

### Decision · Program_Chairs · 2026-01-26

Reject